



# CANOPS-GRB v1.0: a new Earth system model for simulating the evolution of ocean-atmosphere chemistry over geologic timescales

Kazumi Ozaki[1,2], Devon B. Cole[3], Christopher T. Reinhard[2,3,4], and Eiichi Tajika[5]

[1]Department of Environmental Science, Toho University, Funabashi, Chiba 274-8510, Japan
[2]NASA Nexus for Exoplanet System Science (NExSS)
[3]School of Earth and Atmospheric Sciences, Georgia Institute of Technology, Atlanta, GA 30332, USA
[4]NASA Interdisciplinary Consortia for Astrobiology Research (ICAR), Alternative Earths Team, Riverside, CA, USA
[5]Department of Earth and Planetary Science, The University of Tokyo, Bunkyo-ku Tokyo 113-0033, Japan

*Correspondence to*: Kazumi Ozaki (kazumi.ozaki@sci.toho-u.ac.jp)

**Abstract**

A new version of the Earth system model of intermediate complexity (CANOPS-GRB) was developed for use in quantitatively assessing the dynamics and stability of atmospheric and oceanic chemistry over geologic timescales. The new release is designed to represent the coupled major element cycles of C, N, P, O, and S, as well as the global redox budget (GRB) in Earth's exogenic (ocean-atmosphere-crust) system, using a process-based approach. This framework provides a mechanistic model of the evolution of atmospheric and oceanic $O_2$ levels on geologic timescales and enables comparison with a wide variety of geological records to further constrain the processes driving Earth's oxygenation. A complete detailed description of the resulting Earth system model and its new features are provided. The performance of CANOPS-GRB is then evaluated by comparing a steady-state simulation under present-day conditions with a comprehensive set of oceanic data and existing global estimates of bio-element cycling. The dynamic response of the model is also examined by varying phosphorus availability in the exogenic system. CANOPS-GRB reliably simulates the short- and long-term evolution of the coupled C-N-P-$O_2$-S biogeochemical cycles and is generally applicable across any period of Earth's history given suitable modifications to boundary conditions and forcing regime. The simple and adaptable design of the model also makes it useful to interrogate a wide range of problems related to Earth's oxygenation history and Earth-like exoplanets more broadly. The model source code is available on GitHub, and represents a unique community tool for investigating the dynamics and stability of atmospheric and oceanic chemistry on long timescales.





**Short summary:**

A new biogeochemical model (CANOPS-GRB v1.0) for assessing the redox stability and dynamics of the ocean-atmosphere system on geologic timescales has been developed. In this paper, we present a full description of the model and its performance. CANOPS-GRB is a useful tool for understanding the factors regulating atmospheric $O_2$ levels and has the potential to greatly refine our current understanding of Earth's oxygenation history.

## 1 Introduction

A quarter century has passed since the first discovery of exoplanets (Mayor and Queloz, 1995). In the next quarter century, a full-scale search for signs of life—biosignatures—on Earth-like exoplanets is one of the primary objectives of the next generation of exoplanetary observational surveys (National Academies of Sciences and Medicine, 2019; The LUVOIR Team, 2019). The definition of biosignatures includes a variety of signatures that require biological activity for their origin (Des Marais et al., 2002; Lovelock, 1965; National Academies of Sciences and Medicine, 2019; Sagan et al., 1993; Schwieterman et al., 2018), but atmospheric composition has received the most interdisciplinary attention since the dawn of the search for life beyond our own planet (Hitchcock and Lovelock, 1967; Lovelock, 1972, 1965, 1975; Sagan et al., 1993) because of its potential for remote detectability. Indeed, it is likely that deciphering of exoplanetary atmospheric composition based on spectroscopic information will, at least for the foreseeable future, be our only promising means for life detection beyond our solar system. However, the detection of atmospheric composition cannot immediately answer the question of the presence or absence of a surface biosphere because significant gaps remain in our understanding of the relationships between atmospheric composition and biological activity occurring at the surface on life-bearing exoplanets. Many of these gaps arise from a lack of robust theoretical and quantitative frameworks for the emergence and maintenance of remotely detectable atmospheric biosignatures in the context of planetary biogeochemistry.

It is also important to emphasize that the abundance of atmospheric biosignature gases of living planets will evolve via an intimate interaction between life and global biogeochemical cycles of bio-essential elements across a range of timescales. Indeed, the abundances of biosignature gases such as molecular oxygen ($O_2$) and





methane ($CH_4$) in Earth's atmosphere have evolved dramatically through coevolutionary interaction with
Earth's biosphere for nearly 4 billion years—through remarkable fluctuations in atmospheric chemistry and
climate (Catling and Kasting, 2017; Catling and Zahnle, 2020; Lyons et al., 2014). To the extent that the
coupled evolution of life and the atmosphere is a universal property of life-bearing planets that maintain robust
atmospheric biosignatures, the construction of a biogeochemical framework for diagnosing atmospheric
biosignatures should be a subject of urgent interdisciplinary interest.
Establishing a mechanistic understanding of our own planet's evolutionary history is also an important
milestone for the construction of a search strategy for life beyond our solar system, as it provides the first step
towards understanding how remotely detectable biosignatures emerge and are maintained on a planetary scale.
While numerous atmospheric biosignature gases have been proposed, the most promising candidates have
been 'redox-based' species, such as $O_2$, ozone ($O_3$), and $CH_4$ (Krissansen-Totton et al., 2018; Meadows, 2017;
Meadows et al., 2018; Reinhard et al., 2017a). In particular, $O_2$ is of great interest to astrobiologists because
of its crucial role in metabolism on Earth. Thus, a considerable effort has been devoted over recent decades to
quantitatively and mechanistically understand Earth's oxygenation history. In particular, a recent surge in the
generation of empirical records for Earth's redox evolution has yielded substantial progress in our 'broad
stroke' understanding of Earth's oxygenation history and has shaped our view of biological evolution (Kump,
2008; Lyons et al., 2014). One of the intriguing insights obtained from the accumulated geochemical records
is that atmospheric $O_2$ levels might have evolved more dynamically than previously thought—our current
paradigm of Earth's oxygenation history suggests that atmospheric $O_2$ levels may have risen and then
plummeted during the early Proterozoic, then remained low (probably <10% of the present atmospheric level;
PAL) for much of the ~1 billion years leading up to the catastrophic climate system perturbations and the
initial diversification of complex life during the late Proterozoic.
The possibility of low but 'post-biotic' atmospheric $O_2$ levels during the mid-Proterozoic has important
ramifications not only for our basic theoretical understanding of long-term $O_2$ cycle stability on a planet with
biological $O_2$ production, but also for biosignature detectability (Reinhard et al., 2017a). However, our
quantitative and mechanistic understanding of the Earth's $O_2$ cycle in deep time is still rudimentary at present.



For example, one possible explanation for low atmospheric $O_2$ levels during the mid-Proterozoic is simply a
less active or smaller biosphere (Crockford et al., 2018; Derry, 2015; Laakso and Schrag, 2014; Ozaki et al.,
2019a). However, mechanisms for regulating biotic $O_2$ generation rates and stabilizing atmospheric $O_2$ levels
at low levels on billion-year timescales remain obscure. As a result, the level of atmospheric $O_2$ and its stability
during the early-mid Proterozoic are the subject of vigorous debate (Bellefroid et al., 2018; Canfield et al.,
2018; Cole et al., 2016; Planavsky et al., 2018; Planavsky et al., 2016; Tang et al., 2016; Zhang et al., 2016).
Perhaps even more importantly, a relatively rudimentary quantitative framework for probing the dynamics
and stability of the oxygen cycle leads to the imprecision of geochemical reconstructions of ocean-atmosphere
$O_2$ levels.
Planetary atmospheric $O_2$ levels are governed by a kinetic balance between sources and sinks. Feedback arises
because the response of source/sink fluxes to changes in atmospheric $O_2$ levels is intimately interrelated to
each other. Since the biogeochemical cycles of C, N, P, and S exert fundamental control on the redox budget
through non-linear interactions and feedback mechanisms, a mechanistic understanding of these
biogeochemical cycles is critical for understanding Earth's $O_2$ cycle. However, the wide range of timescales
that characterize C, N, P, $O_2$ and S cycling through the reservoirs of the Earth system makes it difficult to fully
resolve the mechanisms governing the dynamics and stability of atmospheric $O_2$ levels from geologic records.
From this vantage, developing new quantitative tools that can explore biogeochemical cycles under conditions
very different from those of the present Earth is an important pursuit.

This study is motivated by the conviction that an ensemble of 'open' Earth system modeling frameworks with
explicit and flexible representation of the coupled C-N-P-$O_2$-S biogeochemical cycles will ultimately be
required to fully understand the dynamics and stability of Earth's $O_2$ cycle and its controlling factors. In
particular, a coherent mechanistic framework for understanding the global redox ($O_2$) budget (GRB) is critical
for filling remaining gaps in our understanding of Earth's oxygenation history and the cause-and-effect
relationships with an evolving biosphere. Here, we develop a new Earth system model, named CANOPS-
GRB, which implements the coupled biogeochemical cycles of C-N-P-$O_2$-S within the Earth's surface system
(ocean-atmosphere-crust). The model design (such as the complexity of the processes and spatial-temporal



resolution of the model) is constrained by the requirement of simulation length (>100 million years) and actual
model run-time. A lack of understanding of biogeochemistry in deep-time and availability and quality of
geologic records also limit the model structure. With this in mind, we aim for a comprehensive, simple, yet
realistic representation of biogeochemical processes in the Earth system, yielding a unique tool for
investigating coupled biogeochemical cycles within the Earth system over a wide range of time scales. We
have placed particular emphasis on the development of a global redox budget in the ocean-atmosphere-crust
system given its importance in the secular evolution of atmospheric $O_2$ levels. CANOPS-GRB is an initial
step towards developing the first large-scale biogeochemistry evolution model suited for the wide range of
redox conditions, including explicit consideration of the coupled C-N-P-$O_2$-S cycles and the major biogenic
gases in planetary atmospheres ($O_2$ and $CH_4$).



## 2 Model description

Here we present a full description of a new version of the Earth system model CANOPS—CANOPS-GRB—which is designed to facilitate simulation for a wide range of biogeochemical conditions so as to permit quantitative examination of evolving ocean-atmosphere chemistry throughout Earth's history. Below we first describe the concept of model design (Sect. 2.1). Next, we describe the overall structure of the model and the basic design of global biogeochemical cycles (Sect. 2.2 and 2.3). That is followed by a detailed description of each sub-model.

### 2.1 CANOPS-GRB in the hierarchy of biogeochemical models

A full understanding of Earth's evolving $O_2$ cycle requires a quantitative framework that includes mechanistic links between biological metabolism, ocean-atmosphere chemistry, and geologic processes. Such a framework must also represent the feedbacks between ocean-atmosphere redox state and biogeochemical cycles of redox-dependent bio-essential elements. Over recent decades, considerable progress has been made in quantifying the feedbacks between atmospheric $O_2$ levels and the coupled C-N-P-$O_2$-S biogeochemical cycles over geological timescales (Belcher and McElwain, 2008; Berner, 2004b; Betts and Holland, 1991; Bolton et al., 2006; Colman et al., 2000; Holland, 1978; Lasaga and Ohmoto, 2002; Slomp and Van Cappellen, 2007; Van Cappellen and Ingall, 1994). Refinements to our understanding of mechanisms regulating Earth's surface redox state have been implemented in low-resolution box models where the ocean-atmosphere system is expressed by a few boxes (Alcott et al., 2019; Bergman et al., 2004; Claire et al., 2006; Goldblatt et al., 2006; Handoh and Lenton, 2003; Laakso and Schrag, 2014; Lenton and Watson, 2000a, b; Petsch and Berner, 1998; Van Cappellen and Ingall, 1996). These models offer insights into basic system behavior and can illuminate the fundamental mechanisms that exert the most leverage on biogeochemical cycles because of their simplicity, transparency and low computational demands. However, these model architectures also have important quantitative limitations. For example, with low spatial resolution the modeler needs to assume reasonable (but *a priori*) relationships relating to internal biogeochemical cycles in the system. For instance, because of a lack of high vertical resolution, oceanic box models (Knox and McElroy, 1984; Sarmiento and Toggweiler, 1984; Siegenthaler and Wenk, 1984) usually overestimate the sensitivity of atmospheric $CO_2$ levels to biological activity at high-latitude surface ocean relative to projections by general circulation models (Archer et al.,





2000). Oceanic biogeochemical cycles and chemical distributions are also characterized by strong vertical and
horizontal heterogeneities, which have the potential to affect the strength of feedback processes (Ozaki et al.,
2011). In other words, the low-resolution box modeling approach might overlook the strength and response
of the internal feedback loops. Thus, the development of an ocean model with high resolution of ocean interior
and reliable representation of water circulation is required to investigate the mechanisms controlling
atmospheric $O_2$ levels under conditions very different from those of the modern Earth.

In the last decade, comprehensive Earth system models of intermediate complexity (EMICs) have also been
developed and extended to include ocean sediments and global C cycling (Lord et al., 2016; Ridgwell and
Hargreaves, 2007). Such models can be integrated over tens of thousands of years, allowing experimentation
with hypothetical dynamics of global biogeochemical cycles in the geological past (Olson et al., 2016;
Reinhard et al., 2020). However, a key weakness of existing EMICs is the need to parameterize (or ignore)
boundary (input/output) fluxes—either due to the computational expense of explicitly specifying boundary
conditions or due to poorly constrained parameterizations. For example, the oceanic P cycle is usually treated
as a closed system, limiting the model's applicability to timescales less than the oceanic P residence time
(~15−20 kyr). Further, boundary conditions such as continental configuration and oceanic bathymetry are
variable or poorly constrained in deep time and the use of highly complex models is difficult to justify the
computational cost. Finally, exploration of hypotheses concerning the biogeochemical dynamics in deep time
often require large model ensembles across broad parameter space given the scope of uncertainty. This makes
the computational cost of EMICs intractable at present for many key questions.

The CANOPS-GRB model is designed to capture the major components of Earth system biogeochemistry on
timescales longer than a millennium, but is simple enough to allow for runs on the order of a billion model
years. The model structure is also designed so that the model captures the essential biogeochemical processes
regulating the global $O_2$ budget, while keeping the calculation cost as moderate as possible. For example, the
simple relationships of biogeochemical transport processes at the interface of Earth system (hydrogen escape
to space, early diagenesis in marine sediments, and weathering) are employed based on the systematic
application of 1-D models  in previous studies (Bolton et al., 2006; Claire et al., 2006; Daines et al., 2017;


Middelburg et al., 1997; Wallmann, 2003a), providing a powerful, computationally efficient means for
exploring the Earth system under a wide range of conditions. The resultant CANOPS-GRB model can be run
on a standard personal computer on a single CPU with an efficiency of approximately 6 million model years
per CPU hour. In other words, model runs in excess of $10^9$ model years are tractable with modest wall times
(approximately 7 days). The model is thus not as efficient as simple box models, but is highly efficient relative
to EMICs, making sensitivity experiments and exploration of larger parameter space over a billion years
feasible, particularly with implementation on a high-performance computing cluster (see Cole et al., 2022).
CANOPS-GRB thus occupies a unique position within the hierarchy of global biogeochemical cycle models,
rendering it a useful tool for the development of more comprehensive, low- to intermediate-complexity models
of Earth system on very long timescales.

**2.2 Overall model structure**

The overall structure of the model is shown in Fig. 1. The model consists of ocean, atmosphere and
sedimentary reservoirs. The core of the model is an ocean model, comprising a high-resolution 1-D
intermediate-complexity box model of the global ocean (Sect. 2.4). The ocean model is coupled to a
parameterized marine sediment module (Sect. 2.4.4) and a one box model of the atmosphere (Sect. 2.6). The
atmospheric model includes $O_2$ and $CH_4$ as chemical components, and abundances of these molecules are
calculated based on the mass balance between sources and sinks (e.g., biogenic fluxes of $O_2$ and $CH_4$ from the
ecosystems and photochemical reactions). The net air-sea gas exchange of chemical species ($O_2$, $H_2S$, $NH_3$
and $CH_4$) is quantified according to the stagnant film model (Kharecha et al., 2005; Liss and Slater, 1974)
(Sect. 2.4.5). The ocean and atmosphere models are embedded in a 'rock cycle' model that simulates the
evolution of sedimentary reservoir sizes on geologic timescales (Sect. 2.5). Three sedimentary reservoirs
(organic carbon, *ORG*; pyrite sulfur, *PYR*; and gypsum sulfur, *GYP*) are considered in the CANOPS-GRB
model. These reservoirs interact with the ocean-atmosphere system through weathering, outgassing, and burial.




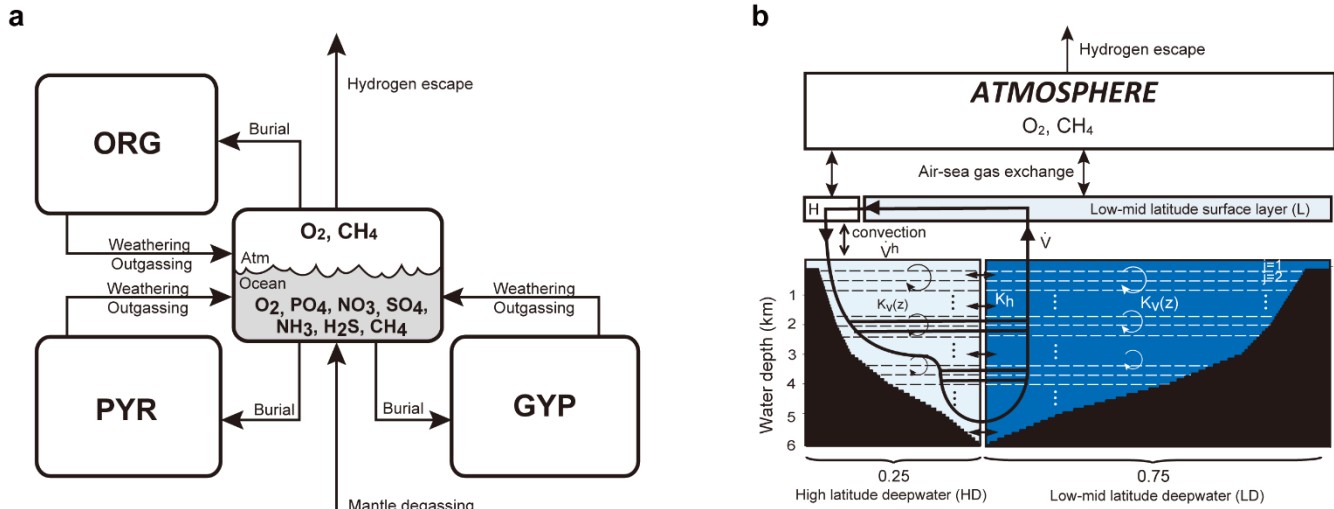

**Figure 1** CANOPS-GRB model configuration. (**a**) The schematic of material cycles in the surface (ocean-atmosphere-crust) system. Three sedimentary reservoirs, organic carbon (*ORG*), pyrite sulfur (*PYR*), and gypsum sulfur (*GYP*), are considered. Sedimentary reservoirs interact with the ocean-atmosphere system via weathering, volcanic degassing, and burial. No interaction with the mantle is included, except for the input of reduced gases from the mantle. Total mass of sulfur is conserved in the surface system. (**b**) Schematic of ocean and atmosphere modules. "L" and "H" denote the low-mid latitude mixed surface layer and high-latitude surface layer, respectively. An ocean area of 10% is assumed for H. River flux for each region is proportional to the areal fraction. Ocean interior is divided into two sectors, high-latitude deep water (HD) and low-mid latitude deep water (LD), which are vertically resolved. The area of HD is 25% of the whole ocean. The deep overturning circulation, $\dot{V}$, equals the poleward flow in the model surface layer (from L to H). $K_v^l(z)$ and $K_v^h(z)$ are the vertical eddy diffusion coefficients in the LD and HD regions, respectively. $K_{hor}$ and $\dot{V}_h$ are the horizontal diffusion coefficient and polar convection, respectively. The black hatch represents the seafloor topography assumed. The parameters regarding geometry and water transport are tabulated in Table 3.

The ocean model is a vertically resolved transport-reaction model of the global ocean, which was originally developed by Ozaki et al. (2011) and Ozaki and Tajika (2013). The model consists of 122 boxes across two regions; a low-mid latitude region and a high-latitude region (Fig. 1b). The ocean model describes water transport processes as exchange fluxes between boxes and via eddy diffusion terms. More specifically, ocean circulation is modelled as an advection-diffusion model of the global ocean—a general and robust scheme that is capable of producing well-resolved modern profiles of circulation tracers using realistic parameter values (the physical set up of the model can be found in Sect. 2.4.1 and 2.4.2). The biogeochemical sub-model provides a mechanistic description of the marine biogeochemical cycles of C, P, N, $O_2$, and S (Sect. 2.4.3).





This includes explicit representation of a variety of biogeochemical processes such as biological productivity
in the sunlit surface oceans, a series of respiration pathways and secondary redox reactions under oxic and
anoxic conditions (Sect. 2.4.3), and deposition, decomposition, and burial of biogenic materials in marine
sediments (Sect. 2.4.4), allowing a mechanistically based examination of biogeochemical processes. The suite
of metabolic reactions included in the model is listed in Table 1. Ocean biogeochemical tracers considered in
the CANOPS-GRB model are phosphate ($PO_4^{3-}$), nitrate ($NO_3^-$), total ammonia ($\Sigma NH_3$), dissolved oxygen
($O_2$), sulfate ($SO_4^{2-}$), total sulfide ($\Sigma H_2S$), and methane ($CH_4$). Ocean model performance was tested for the
modern-day ocean field observational data (Sect. 3). Simulation results were also compared to previously
published integrated global flux estimates.

Table 1: Biogeochemical reactions considered in the CANOPS-GRB model.

| Process | Stoichiometry[*] | Reaction # |
|---|---|---|
| Ammonia assimilation | $\alpha CO_2 + \beta NH_4^+ + H_3PO_4 + \alpha H_2O \rightarrow OM + \alpha O_2$ | R1 |
| Nitrate assimilation | $\alpha CO_2 + \beta NO_3^- + H_3PO_4 + (\alpha+\beta)H_2O + 2\beta H^+ \rightarrow OM + (\alpha+2\beta)O_2$ | R2 |
| Nitrogen fixation | $N_2 + 5H_2O \rightarrow 2NH_4^+ + 2OH^- + \frac{3}{2}O_2$ | R3 |
| Aerobic respiration | $OM + \alpha O_2 \rightarrow \alpha CO_2 + \beta NH_4^+ + H_3PO_4 + \alpha H_2O$ | R4 |
| Denitrification | $OM + \frac{4}{5}\alpha NO_3^- + \frac{4}{5}\alpha H^+ \rightarrow \alpha CO_2 + \beta NH_4^+ + H_3PO_4 + \frac{7}{5}\alpha H_2O + \frac{2}{5}\alpha N_2$ | R5 |
| Sulfate reduction | $OM + \frac{1}{2}\alpha SO_4^{2-} + \alpha H^+ \rightarrow \alpha CO_2 + \beta NH_4^+ + H_3PO_4 + \alpha H_2O + \frac{1}{2}\alpha H_2S$ | R6 |
| Methanogenesis | $OM \rightarrow \frac{1}{2}\alpha CO_2 + \frac{1}{2}\alpha CH_4 + \beta NH_4^+ + H_3PO_4$ | R7 |
| Nitrification | $NH_4^+ + 2O_2 \rightarrow NO_3^- + H_2O + 2H^+$ | R8 |
| Aerobic H$_2$S oxidation[†] | $\Sigma H_2S + 2O_2 \rightarrow SO_4^{2-} + 2H^+$ | R9 |
| Aerobic CH$_4$ oxidation | $CH_4 + 2O_2 \rightarrow CO_2 + 2H_2O$ | R10 |
| Anaerobic CH$_4$ oxidation | $CH_4 + SO_4^{2-} \rightarrow HS^- + HCO_3^- + H_2O$ | R11 |
| Photooxidation of CH$_4$ | $CH_4 + 2O_2 \rightarrow CO_2 + 2H_2O$ | R12 |
| Hydrogen escape to space | $CH_4 + O_2 \rightarrow CO_2 + 4H\uparrow$ | R13 |

[*]OM denotes organic matter, $(CH_2O)_\alpha(NH_4^+)_\beta H_3PO_4$
[†]$\Sigma H_2S = H_2S + HS^-$





The CANOPS model has been extended and altered a number of times since first publication. The description of biogeochemical cycles in the original version of CANOPS (Ozaki and Tajika, 2013; Ozaki et al., 2011) does not include the S and $CH_4$ cycles because of their aims to investigate the conditions for the development of oceanic anoxia/euxinia on timescales less than a million years during the Phanerozoic. More recently, Ozaki et al. (2019) implemented an open system modeling approach for the global S and $CH_4$ cycles, enabling quantitative analysis of global redox budget for given atmospheric $O_2$ levels and crustal reservoir sizes. In this version of CANOPS atmospheric $O_2$ levels and sedimentary reservoirs are treated as boundary conditions because imposing them simplifies the model and significantly reduces computing time. However, this approach does not allow exploration of the dynamic behavior of atmospheric $O_2$ in response to other boundary conditions. In the newest version presented here, significant improvements in the representation of global biogeochemistry were achieved by (1) an explicit calculation of atmospheric $O_2$ levels based on atmospheric mass balance (Sect. 2.6), (2) expansion of the model framework to include secular evolution of sedimentary reservoirs (Sect. 2.5.5), and (3) simplification of the global redox budget between the surface (ocean-atmosphere-crust) system and the mantle (Sect. 2.3.5). These improvements are in line with the requirement of an 'open' Earth system model, which is necessary for a systematic, quantitative understanding of Earth's oxygenation history.

## 2.3 Global biogeochemical cycles

We construct a comprehensive biogeochemical model in order to investigate the interaction between dynamic behaviors of Earth's oxygenation history and its biogeochemical processes, as well as redox structure of the ocean. Here we provide the basic implementation of global biogeochemical cycles of C, P, N, and S, with particular emphasis on processes of mass exchange between reservoirs that play a critical role in global redox budget (Fig. 2). Our central aim here is to make the overall design of biogeochemical cycles clear. The details of each sub-model are provided in the following sections.





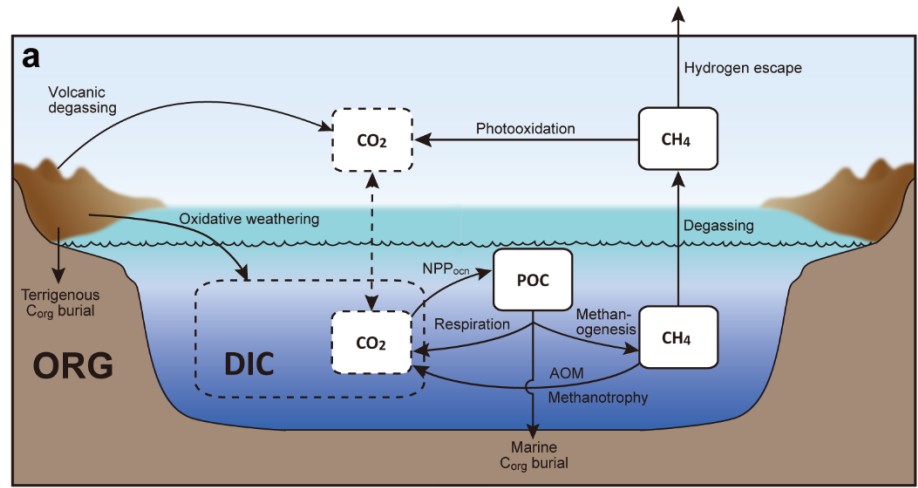

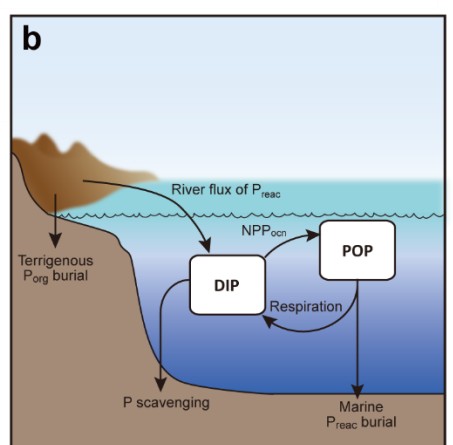

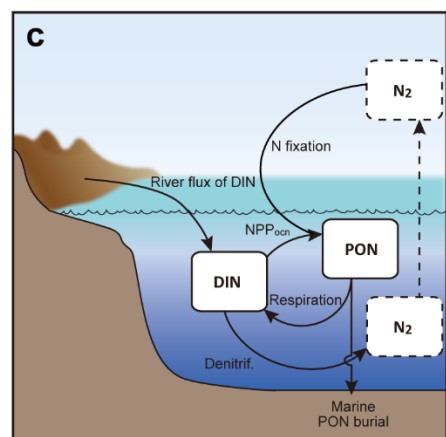

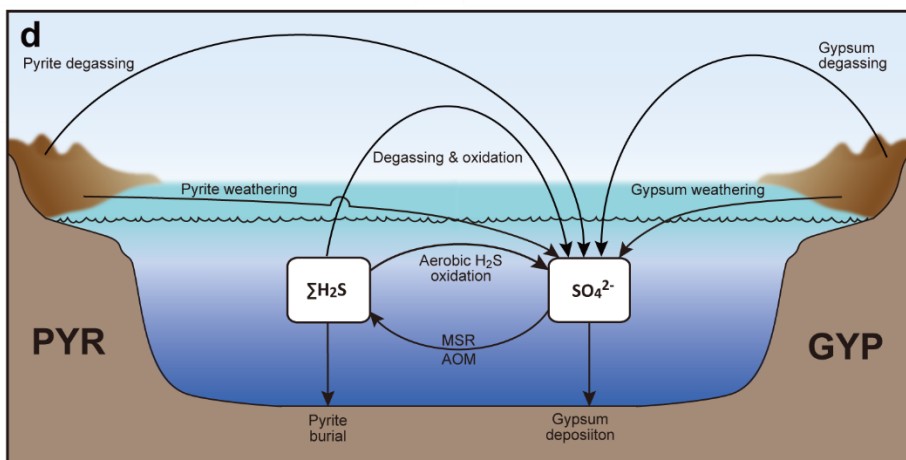





**Figure 2** Schematics of global biogeochemical cycles. (**a**) Global C cycle. The primary source of C for the ocean-atmosphere system is volcanic degassing and oxidative weathering of sedimentary organic carbon, whereas primary sink is burial of marine and terrigenous organic matter into sediments. Inorganic carbon reservoirs (depicted as dashed boxes) and DOC are not considered. $NPP_{ocn}$ = marine net primary production. DIC = dissolved inorganic carbon. POC = particulate organic carbon. MSR = microbial sulfate reduction. AOM = anoxygenic oxidation of methane. CANOPS-GRB includes $CH_4$ generation via methanogenesis and its oxidation reactions via methanotrophy and AOM in the ocean interior, as well as $CH_4$ degassing flux to the atmosphere and its photooxidation. The rates of $CH_4$ photooxidation and hydrogen escape to space are calculated based on parameterizations proposed by previous studies (Claire et al., 2006; Goldblatt et al., 2006). Note that $CH_4$ flux from land biosphere is not shown here. (**b**) Global P cycle schematic. Weathering of reactive P ($P_{reac}$) is the ultimate source, whereas burial in sediments is the primary sink. A part of the weathered P is buried as terrigenous organic P, and the remaining is delivered to the ocean. The redox-dependent P burial in marine sediments is modelled by considering three phases (organic P, Fe-sorbed P, and authigenic P). DIP = dissolved inorganic phosphorus. POP = particulate organic phosphorus. The hypothetical P scavenging via Fe-species in anoxic-ferruginous waters is depicted, but it is not modelled in our standard model configuration. (**c**) Global N cycle schematic. Two inorganic nitrogen species (ammonium and nitrate), which are lumped into DIN (dissolved inorganic nitrogen), are transformed each other via denitrification and nitrification. The primary source is nitrogen fixation and riverine flux, whereas primary sink is denitrification and burial in marine sediments. PON = particulate organic nitrogen. The nitrogen weathering/riverine flux is assumed to be equal to the burial flux so that there is no mass imbalance in global N budget. Aeolian delivery of N from continent to the ocean is not included. (**d**) Global S cycle schematic. Two sedimentary reservoirs (pyrite sulfur, PYR, and gypsum sulfur, GYP) and two sulfur species ($SO_4^{2-}$ and $\Sigma H_2S$) in the ocean are transformed each other via volcanic outgassing, weathering, burial, MSR, AOM, and sulfide oxidation reactions. Weathering and volcanic inputs are the primary source of S to the ocean, and burial of pyrite and gypsum in marine sediments is the primary sink. It is assumed that hydrogen sulfide escaping from the ocean to the atmosphere is completely oxidized and returns to the ocean as sulfate. The organic sulfur cycle is ignored in this study.

### 2.3.1 Carbon cycle

The CANOPS-GRB model includes particulate organic carbon (POC), atmospheric $CH_4$, dissolved $CH_4$ in the ocean, and sedimentary organic carbon (ORG) as carbon reservoirs (Fig. 2a). Atmospheric $CO_2$, dissolved inorganic carbon (DIC), and dissolved organic carbon (DOC) are not explicitly modelled in the current version of the model. The primary sources of carbon for the ocean-atmosphere system are volcanic degassing and oxidative weathering of sedimentary organic carbon, while the primary sink is burial of marine and terrigenous organic matter in sediments.



**Organic carbon cycle**

The biogeochemical model is driven by the cycling of the primary nutrient phosphorus, which is assumed to be the ultimate limiting factor for biological productivity (see Sect. 2.4.3). Previous versions of CANOPS do not take into account the impact of the activity of terrestrial ecosystem on the global $O_2$ budget. In the CANOPS-GRB model, we improve on this by evaluating the activity levels of terrestrial and marine ecosystems separately: The global net primary production (NPP), $J_{NPP}$ (in terms of organic C), is given as a sum of the oceanic ($J_{NPP}^{ocn}$) and terrestrial ($J_{NPP}^{lnd}$) NPP:

$$J_{NPP} = J_{NPP}^{ocn} + J_{NPP}^{lnd}.$$
(1)

Biological production in the ocean surface layer depends on P availability while nutrient assimilation efficiency is assumed to be lower in the high latitude region (Sect. 2.4.3). Terrestrial NPP is affected by the atmospheric $O_2$ level (Sect. 2.5.1). In this study, the flux (in terms of moles per year) is expressed with a capital $J$, whereas the flux density (in terms of moles per square meter per year) is expressed with a lowercase $j$.

In our standard model configuration, oceanic primary production follows canonical Redfield stoichiometry (C:N:P = 106:16:1) (Redfield et al., 1963). Flexible C:N:P stoichiometry of particulate organic matter (POM) can be explored by changing a user-flag. Nutrients (P and N) are removed from seawater in the photic zone via biological uptake, and exported as POM to deeper aphotic layers. The exported POM sinks through the water column with a speed of $v_{POM}$ (the reference value is 100 m d$^{-1}$). As it settles through water column, POM is subject to decomposition via a series of respiration pathways dependent on the redox state of proximal seawater (Sect. 2.4.3). This gives rise to the release of dissolved constituent species back into seawater. Within each layer a fraction of POM is also intercepted by a sediment layer at the bottom of each water depth. Fractional coverage of every ocean layer by seafloor is calculated based on the prescribed bathymetry (Sect. 2.4.1). Settling POM reaching the seafloor undergoes diagenetic alteration (releasing additional dissolved species into seawater) and/or permanent burial. The ocean model has 2×60 sediment segments, and for each segment the rates of organic matter decomposition and burial are calculated by semi-empirical relationships extracted from ocean sediment data and 1-D modelling of early diagenesis (Sect. 2.4.4). Specifically, the





organic C ($C_{org}$) burial at each water depth is calculated based on the burial efficiency ($BE_{org}$), which is defined
as the fraction of POC buried in sediments relative to that deposited on the seafloor at each water depth and
is also a function of sedimentation rate and bottom water $O_2$ levels. Organic matter not buried is subject to
decomposition.

The key biogeochemical fluxes of our reference state (mimicking the present condition) are summarized in
Table 2. The reference value for burial rate of terrigenous $C_{org}$ is set at 3 Tmol C $yr^{-1}$, assuming that burial of
terrigenous organic matter accounts for ~20% of the total burial. Combined with the burial rate of marine $C_{org}$
in our standard run, the total burial rate is 14.3 Tmol C $yr^{-1}$, representing the dominant $O_2$ source flux to the
modern ocean-atmosphere system. At steady-state, this is balanced by oxidative weathering and volcanic
outgassing of sedimentary $C_{org}$: The reference value of oxidative weathering of organic matter is determined
as 13.0 Tmol C $yr^{-1}$ based on the global $O_2$ budget (Sect. 2.3.5). Previous versions of CANOPS (Ozaki et al.,
2019a) treat sedimentary reservoirs as a boundary condition. This model limitation is removed in the
CANOPS-GRB model—the reservoir size of sedimentary $C_{org}$ (*ORG*) freely evolves based on the mass
balance through burial, weathering, and volcanic outgassing (Sect. 2.5.5). We adopted an oft-quoted value of
1250 Emol (E = $10^{18}$) for our reference value of the *ORG*, based on literature survey (Berner, 1989; Garrels
and Lerman, 1981).





Table 2: Key biogeochemical fluxes obtained from the reference run. * denotes the reference value. Tmol =
$10^{12}$ mol.

| Fluxes | Label | Value | Comments |
|---|---|---|---|
| **Carbon cycle (Tmol C yr$^{-1}$)** | | | |
| Marine NPP | $J_{NPP}^{ocn,*}$ | 3794 | Simulated (Eq. 24) |
| Terrestrial NPP | $J_{NPP}^{lnd,*}$ | 5000 | Prescribed (Prentice et al., 2001) |
| Marine organic C burial | $J_{org}^{b,ocn,*}$ | 11.28 | Simulated (Eq. 40) |
| Terrestrial organic C burial | $J_{org}^{b,lnd,*}$ | 3.0 | Prescribed (This study) |
| Oxidative organic C weathering | $J_{org}^{w,*}$ | 13.03 | Tuned (Eq. 15) |
| Organic C degassing | $J_{org}^{m,*}$ | 1.25 | Prescribed (Bergman et al., 2004) |
| **Phosphorus cycle (Tmol P yr$^{-1}$)** | | | |
| Reactive P weathering | $J_P^{w,*}$ | 0.158 | Eqs. (4, 84) |
| Terrestrial organic P burial | $J_P^{b,lnd,*}$ | 0.003 | Eq. (85) |
| Riverine reactive P flux | $J_P^{r,*}$ | 0.155 | Tuned (This study) |
| Marine organic P burial | $J_{Porg}^{b,*}$ | 0.0438 | Simulated (Eq. 51) |
| Fe-sorbed P burial | $J_{PFe}^{b,*}$ | 0.0323 | Simulated (Eq. 53) |
| Ca-bound P burial | $J_{PCa}^{b,*}$ | 0.0788 | Simulated (Eq. 54) |
| **Nitrogen cycle (Tg N yr$^{-1}$)** | | | |
| Nitrogen fixation | $J_{Nfix}^{*}$ | 180.5 | Simulated |
| Denitrification in the water column | $J_{deni}^{wc,*}$ | 102.5 | Simulated |
| Benthic denitfirication | $J_{deni}^{sed,*}$ | 62.4 | Simulted (Eq. 55) |
| Marine organic N burial | $J_{Norg}^{b,*}$ | 15.8 | Simulated (Eq. 56) |
| Organic N weathering | $J_{Norg}^{w,*}$ | 15.8 | $= J_{Norg}^{b,*}$ |
| **Sulfur cycle (Tmol S yr$^{-1}$)** | | | |
| Pyrite weathering | $J_{pyr}^{w,*}$ | 1.0 | Prescribed (This study) |
| Gypsum weathering | $J_{gyp}^{w,*}$ | 1.6 | Prescribed (This study) |
| Pyrite degassing | $J_{pyr}^{m,*}$ | 0.3 | Prescribed (This study) |
| Gypsum degassing | $J_{gyp}^{m,*}$ | 0.5 | Prescribed (This study) |
| Pyrite burial | $J_{pyr}^{b,*}$ | 1.3 | Simulated |
| Gypsum burial | $J_{gyp}^{b,*}$ | 2.1 | Prescribed (This study) |






**Methane cycle**

The ocean model includes biogenic $CH_4$ generation via methanogenesis and its oxidation reactions via methanotrophy and anoxygenic oxidation of methane (AOM) in the ocean interior (R10 and R11 in Table 1), as well as $CH_4$ degassing flux to the atmosphere. The land model also calculates the biogenic $CH_4$ flux from the terrestrial ecosystem to the atmosphere using a transfer function (Sect. 2.5.2). The abundance of $CH_4$ in the atmosphere is explicitly modelled as a balance of its source (degassing from marine and terrestrial ecosystems) and sink (photooxidation and hydrogen escape), where $CH_4$ sink fluxes are calculated according to parameterized $O_2$ dependent functions proposed by previous studies. More specifically, the oxidation rate of $CH_4$ in the upper atmosphere is calculated based on the empirical parameterization obtained from a 1-D photochemistry model (Claire et al., 2006). The rate of hydrogen escape to space is evaluated with the assumption that it is diffusion limited and that $CH_4$ is a major H-containing chemical compound carrying hydrogen to the upper atmosphere (Goldblatt et al., 2006). No continental abiotic or thermogenic $CH_4$ fluxes are taken into account, because previous estimates of the modern fluxes are negligible relative to the biogenic flux, although we realize that it could have played a role in the global redox budget (<0.3 Tmol yr$^{-1}$; Fiebig et al., 2009). We also note that the current version of the model does not include the possibility of aerobic $CH_4$ production in the sea (Karl et al., 2008). Our reference run calculates atmospheric $CH_4$ to be 0.16 ppmv (Sect. 3.3), slightly lower than that of the preindustrial level of 0.7 ppmv (Etheridge et al., 1998; Raynaud et al., 1993), but we consider this to be within reasonable error given unknowns in the $CH_4$ cycle.

**2.3.2 Phosphorus cycle**

Phosphorus is an essential element for all life on Earth and it is regarded as the 'ultimate' bio-limiting nutrient for primary productivity on geologic time scales (Tyrrell, 1999). Thus, the P cycle plays a prominent role in regulating global $O_2$ levels. In the CANOPS-GRB model, we model the reactive (i.e., bioavailable) P ($P_{reac}$) cycling in the system and ignore non-bioavailable P. Specifically, dissolved inorganic P (DIP) and particulate organic P (POP) are explicitly modelled (Fig. 2b), whereas dissolved organic P (DOP) is ignored.





On geologic timescales, the primary source of P to the ocean-atmosphere system is continental weathering:
Phosphorus is released through the dissolution of apatite which exists as a trace mineral in silicate and
carbonate rocks (~0.1wt%; Föllmi, 1996). The total $P_{reac}$ flux via weathering, $J_P^w$, is given as follows:

$$J_P^w = f_P f_R J_P^{w,*},$$  (2)

where * denotes the reference value, and $f_P$ and $f_R$ are parameters that control the availability of P in the system.
Specifically, $f_R$ is a global erosion factor representing the impact of tectonic activity on total terrestrial
weathering rate, and $f_P$ represents the availability of $P_{reac}$, which is used in a sensitivity experiment to assess
the response of atmospheric $O_2$ levels to changing $P_{reac}$ availability (Sect. 4.1). A fraction of the weathering
flux $J_P^w$ is removed via burial on land, while the remainder is transported to the ocean (Sect. 2.5.2):

$$J_P^{b,lnd} = k_{11} V J_P^w,$$  (3)

$$J_P^r = (1 - k_{11} V) J_P^w$$  (4)

where $J_P^{b,lnd}$ and $J_P^r$ denote the burial rate of terrigenous organic P and riverine $P_{reac}$ flux to the ocean,
respectively, $k_{11}$ is a reference value for the fraction of the total P flux removed by the terrestrial biosphere,
and $V$ denotes the vegetation mass normalized to the modern value. These treatments are based on the Earth
system box model COPSE (Bergman et al., 2004; Lenton et al., 2016; Lenton et al., 2018; Lenton and Watson,
2000b) which has been extensively tested and validated against geologic records during the Phanerozoic. In
the CANOPS-GRB model, $J_P^r$ is tuned so that modelled oceanic P inventory of the reference state is consistent
with modern observations of the global ocean (Sect. 3.2.4). Our resulting tuned value is 0.155 Tmol P yr$^{-1}$
falling in the mid-range of published estimates of 0.11–0.33 Tmol P yr$^{-1}$, although previous estimates of the
riverine $P_{reac}$ flux show large uncertainty (Sect. 3.2.4).

Since atmospheric P inputs are equivalent to less than 10% of the continental P supply to the modern oceans
and much of this flux is not bioavailable (Graham and Duce, 1979), we neglect the aeolian flux in this study.
Therefore, riverine input is the primary source of $P_{reac}$ to the ocean. We highlight that open-system modelling
is crucial for realistic simulations of ocean biogeochemistry on timescales longer than the residence time of P
in the ocean (15–20 kyr for the modern ocean) (Hotinski et al., 2000), and in this framework the riverine input





of $P_{reac}$ must be balanced over the long-term by loss to sediments via burial. The change in total marine $P_{reac}$
inventory, $M_P$, is given as follows:

$$\frac{dM_P}{dt} = J_P^r - J_P^{b,ocn},$$  (5)

where $J_P^{b,ocn}$ denotes the total burial flux of $P_{reac}$ in the marine system which is the sum of the burial fluxes of
three reactive phases, i.e. organic P ($P_{org}$), Fe-sorbed P ($P_{-Fe}$), and Ca-bound P ($P_{-Ca}$) (Sect. 2.4.4):

$$J_P^{b,ocn} = J_{Porg}^b + J_{P-Fe}^b + J_{P-Ca}^b.$$  (6)

$O_2$-dependent P burial is taken into account using empirical relationships from previous studies (Slomp and
Van Cappellen, 2007; Van Cappellen and Ingall, 1994, 1996). The burial of $P_{org}$ at each water depth is a
function of burial efficiency, which is controlled by the burial efficiency of organic matter, C/P stoichiometry
of POM, sedimentation rate and bottom water [$O_2$]. We note that the strength of anoxia-induced P recycling
in marine sediments is very poorly constrained, especially in the Precambrian oceans (Reinhard et al., 2017b).
Recent studies also suggest that the P retention potential in marine sediments could be affected not only by
bottom water $O_2$ levels but by redox states (sulfidic vs. ferruginous) and the $Ca^{2+}$ concentration of bottom
waters (Zhao et al., 2020). These are fruitful topics for future research.

We do not explicitly account for P removal via hydrothermal processes, because it is estimated that this
contribution is secondary in the modern marine P cycle (0.014–0.036 Tmol P yr$^{-1}$; Wheat et al., 1996; Wheat
et al., 2003). We note, however, that the hydrothermal contribution to the total P budget in the geologic past
remains poorly constrained. We also note that in anoxic, ferruginous oceans, P scavenging by Fe-minerals
could also play an important role in controlling P availability and the overall budget (Derry, 2015; Laakso and
Schrag, 2014; Reinhard et al., 2017b). Modern observations (Dellwig et al., 2010; Shaffer, 1986; Turnewitsch
and Pohl, 2010) and modeling efforts (Yakushev et al., 2007) of the redoxcline in the Baltic Sea and the Black
Sea suggest an intimate relationship between Mn, Fe, and P cycling. Trapping efficiencies of DIP by settling
authigenic Fe and Mn-rich particles were found to be as high as 0.63 (the trapping efficiency is defined as the
downward flux of P in Mn-, and Fe-oxides divided by the upward flux of DIP) (Turnewitsch and Pohl, 2010).
Although coupled Mn-Fe-P dynamics might have been a key aspect of the biogeochemical dynamics in the
Precambrian oceans, we exclude this process in our standard model due to poor constraints and provide a clear



and simplified picture of basic model behavior. The key features between the P availability and atmospheric
$O_2$ levels are explored by changing $f_P$ in this study (Sect. 4).

### 2.3.3 Nitrogen cycle

In the CANOPS-GRB model, two dissolved inorganic nitrogen (DIN) species (total ammonium $\Sigma NH_4^+$ and
nitrate $NO_3^-$) and particulate organic nitrogen (PON) are explicitly calculated (Fig. 2c). Atmospheric nitrogen
gas is assumed to never limit biospheric carbon fixation, and is not explicitly calculated. Dissolved organic N
(DON) and terrestrial N cycling (e.g., N fixation by terrestrial ecosystems and riverine-terrigenious organic N
transfer) are ignored.

In the surface ocean N assimilation via nitrate and ammonium depends on the availability of these compounds.
If the N required for sustaining a given level of biological productivity is not available, the additional N
required is assumed to be provided by atmospheric $N_2$ via nitrogen fixers. The ocean model explicitly
calculates denitrification and nitrification reactions in the water column and marine sediments (R5 and R8 in
Table 1). The benthic denitrification rate is estimated using a semi-empirical parameterized function obtained
from a 1-D early diagenetic model (see Sect. 2.4.4), while nitrification is modelled as a single step reaction
(R8). $N_2O$ and its related reactions, such as anammox, are not currently included.

The oceanic N cycle is open to external inputs of nitrogen. While the ultimate source of N to the ocean-
atmosphere system is weathering of organic N, nitrogen fixation represents the major input flux to the ocean
with the capacity to compensate for N loss due to denitrification. The time evolution of DIN inventory, $M_N$,
in the ocean can be written as follows:

$$\frac{dM_N}{dt} = \left( J_{Nfix} - J_{deni}^{wc} - J_{deni}^{sed} \right) + \left( J_{Norg}^{w} - J_{Norg}^{b} \right), \tag{7}$$

where $J_{Nfix}$ denotes the N fixation rate, and $J_{deni}^{wc}$ and $J_{deni}^{sed}$ are denitrification rates in the water column and
sediments, respectively. The first set of terms on the right-hand side represent the internal N cycle in the ocean-
atmosphere system, while the second set of terms represent the long-term N budget which interacts with
sedimentary reservoir. Ultimately, loss of fixed N from the ocean-atmosphere system only occurs via burial





of organic N (N$_{\text{org}}$) in sediments, $J_{\text{Norg}}{}^{\text{b}}$. This loss is compensated for by continental weathering, $J_{\text{Norg}}{}^{\text{w}}$, which is assumed to be equal to the burial rate of N$_{\text{org}}$ so that the N cycle has no impact on the global redox budget. In the current version of the model, we ignore aeolian flux and all riverine N fluxes other than weathering since these are minor relative to N fixation (Wang et al., 2019). As a result, modelled N fixation required for oceanic N balance can be regarded as an upper estimate.

### 2.3.4 Sulfur cycle

The original CANOPS ocean model (Ozaki and Tajika, 2013; Ozaki et al., 2011) treated two sulfur species, SO$_4^{2-}$ and ΣH$_2$S, in a closed system: Neither inputs to the ocean from rivers, hydrothermal vents, and submarine volcanoes, nor outputs due to evaporite formation and sedimentary pyrite burial were simulated. This simplification can be justified when the timescale of interest is less than the residence time of the S cycle (~10–20 Myr). The recently-revised CANOPS model (Ozaki et al., 2019a) extends the framework by incorporating the S budget in the ocean. In their model framework, the sedimentary S reservoirs are treated as boundary conditions: The size of sedimentary gypsum and pyrite reservoirs are prescribed and no explicit calculations of mass balance are performed. In CANOPS-GRB, we removed this model limitation and the sedimentary reservoirs are explicitly evaluated based on mass balance which is controlled by burial, outgassing and weathering (see Sect. 2.4). Specifically, seawater SO$_4^2$, ΣH$_2$S, and sedimentary sulfur reservoirs of pyrite sulfur (*PYR*) and gypsum sulfur (*GYP*) are explicitly evaluated in the current version of the model. No atmospheric sulfur species are calculated—all H$_2$S degassing from the ocean to the atmosphere is assumed to be oxidized to sulfate and return to the ocean. The organic sulfur cycle is not considered in this study.

Sulfur enters the ocean mainly from river runoff, $J_{\text{S}}{}^{\text{r}}$, with minor contributions from volcanic outgassing of sedimentary pyrite, $J_{\text{pyr}}{}^{\text{m}}$ and gypsum, $J_{\text{gyp}}{}^{\text{m}}$. The reference value for the riverine flux is set at 2.6 Tmol S yr$^{-1}$, consistent with the published estimate of 2.6±0.6 Tmol S yr$^{-1}$ (Raiswell and Canfield, 2012). The riverine flux is written as the sum of gypsum weathering and oxidative weathering of pyrite: $J_{\text{S}}{}^{\text{r}} = J_{\text{gyp}}{}^{\text{w}} + J_{\text{pyr}}{}^{\text{w}}$. Sulfur weathering fluxes are also assumed to be proportional to the sedimentary reservoir size. Estimates of modern volcanic input fall within the range of 0.3–3 Tmol S yr$^{-1}$ (Catling and Kasting, 2017; Kagoshima et al., 2015;





Raiswell and Canfield, 2012; Walker and Brimblecombe, 1985). We adopted a value of 0.8 Tmol S yr$^{-1}$ for
this flux (Kagoshima et al., 2015). Our total input of 3.4 Tmol S yr$^{-1}$ is also within the range of the previous
estimate of 3.3±0.7 Tmol S yr$^{-1}$ (Raiswell and Canfield, 2012). Sulfur is removed from the ocean either via
pyrite burial, $J_{pyr}{}^{b}$, or gypsum deposition, $J_{gyp}{}^{b}$ (Fig. 2d). The time evolution of the inventory of total S in the
ocean can thus be written, as follows:

$$\frac{d\left(M_{SO_4} + M_{H_2S}\right)}{dt} = \left(J_S^r + J_{pyr}^m + J_{gyp}^m\right) - \left(J_{pyr}^b + J_{gyp}^b\right),$$ (8)

where $M_{SO4}$ and $M_{H2S}$ denote the inventory of sulfate and hydrogen sulfide in the ocean, respectively. Two
sulfur species (SO$_4^{2-}$ and ΣH$_2$S) are transformed via microbial sulfate reduction (MSR) (R6), AOM (R11),
and aerobic sulfide oxidation reactions (R9). The above equation thus can be divided into following equations:

$$\frac{dM_{SO_4}}{dt} = J_S^r + J_{pyr}^m + J_{gyp}^m + J_{H_2S}^{ox} - J_{MSR\&AOM} - J_{gyp}^b,$$ (9)

$$\frac{dM_{H_2S}}{dt} = -J_{H_2S}^{ox} + J_{MSR\&AOM} - J_{pyr}^b,$$ (10)

where $J_{H2S}{}^{ox}$ denotes the oxidation of hydrogen sulfide and $J_{MSR\&AOM}$ is sulfate reduction via MSR and AOM.
Pyrite burial is represented as the sum of pyrite precipitation in the water column and sediments: $J_{pyr}{}^{b} = J_{pyr}{}^{b,wc}$
+ $J_{pyr}{}^{b,sed}$, where the pyrite burial rate in marine sediments is assumed to be proportional to the rate of benthic
sulfide production. The proportional coefficient, pyrite burial efficiency ($e_{pyr}$), is one of the tunable constants
of the model: For normal (oxic) marine sediments $e_{pyr}$ is tuned such that the seawater SO$_4^{2-}$ concentration for
our reference run is consistent with modern observations (Sect. 2.4.3). Pyrite precipitation in the water column
is assumed to be proportional to the concentration of ΣH$_2$S.

Although the present-day marine S budget is likely out of balance because of a lack of major gypsum formation,
the S cycle can be considered to operate at steady state on timescales longer than the residence time of sulfur
in the ocean. According to S isotope mass balance calculations, ~10–45% of the removal flux is accounted for
by pyrite burial, and the remainder is removed via formation of gypsum/anhydrite in the near-modern oceans
(Tostevin et al., 2014). Although gypsum deposition would have been strongly influenced by tectonic activity
(Halevy et al., 2012), we assume that the rate of gypsum deposition on geologic time scales is proportional to





the ion product of $Ca^{2+}$ and $SO_4^{2-}$ (Berner, 2004a) in the low- to mid-latitude surface layer (L), and is defined
as follows:

$$J_{gyp}^{b} = \left( \frac{[Ca^{2+}]_l [SO_4^{2-}]_l}{[Ca^{2+}]^* [SO_4^{2-}]^*} \right) J_{gyp}^{b,*} = f_{Ca} \left( \frac{[SO_4^{2-}]_l}{[SO_4^{2-}]^*} \right) J_{gyp}^{b,*} \tag{11}$$

where $l$ denotes the low- to mid-latitude surface layer and $f_{Ca}$ is a parameter that represents the seawater $Ca^{2+}$
concentration normalized by the present value ($f_{Ca} = 1$ for the reference run). The reference value of gypsum
burial $J_{gyp}^{b,*}$ is determined by assuming that gypsum deposition accounts for ~60% of the total S removal from
the near-modern ocean.
**2.3.5 Global redox budget**
In the previous version of the CANOPS model (Ozaki et al., 2019a), the atmospheric $O_2$ level was prescribed
as a boundary condition, rather than modeled in order to limit computational demands. In this study, we
remove this model limitation by introducing an explicit mass balance calculation of atmospheric $O_2$ (Sect.
2.6.3). This improvement allows us to explore the dynamic response of $O_2$ levels in the ocean-atmosphere
system (Sect. 4).

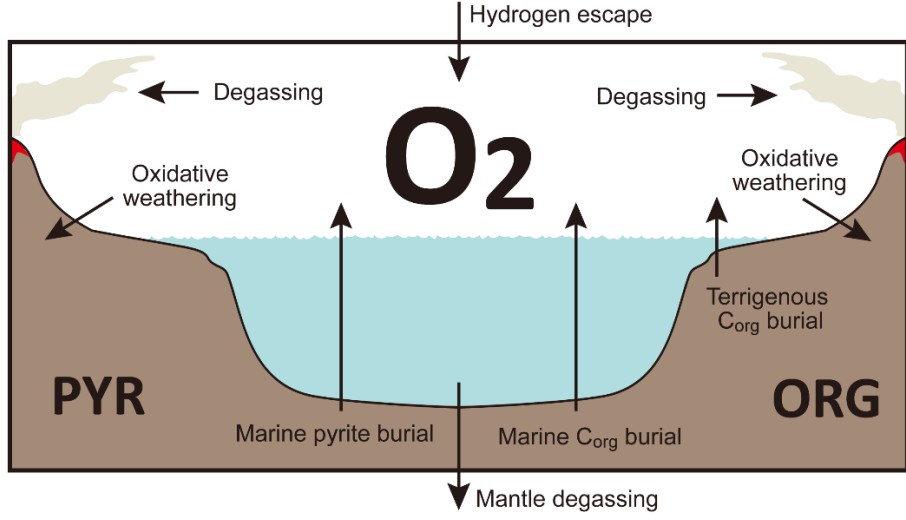


**Figure 3** Schematics of global redox ($O_2$) budget. Arrows represent the $O_2$ flux. The primary source is burial
of organic carbon and pyrite sulfur in sediments and hydrogen escape to space. The primary sink is volcanic
outgassing and weathering of crustal organic matter and pyrite. PYR = sedimentary reservoir of pyrite sulfur.
ORG = sedimentary reservoir of organic carbon. CANOPS-GRB tracks the global redox ($O_2$) budget for each
simulation.




The newly released CANOPS-GRB is designed to be a part of a comprehensive global redox budget (GRB)
framework (Fig. 3) (Catling and Kasting, 2017; Ozaki and Reinhard, 2021). Here GRB is defined for the
combined ocean-atmosphere system. In this study we track GRB in terms of $O_2$ equivalents. The ultimate
source of $O_2$ is the activity of oxygenic photosynthesis (and subsequent burial of reduced species, such as
organic matter and pyrite sulfur, in sediments), whereas the primary sink of $O_2$ is the oxidative weathering of
organic carbon and pyrite which are assumed to be $O_2$-dependent (Sect. 2.5.3). On timescales longer than the
residence time of $O_2$ in the ocean-atmosphere system, $O_2$ source fluxes should be balanced by sink fluxes.
Specifically, the $O_2$ budget in the coupled ocean-atmosphere system can be expressed as follows:

$$\text{GRB} = \left( J_{org}^{b,ocn} + J_{org}^{b,lnd} - J_{org}^{w} - J_{org}^{m} \right) + 2\left( J_{pyr}^{b} - J_{pyr}^{w} - J_{pyr}^{m} \right) + \left( J_{Hesc} - J_{man} \right), \tag{12}$$

where the first and second set of terms on the right-hand side represent the redox balance via organic carbon
and pyrite sulfur subcycles, respectively. $J_{Hesc}$ in the third term denotes hydrogen escape to space, representing
the irreversible oxidation of the system. For well-oxygenated atmospheres this process plays a minor role in
the redox budget, but for less oxygenated atmospheres with high levels of $CH_4$ this flux could lead to redox
imbalance. In this study we include the input of reducing power (e.g., $H_2$ and CO) from the Earth's interior to
the surface, $J_{man}$, which is assumed to be equal to the value of $J_{Hesc}$ ($J_{man} = J_{Hesc}$) to avoid redox imbalance in
the exogenic system. In reality, mantle degassing and the rate of hydrogen escape are not necessarily equal,
resulting in redox imbalance that may exert a fundamental control on atmospheric redox chemistry on geologic
timescales (Canfield, 2004; Eguchi et al., 2020; Hayes and Waldbauer, 2006; Ozaki and Reinhard, 2021),
however to maintain simplicity we have left this as a topic for future work. As a result, the terms on the right-
hand side must be balanced at steady state. Our model can meet this criterion. Note that the effects of the Fe
cycle on the $O_2$ budget (e.g., the oxidative weathering of Fe(II)-bearing minerals; Ozaki et al., 2019a) are not
currently included for the sake of simplicity.

The CANOPS-GRB model also tracks the $O_2$ budgets for the atmosphere and ocean independently, and these
can be evaluated by the following relationships:

$$ARB = \Phi_{ex}^{air\text{-}sea} + \left( J_{org}^{b,lnd} - J_{org}^{w} - J_{org}^{m} \right) - 2\left( J_{pyr}^{w} + J_{pyr}^{m} \right) + \left( J_{Hesc} - J_{man} \right), \tag{13}$$

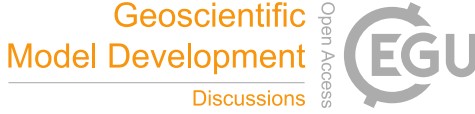

$$ORB = -\Phi_{ex}^{air\text{-}sea} + J_{org}^{b,ocn} + 2J_{pyr}^{b}, \tag{14}$$

where *ARB* and *ORB* denote the redox budget in the atmosphere and ocean, respectively, $\Phi_{ex}^{air\text{-}sea}$ represents
the net exchange of oxidizing power between the ocean and atmosphere via gas exchange ($O_2$ with minor
contributions of $NH_3$, $H_2S$ and $CH_4$). These separate redox budgets are also tracked in order to validate global
budget calculations.

For our reference condition, we obtain the reference value for the oxidative weathering rate of $C_{org}$ ($J_{org}^{w,*}$)
using the redox budget via $C_{org}$ subcycle:

$$J_{org}^{w,*} = J_{org}^{b,ocn,*} + J_{org}^{b,lnd,*} - J_{org}^{m,*}. \tag{15}$$

Given flux values based on the calculated ($J_{org}^{b,ocn,*} = 11.28$ Tmol C yr$^{-1}$) and prescribed ($J_{org}^{b,lnd,*} = 3$ Tmol
C yr$^{-1}$, $J_{org}^{m,*} = 1.25$ Tmol C yr$^{-1}$) values on the right-hand side, $J_{org}^{w,*}$ is estimated as 13.03 Tmol C yr$^{-1}$ (Table

541  2).

**2.4 Ocean model**
Here we undertake a thorough review, reconsideration and revision (where warranted), of all aspects of the
ocean model, including bringing together developments of the model since the original paper (Ozaki and
Tajika, 2013; Ozaki et al., 2011).

The ocean model includes exchange of chemical species with external systems via several processes such as
air-sea exchange, riverine input, and sediment burial. The biogeochemical model also includes a series of
biogeochemical processes, such as the ocean biological pump and redox reactions under oxic-anoxic-sulfidic
conditions. Our ocean model is convenient for investigating Earth system changes on timescales of hundreds
of years or longer and it can be relatively easily integrated, rendering the model unique in terms of
biogeochemical cycle models. CANOPS is also well suited for sensitivity studies and can be used to obtain
useful information upstream of more complex models.

Development of the ocean model included two initial goals: The first goal is to adopt a general and robust
ocean circulation scheme capable of producing well-resolved modern distributions of circulation tracers, using





realistic ventilation rates with a limited number of free parameters. The model's output for circulation tracers
are validated by comparison with modern observations (see Sect. 3). This confirms that our ocean circulation
scheme is adequate for representing the global patterns of water mass transport. The second goal is to couple
the circulation model with an ocean biogeochemical model, and to evaluate performance by comparison with
modern ocean biogeochemical data (see Sect. 3.2). Examination of the distributions and globally-integrated
fluxes of C, N, P, S, and $O_2$ for the modern ocean reveals that the ocean model can capture the fundamentals
of marine biogeochemical cycling.

### 2.4.1 Structure

CANOPS ocean model is a 1-D (vertically resolved) intermediate complexity box model of ocean
biogeochemistry (see Fig. 1b for the schematic structure) originally developed by (Ozaki and Tajika, 2013;
Ozaki et al., 2011). Our model structure is an improved version of the HILDA model (Joos et al., 1991; Shaffer
and Sarmiento, 1995). Unlike simple one-dimensional global ocean models (e.g, Southam et al., 1982), the
HILDA-type model includes explicit high-latitude dynamics whereby the high-latitude surface layer
exchanges properties with the deep ocean. This treatment is crucial for simulating preformed properties and
observed chemical distributions, especially for phosphate and dissolved $O_2$ in a self-consistent manner. Unlike
simple box-type global ocean models, the model has high vertical resolution. This is needed for representing
proper biogeochemical processes which show strong depth dependency. Furthermore, HILDA type models
(Arndt et al., 2011; Shaffer et al., 2008), unlike multi-box-type global ocean models (Hotinski et al., 2000),
use a small number of free parameters to represent ocean physics and biology. The simple and adaptable
structure of the model should make it applicable to a wide range of paleoceanographic problems. It couples a
diffusion-advection model of the global ocean with a biogeochemical model (Sect. 2.4.3) and a parameterized
sediment model (Sect. 2.4.4).

The ocean surface consists of a mixed layer at low-mid latitude (L) and high-latitude (H). Below the surface
layers, we adopt the present-day averaged seafloor topography of (Millero, 2006) (Hypsometric profile is
shown in Fig. 4a). Below the surface water layers, the ocean interior comprises two regions: the high-mid
latitude region (HD) and low-mid latitude region (LD). Each region is subdivided vertically, with high





resolution ($\Delta z$ = 100 m). Each of the 60 ocean layers in each latitude region (120 total) is assigned ocean
sediment properties. The cross-sectional area, volume, and sediment surface area of each box is calculated
from the benthic hypsometry. Inclusion of the bathymetry allows evaluation of the flux of biogenic materials
which settle on, and are buried in, seafloor sediments at each water depth (Sect. 2.3.3 and 2.3.4).

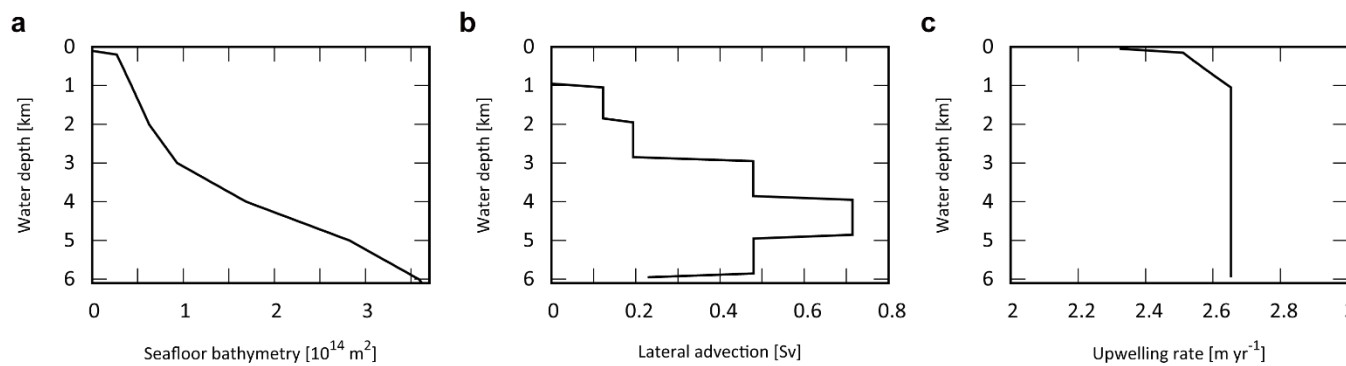


**Figure 4** Ocean bathymetry and water transport. (**a**) Seafloor topography (cumulative seafloor area) (Millero,
2006) adopted in the CANOS-GRB model. (**b**) Lateral water advection from HD to LD section assumed in
the standard run (in Sv). Total advection rate $\dot{V}$ was set at 20 Sv. (**c**) Upwelling rate in the LD region (in m
yr$^{-1}$) of the standard run.
**2.4.2 Transport**
The ocean circulation model represents a general and robust scheme that is capable of producing well-resolved
modern profiles of circulation tracers using realistic parameter values, and the coupled biogeochemical model
(Sect. 2.4.3) and the parameterized sediment model (Sect. 2.4.4).

The time-space evolution of model variables in the ocean is described by a system of horizontally integrated
vertical diffusion equations for non-conservative substances. The tracer conservation equation establishes the
relationship between change of tracer concentration at a given grid point and the processes that can change
that concentration. These processes include water transport by advection and mixing, and sources and sinks
due to biological and chemical transformations. The temporal and spatial evolution of the concentration of a
dissolved component in the aphotic zone is described by a horizontally integrated vertical diffusion equation,



which relates the rate of change of tracer concentration at a given point to the processes that act to change the
tracer concentration:

$$\frac{\partial [X]}{\partial t} = \frac{\partial [X]}{\partial t}\bigg|_{\text{trans}} + \Theta_{\text{bio}} + \Theta_{\text{react}},$$ (16)

where [X] represents horizontally integrated physical variables (such as potential temperature, salinity or $^{14}$C)
or concentration of a chemical component, $t$ denotes time, and $\Theta_{\text{bio}}$ and $\Theta_{\text{react}}$ represents internal sources and
sinks associated with the biological pump and chemical reactions, respectively. An external source/sink term
$\Theta_{\text{ex}}$, which represents riverine input and/or air-sea gas exchange, is added to the surface layers. The first term
on the right hand side of equation (16) represents the physical transport:

$$\frac{\partial [X]}{\partial t}\bigg|_{\text{trans}} = -A^{\text{l,h}}(z)w^{\text{l,h}}(z)\frac{\partial [X]}{\partial z} + \frac{\partial}{\partial z}\left(A^{\text{l,h}}(z)K_v^{\text{l,h}}(z)\frac{\partial [X]}{\partial z}\right) + K_{\text{hor}}\frac{\partial^2 [X]}{\partial y^2}.$$ (17)

The terms on the right-hand side express (from left to right) the advection, vertical diffusion, and horizontal
diffusion. Here, l and h indicate the LD and HD, respectively. The factors $K_v^{\text{l,h}}(z)$, $K_{\text{hor}}$, $A^{\text{l,h}}(z)$, and $w^{\text{l,h}}(z)$
denotes the vertical and horizontal diffusion coefficients, the areal fraction of the water layer at water depth $z$
to the sea surface area, and upwelling (for LD) or downwelling (for HD) velocity, respectively.

In the CANOPS ocean model, ocean circulation and mixing are characterized by five physical parameters: (1)
water transport via thermohaline circulation, $\dot{V}$, associated with high latitude sinking and low-mid latitude
upwelling; (2) constant horizontal diffusion between the aphotic zones, $K_{\text{hor}}$; (3) strong, depth-dependent
vertical diffusion between the aphotic zones in the high latitude region, $K_v^{\text{h}}(z)$; (4) high latitude convection,
$\dot{V}_{\text{h}}$; and (5) depth-dependent vertical diffusion in the low-mid latitude region, $K_v^{\text{l}}(z)$. These parameters are
tuned to give tracer distributions consistent with present-day observations.

**Advection**

Advective water transport in the ocean model represents the major features of modern meridional overturning
circulation. The rate of production of ventilated ocean waters ranges from 14 to 27 Sv (1 Sv = $10^6$ m$^3$ s$^{-1}$) in
the North Atlantic and from 18 to 30 Sv in the Southern Ocean (e.g., Doney et al., 2004; Lumpkin and Speer,





2007). The formation of deepwater effectively supplies "fresh" ventilated water to the abyss. We choose $\dot{V} =$
20 Sv as a reference value, giving a mean overturning time of about 2,140 yr, consistent with the ventilation
time estimated from observations (Broecker and Peng, 1982).

The downwelling of the surface waters at H forms HD that flows into the intermediate to deep oceanic layers
of LD, which, in turn, upwells over L (Fig. 1b). In many one-dimensional ocean models, downwelling water
enters the ocean interior via the deepest model layer (e.g., Shaffer and Sarmiento, 1995; Southam et al., 1982;
Volk and Hoffert, 1985). In the real ocean, downwelling waters are transported along isopycnal layers below
approximately 1,000 m (e.g., Doney et al., 2004; Lumpkin and Speer, 2007; Shaffer and Sarmiento, 1995;
Volk and Hoffert, 1985). Hence, we assume that high-latitude deep water flows into each ocean layer below
1,100 m. While there is some uncertainty in the pattern of lateral advection, the flow is determined in our
model assuming a constant upwelling rate below a depth of 1,100 m in the LD region. The
upwelling/downwelling rate $w^{l,h}(z)$ is then determined by the seafloor topography and the deep water lateral
inflow, assuming continuity. Figure 4b shows the lateral advection of deep waters with a reference circulation
rate $\dot{V}$ of 20 Sv. This assumption provides a plausible upwelling rate, which is consistent with the oft-quoted
value of 2–3 m yr$^{-1}$ (Broecker and Peng, 1982) (Fig. 4c).
**Vertical mixing**
Ocean circulation is dominated by turbulent processes driven by wind and tidal mixing. These processes occur
as eddies which occur at a wide range of spatial scales, from centimeters to whole ocean basins. In numerical
models of ocean circulation, turbulent mixing in the ocean interior is commonly represented as a diffusion
process, characterized by an eddy diffusion coefficient. The vertical eddy diffusion coefficient $K_v(z)$ is
typically on the order of $10^{-5}$ to $10^{-4}$ m$^2$ s$^{-1}$ and it is common to assume a depth-dependence which smoothly
increases from the thermocline ($\sim 10^{-5}$ m$^2$ s$^{-1}$) to the abyss ($\sim 10^{-4}$ m$^2$ s$^{-1}$) using an inverse or hyperbolic tangent
function (e.g., Shaffer et al., 2008; Yakushev et al., 2007). To account for thermocline ventilation, we assumed
a relatively high vertical diffusion coefficient in mid-water depth ($K_l = 6.3 \times 10^{-5}$ m$^2$ s$^{-1}$ for water depth
500–1500 m). We also adopted a higher value for the vertical diffusion coefficient ($K_u = 1.6 \times 10^{-4}$ m$^2$ s$^{-1}$) in





the upper most 500 m of the ocean in order to represent the highly convective Ekman layer in the upper part
of the ocean.

$$
K_v^1(z) = \begin{cases} K_u & (z \geq \text{-500 m}) \\[2mm] K_l & (\text{-500} \geq z \geq \text{-1500 m}) \\[2mm] \kappa^s + \dfrac{\kappa^d - \kappa^s}{2}\left(1 + \tanh\left(\dfrac{z - z^1}{z^1}\right)\right) & (\text{otherwise}) \end{cases} \tag{18}
$$

where $\kappa^s$ and $\kappa^d$ are vertical mixing coefficients, and $z^1$ is the transition length scale (Romaniello and Derry,
2010). In the high latitude region where no permanent thermocline exists, more rapid communication with
deep waters can occur. Previous studies have pointed out that the vertical diffusivities at high latitude can be
very high (up to $O(10^{-2}\ \mathrm{m^2\ s^{-1}})$) (e.g., Sloyan, 2005). To account for this we include high-latitude convection
between H and YD ($\dot{V}_h = 57.4$ Sv) and higher vertical diffusion ($K_v^h(z) = 2\times K_v^1(z)$).

**Horizontal diffusion**

The horizontal diffusivity is included according to Romaniello and Derry (2010). On basin scales, the
horizontal (isopycnal) eddy diffusivity is $10^7$–$10^8$ times larger than the vertical (diapycnal) eddy diffusivity
due to anisotropy of the density field. For a spatial scale of 1,000 km, horizontal eddy diffusion is estimated
to be $O(10^3\ \mathrm{m^2\ s^{-1}})$ (e.g., Ledwell et al., 1998). We adopt this value. As Romaniello and Derry (2010) did, we
assume horizontal mixing follows the pathways of advective fluxes between laterally adjacent regions. The
reciprocal exchange fluxes may be written as

$$
J_{\text{hor}}^{\text{ex}} = K_{\text{hor}} A_{\perp} \frac{\partial [X]}{\partial y} = \frac{K_{\text{hor}} A_{\perp}}{L} \Delta [X], \tag{19}
$$

where $J_{\text{hor}}^{\text{ex}}$ denotes the exchange fluxes between the layers (in mol yr$^{-1}$), $A_{\perp}$ represents the cross-sectional
area separating two adjacent reservoirs, $L$ is a characteristic spatial distance separating the reservoirs, $\Delta[X]$ is
the difference in concentration between two reservoirs (Romaniello and Derry, 2010). By assuming that $L$ is



of the same order as the length of the interface separating the two regions, we can approximate $A_\perp \approx \Delta z \times O(L)$,
where $\Delta z$ is the thickness of the interface separating the two regions. Then we obtain

$$J_{\mathrm{hor}}^{\mathrm{ex}} = K_{\mathrm{hor}} \Delta z \Delta[X]. \qquad (20)$$

Therefore, when we discretize the ocean interior at 100 m spacing approximately 0.1 Sv of reciprocal mixing
occurs between adjacent layers.

## Ocean circulation tracers

We use potential temperature $\theta$, salinity $S$, and radioactive carbon $^{14}$C, as physical tracers. Distributions of
these tracers are determined by the transport mechanisms described above. In this study, we adopt the values
at the surface layers (L and H) as upper boundary conditions: $\theta^l = 15°C$, $\theta^h = 0°C$, $S^l = 35$ psu, $S^h = 34$ psu,
$\Delta^{14}C^l = -40‰$, and $\Delta^{14}C^h = -100‰$. The radioactive decay rate for $^{14}$C is $1.21 \times 10^{-4}$ yr$^{-1}$. Although $^{14}$C can be
incorporated in the biogenic materials and transported into deep water, we ignore this biological effect for
simplicity. The associated error is ~10% of the profiles produced by circulation and radioactive decay (Shaffer
and Sarmiento, 1995). The parameter values used in the ocean circulation model are listed in Table 3.
Table 3: Physical set-up of the ocean circulation model.

| Parameters | Label | Value | Unit | Ref. |
|---|---|---|---|---|
| Ocean surface area | $A$ | $3.62 \times 10^{14}$ | m$^2$ | (Ozaki and Tajika, 2013) |
| Surface area of high-latitude layer (H) | $A^h$ | $0.362 \times 10^{14}$ | m$^2$ | (Ozaki and Tajika, 2013) |
| Depth of mixed layer | $h_m$ | 100 | m | (Ozaki and Tajika, 2013) |
| Grid spacing | $\Delta z$ | 100 | m | (Ozaki and Tajika, 2013) |
| Water depth of ocean bottom | $z_b$ | 6,100 | m | (Ozaki and Tajika, 2013) |
| Ocean overturning rate | $\dot{V}$ | 20 | Sv | (Ozaki and Tajika, 2013) |
| Vertical mixing coefficient ($z < 500$ m) | $K_u$ | 5,000 | m$^2$ yr$^{-1}$ | (Ozaki and Tajika, 2013) |
| Vertical mixing coefficient (500 m $< z <$ 1,500 m) | $K_l$ | 2,500 | m$^2$ yr$^{-1}$ | (Ozaki and Tajika, 2013) |
| Mixing coefficient | $\kappa_s$ | 473 | m$^2$ yr$^{-1}$ | (Romaniello and Derry, 2010) |
| Mixing coefficient | $\kappa_d$ | 3,154 | m$^2$ yr$^{-1}$ | (Romaniello and Derry, 2010) |
| Transition depth for vertical mixing coefficient | $z_l$ | 1,000 | m | (Romaniello and Derry, 2010) |
| High-latitude convection rate | $\dot{V}_h$ | 57.4 | Sv | (Ozaki and Tajika, 2013) |
| Horizontal diffusion coefficient | $K_{\mathrm{hor}}$ | 1,000 | m$^2$ s$^{-1}$ | (Romaniello and Derry, 2010) |





### 2.4.3 Ocean biogeochemical framework

The ocean circulation model is coupled to a biogeochemical model, which includes an explicit representation of a variety of biogeochemical processes in the ocean. The parameters used in the oceanic biogeochemical model are listed in Table 4.

Table 4 Parameter values used in the oceanic biogeochemistry module of CANOPS-GRB.

| Parameter | Label | Value | Unit | Ref. |
|---|---|---|---|---|
| Efficiency factor for phosphate uptake at L | $\varepsilon^l$ | 1.0 | – | (Ozaki and Tajika, 2013) |
| Efficiency factor for phosphate uptake at H | $\varepsilon^h$ | 0.15 | – | (Ozaki and Tajika, 2013) |
| Phosphate half saturation constant | $K_P$ | $1\times10^{-6}$ | mM | (Ozaki and Tajika, 2013) |
| Export ratio | $f_{exp}$ | 0.2 | – | (Ozaki et al., 2019a) |
| Redfield C/P ratio | $\alpha^*$ | 106 | mol mol$^{-1}$ | (Redfield et al., 1963) |
| Redfield N/P ratio | $\beta^*$ | 16 | mol mol$^{-1}$ | (Redfield et al., 1963) |
| POM sinking velocity | $v_{POM}$ | 100 | m d$^{-1}$ | (Ozaki et al., 2011) |
| Mass fraction of $G_1$ | $m_1$ | 0.72 | – | (Ozaki and Tajika, 2013) |
| Mass fraction of $G_2$ | $m_2$ | 0.25 | – | (Ozaki and Tajika, 2013) |
| Mass fraction of $G_3$ | $m_3$ | 0.03 | – | (Ozaki and Tajika, 2013) |
| Decomposition rate of $G_1$ | $k_1$ | 0.6 | d$^{-1}$ | (Ozaki et al., 2011) |
| Decomposition rate of $G_2$ | $k_2$ | 0.1 | d$^{-1}$ | (Ozaki et al., 2011) |
| Decomposition rate of $G_3$ | $k_3$ | 0.0 | d$^{-1}$ | (Ozaki et al., 2011) |
| Half saturation constant for aerobic respiration | $K_{O2}$ | $8\times10^{-3}$ | mM | (Boudreau, 1996) |
| Half saturation constant for denitrification | $K_{NO3}$ | $3\times10^{-2}$ | mM | (Boudreau, 1996) |
| Half saturation constant for MSR | $K_{MSR}$ | 0.2 | mM | This study |
| Half saturation constant for AOM | $K_{AOM}$ | 0.093 | mM | (Beal et al., 2011) |
| Aerobic oxidation rate of ammonium | $k_{R8}$ | $1.825\times10^4$ | mM$^{-1}$ yr$^{-1}$ | (Oguz et al., 2001) |
| Aerobic oxidation rate of sulfide | $k_{R9}$ | $3.65\times10^3$ | mM$^{-1}$ yr$^{-1}$ | (Oguz et al., 2001) |
| Aerobic methane oxidation rate | $k_{R10}$ | $1\times10^7$ | mM$^{-1}$ yr$^{-1}$ | (Van Cappellen and Wang, 1996) |
| Anaerobic methane oxidation rate | $k_{R11}$ | $3\times10^{-4}$ | yr$^{-1}$ | (Ozaki et al., 2019a) |
| Pyrite formation rate in the water column | $k_{pyr}^{wc}$ | 0.01 | yr$^{-1}$ | This study |
| Reference value of seawater sulfate concentration | $[SO_4^{2-}]^*$ | 28.9 | mM | (Ozaki et al., 2019a) |





**Biological production**

The overall biogeochemical cycling scheme is based on the cycling of primary nutrient (phosphate; $PO_4^{3-}$), which limits biological productivity—export production is related to the availability of P within the euphotic zone (Maier-Reimer, 1993; Shaffer et al., 2008; Yamanaka and Tajika, 1996):

$$j_{exp}^{l,h} = \alpha^{l,h} h_m \varepsilon^{l,h} [PO_4^{3-}]^{l,h} \frac{[PO_4^{3-}]^{l,h}}{[PO_4^{3-}]^{l,h} + K_P} \,, \tag{21}$$

where $j_{exp}$ represents new/export production of POC (in unit of mol C m$^{-2}$ yr$^{-1}$), $\alpha$ denotes C:P stoichiometry of POM, $h_m$ is the mixed layer depth, $\varepsilon$ denotes the assimilation efficiency factor for P uptake, and $K_P$ denotes the half-saturation constant. The value of $\varepsilon$ for the low-mid latitude region is assumed to be 1. In contrast, we assume a lower efficiency for high latitude region because biological production tends to be limited by environmental factors other than phosphate availability (e.g., amount of solar radiation, mixed layer thickness, sea-ice formation, and iron availability). This is used as one of the fitting parameters in the model. Downwelling waters contain a certain level of nutrients (i.e., preformed nutrients).

In our standard run, the stoichiometry of organic matter is parameterized using the canonical Redfield ratio (C:N:P = 106:16:1) (Redfield et al., 1963). However, we note that flexible C:N:P stoichiometry has been the subject of recent discussion. In the modern oceans, C:N:P ratios of exported POM vary across latitude, reflecting ecosystem structure (Galbraith and Martiny, 2015). Local observations (and laboratory experiments) suggest that the C:N:P ratio of cyanobacteria is a function of seawater $PO_4^{3-}$ concentration (Larsson et al., 2001). The evolutionary perspective has also been discussed (Quigg et al., 2003; Sharoni and Halevy, 2022). In the previous version of the CANOPS model, the C-N-P stoichiometry of primary producers responds dynamically to P availability in the surface layer (Reinhard et al., 2017b):

$$\alpha = \alpha^* + \frac{\alpha_{max} - \alpha^*}{2}\left(1 + \tanh\left(\frac{\gamma_{P0} - [PO_4^{3-}]}{\gamma_{P1}}\right)\right), \tag{22}$$

$$\beta = \beta^* + \frac{\beta_{max} - \beta^*}{2}\left(1 + \tanh\left(\frac{\gamma_{P0} - [PO_4^{3-}]}{\gamma_{P1}}\right)\right), \tag{23}$$





where $\alpha$ and $\beta$ represent the C/P ratio and N/P ratio of POM, * denotes the canonical Redfield ratios, max
denotes the maximum value ($\alpha_{max} = 400$ and $\beta_{max} = 60$), and $\gamma_{P0}$ and $\gamma_{P1}$ are tunable constants ($\gamma_{P0} = 0.1$ $\mu$M
and $\gamma_{P1} = 0.03$ $\mu$M) (Kuznetsov et al., 2008). In the CANOPS-GRB model, this dynamic response of POM
stoichiometry can be explored by changing the user-flag from the standard static response. In this study, we
do not explore the impacts of flexible POM stoichiometry on global biogeochemistry (i.e., $\alpha_{max} = \alpha^*$ and $\beta_{max}$
$= \beta^*$).

Biological production in the surface mixed layer increases the concentration of dissolved $O_2$ and reduces the
concentrations of DIP and DIN according to the stoichiometric ratio (R1 and R2; Table 1). DIN consumption
is partitioned between nitrate and ammonium, assuming that ammonium is preferentially assimilated.
CANOPS-GRB evaluates the availability of fixed N in the surface ocean, and any N deficiency required for a
given level of productivity is assumed to be compensated for on geologic time scales by N fixers. In other
words, it is assumed that biological N fixation keeps pace with P availability, so that P (not N) ultimately
determines oceanic biological productivity.

To date, models of varying orders of complexity have been developed to simulate oceanic primary production
and nutrient cycling in the euphotic layer, from a single nutrient and single phytoplankton component system
to the inclusion of multiple nutrients and trophic levels in the marine ecosystem, usually coupled to physical
models (e.g., Oguz et al., 2000; Yakushev et al., 2007). To avoid this level of complexity, we introduce a
parameter, $f_{exp}$, called export ratio (Sarmiento and Gruber, 2006), which relates the flux densities of export
production and NPP, as follows:

$$j_{NPP}^{ocn} = \frac{j_{exp}}{f_{exp}},\tag{24}$$

where $j_{NPP}^{ocn}$ denotes the NPP in terms of mol C m$^{-2}$ yr$^{-1}$. In the modern ocean globally averaged value of $f_{exp}$
is estimated at 0.2 (Laws et al., 2000), and we assumed this value in this study. The rate of recycling of organic
matter in the photic zone is thus given by





$$j_{\text{recy}} = j_{\text{NPP}}^{\text{ocn}} - j_{\text{exp}} = \frac{1 - f_{\text{exp}}}{f_{\text{exp}}} j_{\text{exp}}. \tag{25}$$

The respiration pathway of $j_{\text{recy}}$ depends on the availability of terminal electron acceptors ($O_2$, $NO_3^-$ and $SO_4^{2-}$).
Following exhaustion of these species as terminal electron acceptors, organic matter remineralization occurs
by methanogenesis (R7). See below for the treatment of organic matter remineralization in the water column.

**Biological pump**

Most POM exported to the deep sea is remineralized in the water column before reaching the seafloor (e.g.,
Broecker and Peng, 1982). Nutrients returning to seawater at intermediate depths may rapidly return to the
surface ocean and support productivity. The remaining fraction of POM that reaches the sediment ultimately
exerts an important control on oceanic inventories of nutrients and $O_2$. An adequate representation of the
strength of biological pump is therefore critical to any descriptions of global biogeochemical cycles.

The governing equation of the concentration of biogenic particles $G$ is

$$\frac{\partial G}{\partial t} + v_{\text{POM}} \frac{\partial G}{\partial z} = -rG \tag{26}$$

where $r$ is a decomposition rate and $v_{\text{POM}}$ is the settling velocity of POM in the water column. We assume a
settling velocity of 100 m d$^{-1}$ for our reference value (e.g., Suess, 1980), although a very wide range of values
and depth-dependency have been reported (e.g., Berelson, 2001b). Therefore, the settling velocity is fast
enough to neglect advective and diffusive transport of biogenic particles. Note that the settling velocity would
affect the intensity of biological pump and chemical distribution in the ocean interior. Considering the ballast
hypothesis in the modern ocean (Armstrong et al., 2001; Francois et al., 2002; Ittekkot, 1993; Klaas and Archer,
2002), the settling velocity of POM in the geological past would have been different from the modern ocean.
As Kashiyama et al. (2011) pointed out, there would be a critical aspect among sinking rate of POM, intensity
of biological pump and chemical distribution in the ocean. The quantitative and comprehensive evaluation of
their effect is an important issue for the future work (Fakhraee et al., 2020).





In order to solve equation (26) explicitly, a relatively small time step (~1 day) would be required. However,
because the sinking velocity and remineralization of biogenic material are fast processes, we assume that the
POM export and remineralization occurs in the same time step (ignoring the term $\partial G/\partial t$). Then the
concentration of biogenic particles can be solved as follows:

$$G(z + \Delta z) = G(z) \exp\left(-\frac{r\Delta z}{v_{\text{POM}}}\right). \tag{27}$$

where $\Delta z$ is a spatial resolution of the model.

**Organic matter decomposition**

As POM settles through the water column, it is nearly entirely decomposed back to dissolved tracers.
Therefore, decomposition of POM is a key process for modelling biogeochemistry in the ocean. To avoid the
complex treatment of this process (such as repackaging and aggregation/dispersal of particles), various
empirical schemes for POM sinking flux have been proposed, such as exponential (Volk and Hoffert, 1985)
or power law (Martin et al., 1987) functions (Fig. 5). However, the estimation of Volk and Hoffert generally
tends to overestimate in the upper water column (<1.5 km) and underestimate at depth. It is important to note
that data series of sediment trap measurements were obtained from a limited geographic and depth range.
Berelson (2001a) and Lutz et al. (2002) conducted further estimates of the sediment flux and found regional
variability in the sinking flux. Broadly, these data indicate that commonly applied flux relationships generally
tend to overestimate flux to depth.



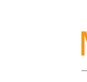

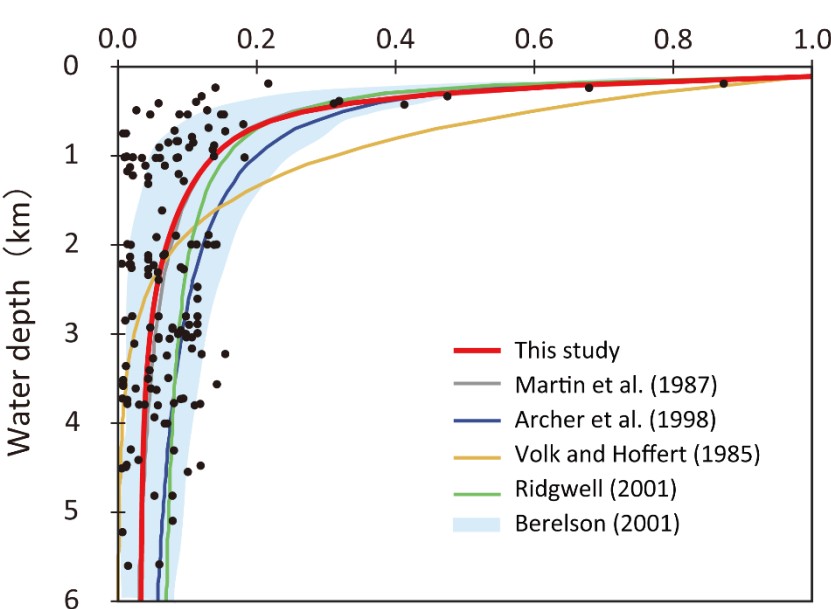


**Figure 5** Empirical relationships between POC settling flux normalized to export production (Lutz et al., 2002) and water depth (Archer et al., 1998; Berelson, 2001a; Martin et al., 1987; Volk and Hoffert, 1985). The profile of the CANOPS-GRB model is depicted as a red line. The black dots represent observational data (Honjo, 1980; Honjo and Manganini, 1993; Lutz et al., 2002; Tsunogai and Noriki, 1991, and references therein).

The microbial degradation of different groups of organic matter with different labilities differs over timescales ranging from hours to millions of years. In order to represent the decrease in POM lability with time and water depth, we adopt the so-called multi-G model (Westrich and Berner, 1984) that describes the detailed kinetics of organic matter decomposition (Ozaki and Tajika, 2013; Ozaki et al., 2011). In the CANOPS model, POM is described using two degradable fractions ($G_1$ and $G_2$) and one inert ($G_3$) fraction using different rate constants $k_i$ (i=1, 2, 3) for each component. Rate constants are tuned on the basis of consistency with the typical profile of the POM sinking flux estimated from sediment trap studies (Fig. 5). In this study, constant stoichiometries between C, N, and P during the remineralization of POM are assumed throughout the water column, taking values equal to those characterizing mean export production.



The electron acceptor used in the respiration reaction changes from dissolved $O_2$ to other oxidants (e.g., $NO_3^-$
and $SO_4^{2-}$) as $O_2$ becomes depleted. The respiration pathway is controlled by the free energy change per mole
of organic carbon oxidized. The organic matter decomposition is performed by the oxidant which yields the
greatest free energy change per mole of organic carbon oxidized. When the oxidant is depleted, further
decomposition will proceed utilizing the next most efficient (i.e., the most energy producing) oxidant until
either all oxidants are consumed or oxidizable organic matter is depleted (e.g., Berner, 1989; Froelich et al.,
1979). In oxic waters, organic matter is remineralized by an aerobic oxidation process (R4). As dissolved $O_2$
is depleted, $NO_3^-$ and/or $SO_4^{2-}$ will be used (R5 and R6). Denitrification is carried out by heterotrophic bacteria
under low concentrations of dissolved $O_2$, if there is sufficient nitrate. For anoxic, sulfate-lean oceans, organic
matter production from carbon dioxide (methanogenesis) or fermentation of organic matter to produce
methane and carbon dioxide will occur (R7). In the CANOPS-GRB model, we parameterized the dependence
of decomposition of POM with a Michaelis-Menten type relationship with respect to the terminal electron
acceptors:

$$R_4 = \frac{[O_2]}{K_{O_2} + [O_2]} (\sum k_i G_i) \tag{28}$$

$$R_5 = \frac{K'_{O_2}}{K'_{O_2} + [O_2]} \frac{[NO_3^-]}{K_{NO_3} + [NO_3^-]} (\sum k_i G_i) \tag{29}$$

$$R_6 = \frac{K'_{O_2}}{K'_{O_2} + [O_2]} \frac{K'_{NO_3}}{K'_{NO_3} + [NO_3^-]} \frac{[SO_4^{2-}]}{K_{MSR} + [SO_4^{2-}]} (\sum k_i G_i) \tag{30}$$

$$R_7 = \frac{K'_{O_2}}{K'_{O_2} + [O_2]} \frac{K'_{NO_3}}{K'_{NO_3} + [NO_3^-]} \frac{K'_{MSR}}{K'_{MSR} + [SO_4^{2-}]} (\sum k_i G_i)$$
$$= \left(1 - R_{O_2} - R_{NO_3} - R_{SO_4}\right)(\sum k_i G_i) \tag{31}$$

where $K_{O2}$, $K_{NO3}$, and $K_{MSR}$ are Monod constants, and $K'_{O2}$, $K'_{NO3}$, $K'_{MSR}$ are inhibition constants. The Monod-
type expressions are most widely used in mathematical models of POM decomposition processes (e.g.,
Boudreau, 1996). The oxidants for organic matter decomposition change gradually, depending on the amount
of each oxidant. The parameter values are based on the previous studies on early diagenetic processes in
marine sediments (Boudreau, 1996; Van Cappellen and Wang, 1996). $SO_4^{2-}$ has been one of the major





components of the Phanerozoic oceans and has been an important oxidizing agent in anaerobic systems. In
the original CANOPS model (Ozaki and Tajika, 2013; Ozaki et al., 2011), it was assumed that the saturation
constant $K_{MSR}$ is zero, meaning that the $SO_4^{2-}$ is never a limiting factor. In contrast, during the Precambrian,
seawater $SO_4^{2-}$ could have been extremely low (Lyons and Gill, 2010). The half saturation constant for MSR
($K_{MSR}$) determines the degree to which MSR contributes to the total respiration rates. However, estimates for
$K_{MSR}$ in natural environments and pure cultures vary over several orders of magnitude (~0.002–3 mM) (Pallud
and Van Cappellen, 2006; Tarpgaard et al., 2011). We assume a reference value of 0.2 mM for this study.

Finally, temperature may also have played an important role in organic matter decomposition rates. The
dependence of ammonification on temperature is sometimes described by an exponential function or $Q_{10}$
function (e.g., Yakushev et al., 2007). While we recognize that the temperature dependency of organic matter
decomposition might have played an important role in oceanic biogeochemical cycles in geological past
(Crichton et al., 2021), these dynamics are not included in our standard model.
**Secondary redox reactions**
Total ammonia ($\Sigma NH_3$), total sulfide ($\Sigma H_2S$), and methane ($CH_4$), produced during organic matter degradation,
are subject to oxidation by $NO_3^-$, $SO_4^{2-}$, and $CO_2$ via a set of secondary redox reactions (Table 1). Rate
constants for these reactions are taken from the literature. The ocean model includes nitrification (R8), total
sulfide oxidation by $O_2$ (R9), aerobic oxidation of $CH_4$ by $O_2$ (R10), and AOM by $SO_4^{2-}$ (R11). Nitrification,
the oxidation of ammonium to nitrate, occurs in several stages and is accomplished mainly by
chemolithotrophic bacteria (Sarmiento and Gruber, 2006). In this study, we treat all nitrification reactions as
a combined reaction (R8). The rate of this process is assumed to depend on the concentration of both oxygen
and ammonia as follows:

$$R_8 = k_{R8}[NH_4^+][O_2]. \tag{32}$$


The oxidation of sulfide formed in anoxic waters by MSR can also be written as a series of reactions (e.g.,
Yakushev and Neretin, 1997), but we treat it as an overall reaction (R9). The rate of this secondary redox
reaction is also formulated using bimolecular rate law:





$$R_9 = k_{R9}[\Sigma H_2 S][O_2]. \tag{33}$$

The rate constant for this process has been shown to vary significantly as a function of several redox-sensitive
trace metals which act as catalysts (Millero, 1991). Here we assume $k_{R9} = 3650$ mM$^{-1}$ yr$^{-1}$ based on the
observations of suboxic layer of the Black Sea (Oguz et al., 2001).

In the original CANOPS model (Ozaki et al., 2019a; Ozaki and Tajika, 2013), syngenetic pyrite formation in
the water column was not considered. In a more recent revision of the model, this process was added (Cole et
al., 2022) and parameterized such that iron sulfide formation is assumed to be proportional to the hydrogen
sulfide concentration:

$$R_{pyr}^{wc} = k_{pyr}^{wc}[\Sigma H_2 S], \tag{34}$$

where $k_{pyr}^{wc}$ is a model constant (its reference value is set at 0.01 yr$^{-1}$). This constant is a function of the ferrous
iron concentration in seawater, but it is the subject of large uncertainty. The total flux (in mol S yr$^{-1}$) can be
obtained by integrating the precipitation flux density over the whole ocean:

$$J_{pyr}^{wc} = \int R_{pyr}^{wc} \frac{dV}{dz} dz. \tag{35}$$


CH$_4$ can be oxidized not only by O$_2$ in aerobic waters (R10) but by SO$_4$ in anoxic waters (R11). The aerobic
oxidation of CH$_4$ is formulated using a bimolecular rate law:

$$R_{10} = k_{R10}[CH_4][O_2]. \tag{36}$$

The rates of AOM is formulated using a Monod-type law (Beal et al., 2011):

$$R_{11} = k_{R11}[CH_4] \frac{[SO_4^{2-}]}{K_{AOM} + [SO_4^{2-}]}. \tag{37}$$

Rate constants for above reactions are taken from the literature (Table 4). Secondary redox reactions were
calculated implicitly with an operator splitting scheme (Steefel and MacQuarrie, 1996) so as to maintain
numerical stability.





### 2.4.4 Sediment-water exchange

The burial of biogenic material in marine sediments plays a critical role in global biogeochemical cycles, especially with respect to the marine budgets of nutrients, carbon, and sulfur. This is intimately linked to atmospheric $O_2$ levels on geologic timescales. Specifically, the burial rate of $C_{org}$ in marine sediments exerts a primary control on the evolution of atmospheric $O_2$ levels throughout Earth's history. Given the complexity of biogeochemical processes within sediments and our limited knowledge on many of the early diagenetic processes, we adopt some semi-empirical relationships extracted from ocean sediment data. This approach, rather than explicit modelling, is also required to reduce the computational cost of the simulation on timescales >100 Myr. The related parameter values are in Table 5.

Table 5: Parameters used in the sediment-water interface module of CANOPS-GRB.

| Parameters | Label | Value | Unit | Ref. |
|---|---|---|---|---|
| Scale constant for benthic aerobic respiration | $k$ | 0.02 | $yr^{-1}$ | (Ozaki et al., 2019a) |
| Burial efficiency of $C_{org}$ at zero sediment accumulation rate | $be_1$ | 5.0 | % | (Dale et al., 2012) |
| Burial efficiency of $C_{org}$ at infinite sediment accumulation rate | $be_2$ | 75 | % | (Dale et al., 2012) |
| Centre of the regression for burial efficiency of organic C | $a$ | 0.01 | $g\ cm^{-2}\ yr^{-1}$ | (Dale et al., 2012) |
| Pyrite burial efficiency in oxic sediments | $e_{pyr}^{oxic}$ | 0.117 | – | This study |
| Pyrite burial efficiency in anoxic sediments | $e_{pyr}^{anox}$ | 1 | – | (Ozaki et al., 2019a) |
| $O_2$ threshold value for P burial efficiency | $oxic$ | 0.25 | mM | (Ozaki et al., 2011) |
| $C_{org}/N_{org}$ ratio of buried sediments | $(C_{org}/N_{org})_b$ | 10 | $mol\ mol^{-1}$ | (Ozaki and Tajika, 2013) |

### POM deposition

The fraction of settling POM that reaches the sediment surface, $J_{org}^{dep}$ (in mol C $yr^{-1}$) is a function of both the settling flux density, $j_{org}^{dep}$ (in mol C $m^{-2}$ $yr^{-1}$), and topography (Fig. 4a):

$$J_{org}^{dep} = \int_{z_1}^{z_2} j_{org}^{dep}(z) \frac{dA}{dz} dz,$$ (38)

where the settling flux density can be written as follows:

$$j_{org}^{dep} = v_{POM} G.$$ (39)

where $G$ is the concentration of POM, and $v_{POM}$ denotes the sinking velocity.





**Carbon cycling**

Interactions between the ocean and underlying sediments play an important role in influencing whole-ocean chemical and nutrient inventories on geologic time scales. POM deposited to the seafloor is subject to decomposition during diagenetic processes associated with burial in marine sediments. Only a small fraction of organic matter will ultimately be buried and removed from the surface environment. However, understanding what factors control the preservation of organic matter in marine sediments has been a controversial topic, and we still lack a robust understanding of this process. With this issue in mind, we adopt an empirical approach obtained using the observational data from previous studies.

The burial flux density of $C_{org}$ at each water depth, $j_{org}^{b,ocn}$ (in terms of mol C m$^{-2}$ yr$^{-1}$), is calculated based on burial efficiency, $BE_{org}$:

$$j_{org}^{b,ocn} = BE_{org}\, j_{org}^{dep}\,. \tag{40}$$

Burial efficiency is defined as the fraction of organic matter buried in sediments relative to the total depositional flux. Burial efficiency is described by simplified parametric laws based on empirical relationships from modern day observations. Previous studies demonstrate strong dependency of this term on total sedimentation rate, $SR$ (e.g., Henrichs and Reeburgh, 1987). Figure 6 demonstrates the relationship between $BE_{org}$ and $SR$ compiled from literature surveys. The sedimentation rate in the modern ocean varies over about five orders of magnitude, with a primary dependence on material supplied from the continents. There is a strong relationship, especially for $SR$ less than 0.01 cm yr$^{-1}$. In contrast to the strong $SR$ dependence under oxic conditions, anoxic settings show a much weaker dependence of $BE_{org}$ on $SR$ (Betts and Holland, 1991; Henrichs and Reeburgh, 1987) (Fig. 6). In this study, the following relationship proposed by (Henrichs and Reeburgh, 1987) is adopted for sediments underlying well-oxygenated bottom water ($O_2$ concentration of bottom water, $[O_2]_{bw} > 200\ \mu$M):

$$BE_{org} = \frac{SR^{0.4}}{2.1}\,. \tag{41}$$

Given that $BE_{org}$ depends on the $[O_2]_{bw}$ (Katsev and Crowe, 2015; Lasaga and Ohmoto, 2002), we adopt the following formulation for sediments underlying less-oxygenated bottom waters ($[O_2]_{bw} < 30\ \mu$M) (Dale et al., 2012):





$$BE_{org} = b_2 + \frac{b_1 - b_2}{1 + SR/a}, \tag{42}$$

where $a = 0.019$, $b_1 = 0.05$, and $b_2 = 0.7$, respectively. For intermediate $[O_2]_{bw}$ levels, $BE_{org}$ is evaluated as a
function of $[O_2]_{bw}$ with a log-linear interpolation method. Note that the original CANOPS model (Ozaki and
Tajika, 2013; Ozaki et al., 2011) adopted Eq. (41) without considering the $O_2$ dependency, whereas more
recent versions employ Eq. (42) for both oxic and anoxic sediments with different values of $a$, $b_1$ and $b_2$. In
the CANOPS-GRB, we adopted both equations, because of the sake of more accurate reproduction of $C_{org}$
burial distribution in the modern ocean (Sect. 3.2.2).

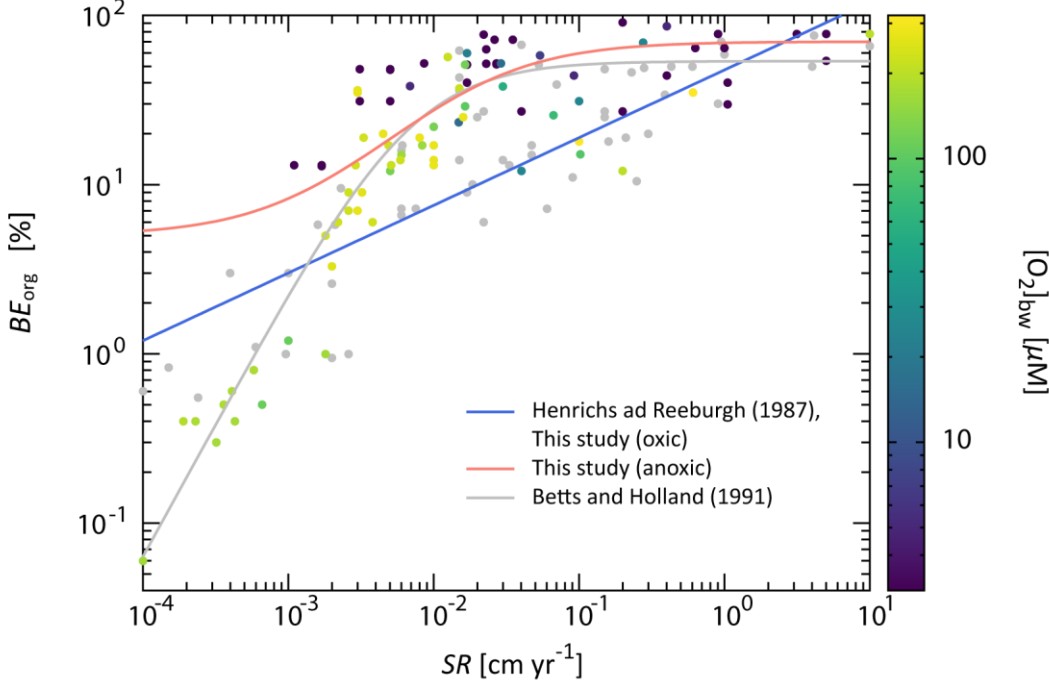


**Figure 6** Burial efficiency of organic carbon ($BE_{org}$) as a function of sedimentation rate ($SR$). The dots denote
the observational data compiled from literature survey (Betts and Holland, 1991; Canfield, 1993; Hartnett et
al., 1998; Henrichs and Reeburgh, 1987; Tromp et al., 1995). The color represents the $O_2$ concentration of
bottom water, $[O_2]_{bw}$, with grey dots for the unknown $[O_2]_{bw}$ value. Black and grey solid lines are previously
proposed empirical relationships (Betts and Holland, 1991; Henrichs and Reeburgh, 1987), whereas blue and
red lines are the relationship for well-oxygenated ($[O_2]_{bw} > 200 \ \mu M$) and anoxic ($[O_2]_{bw} < 30 \ \mu M$) marine
sediments adopted in the CANOPS-GRB model.





Sedimentation rate depends strongly on water depth and distance from shore (Hedges et al., 1999), and we
apply the relationship between water depth, $z$, and the reference value of $SR$ shown by Tromp et al. (1995)
(Fig. 7).

$$z = 2700 \times erfc\left(2.1 + \log SR^*\right). \tag{43}$$

Using these formulas with seafloor topography (Fig. 4a) and $j_{org}^{dep}$ (Eq. 39), we can calculate $j_{org}^{b,ocn}$ for each
ocean depth. In the CANOPS-GRB model, we also introduce an erosion factor, $f_R$, representing the global
weathering/sedimentation rate (Sect. 2.4.3). Given the intimate coupling between global erosion rate and mass
transfer from continents to the ocean, $SR$ scales with the erosion factor ($f_R = 1$ for our reference run):

$$SR(z) = f_R\, SR(z)^*. \tag{44}$$


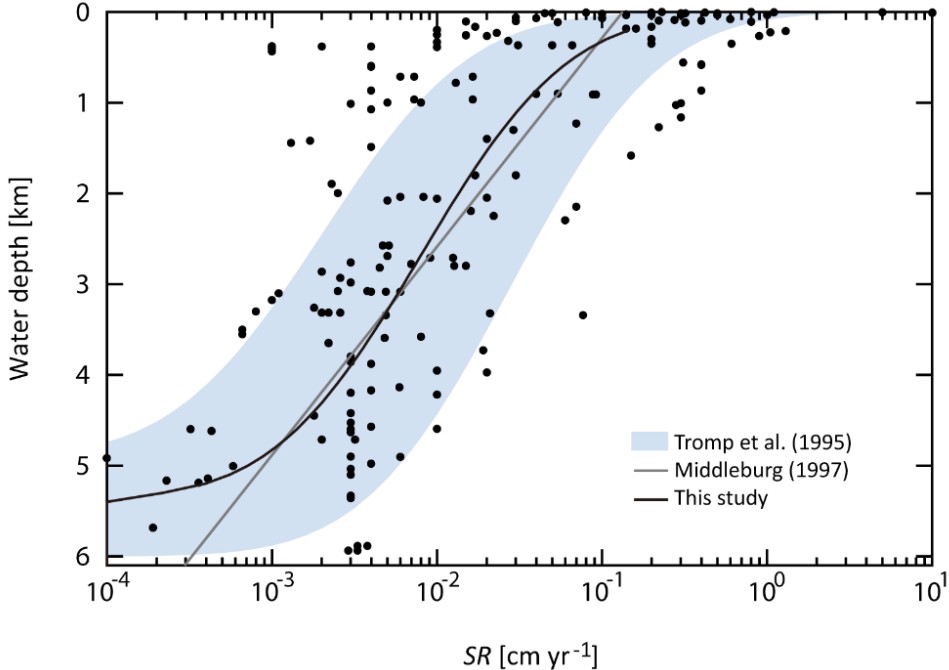


**Figure 7** Sedimentation rate as a function of water depth. Data (black dots) were compiled from literature
survey (Baturin, 2007; Betts and Holland, 1991; Cha et al., 2005; Colman et al., 2000; Reimers et al., 1992;
Tromp et al., 1995). Black line represents the relationship assumed in the CANOPS-GRB model. Previously
estimated empirical relationships (Middelburg et al., 1997; Tromp et al., 1995) are also shown.






Organic matter that is not buried is subject to decomposition. The decomposition rate at each water depth is
given as follows:

$$j_{\text{recy}}^{\text{sed}} = j_{\text{org}}^{\text{dep}} - j_{\text{org}}^{\text{b,ocn}} = \left(1 - BE_{\text{org}}\right) j_{\text{org}}^{\text{dep}}. \tag{45}$$

The respiration pathway used in the benthic decomposition is evaluated based on semi-empirical relationships
obtained by 1-D early diagenesis models (see below). The degradation of organic matter in sediments is
coupled to the availability of terminal electron acceptors, such as $O_2$, $NO_3^-$, and $SO_4^{2-}$ in the water column.
The fraction of aerobic degradation in total sedimentary respiration, $f_{\text{aero}}$, is calculated based on oxygen
exposure time ($\tau_{\text{OET}}$):

$$f_{\text{aero}} = \left(1 - f_{\text{deni}}\right)\left(1 - e^{-k\tau_{\text{OET}}}\right), \tag{46}$$

where $f_{\text{deni}}$ denotes the fraction of denitrification and $k$ is an empirical constant. $\tau_{\text{OET}}$ is given by

$$\tau_{\text{OET}} = \frac{\text{OPD}}{SR}, \tag{47}$$

where OPD is the oxygen penetration depth (cm) and $SR$ denotes a linear sedimentation rate (cm yr$^{-1}$). In the
CANOPS-GRB model OPD is calculated by a simplified parametric law obtained from a 1-D early-diagenetic
model of C and $O_2$. We performed a series of experiments ($n = 5,652$) in order to parameterize OPD as a
polynomial function with the following variables: sedimentation rate $SR$ (cm yr$^{-1}$), bottom water $O_2$
concentration $[O_2]_{\text{bw}}$ ($\mu$M), depositional flux of POC $j_{\text{org}}^{\text{dep}}$ (mmol C cm$^{-2}$ yr$^{-1}$) and bottom water temperature
$T_{\text{bw}}$ (°C). The variables are allowed to vary over a parameter space spanning $10^{-4}$ cm yr$^{-1}$ $< SR < 10^1$ cm yr$^{-1}$,
$10^0$ $\mu$M $< [O_2]_{\text{bw}} < 10^3$ $\mu$M, $10^{-4}$ mmol C cm$^{-2}$ yr$^{-1}$ $< j_{\text{org}}^{\text{dep}} < 10^1$ mmol C cm$^{-2}$ yr$^{-1}$, and 0 °C $< T_{\text{bw}} < 30$ °C.

$$\begin{aligned}
\log \text{OPD} = {} & a_0 + a_1 \log SR + a_2 \log[O_2]_{\text{bw}} + a_3 \log j_{\text{org}}^{\text{dep}} + a_4 (\log SR)^2 \\
& + a_5 (\log[O_2]_{\text{bw}})^2 + a_6 (\log j_{\text{org}}^{\text{dep}})^2 + a_7 (\log SR)(\log[O_2]_{\text{bw}}), \\
& + a_8 (\log[O_2]_{\text{bw}})(\log j_{\text{org}}^{\text{dep}}) + a_9 (\log SR)(\log j_{\text{org}}^{\text{dep}}) + a_{10} T_{\text{bw}}
\end{aligned} \tag{48}$$

where $a_0 = -2.24869$, $a_1 = 0.110645$, $a_2 = 1.12569$, $a_3 = -0.281005$, $a_4 = 0.014827$, $a_5 = -0.124721$, $a_6 =$
0.0894604, $a_7 = 0.00279531$, $a_8 = -0.127797$, $a_9 = 0.0017995$, and $a_{10} = 0.0085171$. This parametric fit
provides a rapid means of obtaining OPD from a 1-D early diagenetic model of C and $O_2$ (Fig. 8). Note that
Eq. (48) is verified for $[O_2]_{\text{bw}} > 1$ $\mu$M. When bottom water $O_2$ concentration is lower than 1 $\mu$M, OPD is set at
zero.




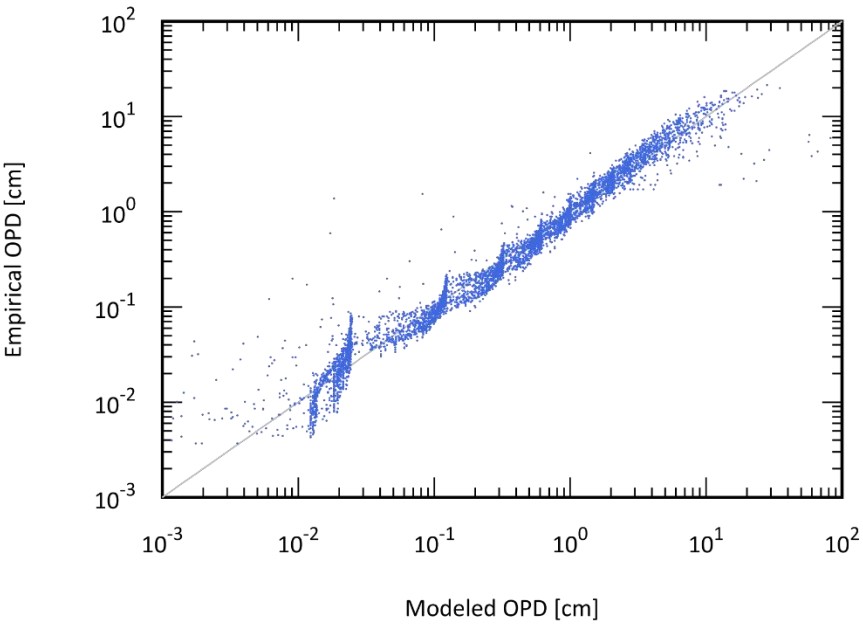


**Figure 8** The correlation between the simulated OPD and the OPD obtained from an empirical relationship of equation (48) ($n = 5652$). Gray line denotes the 1:1 line. $r^2 = 0.9595$.


**Phosphorus cycling**
Marine P inventory is controlled not only by the riverine P input flux from land but also by the efficiency of
P recycling in marine sediments (Van Cappellen and Ingall, 1994). Because the estimated P diffusive flux
from seafloor sediments is much greater than the riverine P flux (Delaney, 1998; Hensen et al., 1998;
McManus et al., 1997; Ruttenberg, 2003; Wallmann, 2003a; Wallmann, 2010a), changes in diagenetic
processes affecting P recycling and burial in marine sediments could have a significant impact on global
oceanic biogeochemical cycles.




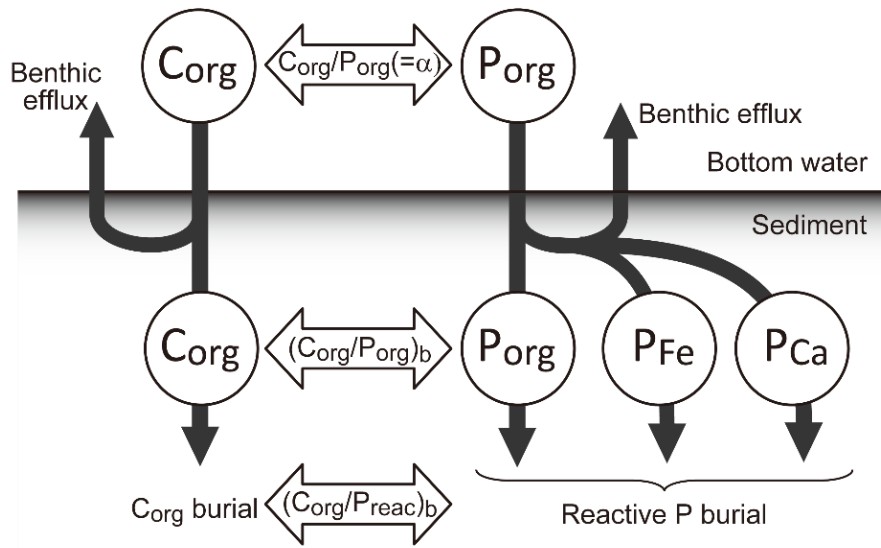


**Figure 9** Schematic of P burial in marine sediments. The primary source of P to the sediment is the deposition of organic matter, which represents the C:P ratio of primary producers, $\alpha$. Most of the deposited organic P is decomposed before burial and the DIP released to pore waters diffuses to the bottom water. A fraction of the liberated P is trapped by iron hydroxides or buried as authigenic minerals (e.g., carbonate fluorapatite). Phosphorus deposited in sediments is a subject of decomposition and sink-switching. Three reactive phases, organic P ($P_{org}$), Fe-sorbed P ($P_{Fe}$), and Ca-bound P ($P_{Ca}$), are considered in the CANOPS-GRB model. The burial of these species are redox-dependent: burial efficiency is affected by bottom water $O_2$ concentration. Because of the sink-switching, sedimentary $C_{org}/P_{reac}$, rather than $C_{org}/P_{org}$ ratios, provides a correct measure of the retention versus diffusive loss of remineralized P.

964

A schematic of benthic P cycling is shown in Fig. 9. The majority of organic matter delivered to the sediment-water interface is regenerated (Jahnke, 1996), but a fraction of DIP released via respiration to pore waters is redistributed to other phases such as iron-hydroxide or carbonate fluorapatite within the sediments. This mechanism is known as 'sink-switching' (e.g., Anderson et al., 2001; Filippelli, 2001), and results in P burial other than organic P playing a more important role in the total P sink (Compton et al., 2000; Ruttenberg, 2003, 1993a). Three different P pools are considered in the CANOPS-GRB model: organic P ($P_{org}$), Fe-hydroxide-sorbed P ($P_{Fe}$), and authigenic Ca-bound P ($P_{Ca}$). The sum of these pools is defined as biologically reactive P ($P_{reac}$) (bioavailable in the ocean to fuel primary productivity). The marine C and P cycles are coupled not only through the $C_{org}/P_{org}$ ratio of POM ($\alpha$) but also through the $C_{org}/P_{reac}$ ratio of marine sediments. It is important to note that as argued by Anderson et al. (2001), the fundamental measure of the retention versus diffusive loss of remineralized P is not the sedimentary $C_{org}/P_{org}$ ratio, but the ratio of $C_{org}/P_{reac}$.




976

Field observations of marine and lacustrine sediments have revealed that the burial efficiency of P depends on

the redox conditions of bottom waters (Ingall and Jahnke, 1994): Phosphorus retention potential is suppressed

under anoxic bottom water conditions. Elevated $C_{org}/P_{reac}$ ratios observed in permanently anoxic environments

suggest preferential regeneration of P relative to C under these conditions (Algeo and Ingall, 2007; Anderson

et al., 2001; Colman et al., 2000; Filippelli, 2001; Ingall and Jahnke, 1997). In the CANOPS-GRB model, P

benthic regeneration rate is calculated at each sediment segment based on the POP depositional flux density

$j_P^{dep}$ ($=j_{org}^{dep}/\alpha$) and P burial efficiency which is a function of both $[O_2]_{bw}$ and $SR$. We assume the following

formulation for the $C_{org}/P_{org}$ ratio of the buried organic phase, on the basis of previous studies (Slomp and Van

Cappellen, 2007; Van Cappellen and Ingall, 1994, 1996):

$$(C_{org}/P_{org})_b = \begin{cases} \dfrac{(C_{org}/P_{org})_b^{oxic}(C_{org}/P_{org})_b^{anox}}{(C_{org}/P_{org})_b^{anox}\dfrac{[O_2]_{bw}}{oxic}+\left(1-\dfrac{[O_2]_{bw}}{oxic}\right)(C_{org}/P_{org})_b^{oxic}}f_\tau & \text{for }[O_2]_{bw}<oxic \\ (C_{org}/P_{org})_b^{oxic}f_\tau & \text{for }[O_2]_{bw}\ge oxic \end{cases}, \quad (49)$$

where $(C_{org}/P_{org})_b^{oxic}$ and $(C_{org}/P_{org})_b^{anox}$ denote organic C/P ratios for fully oxic and anoxic conditions, and

$oxic$ ($=250\ \mu M$) is a threshold value of $[O_2]_{bw}$ below which preferential P regeneration occurs. $(C_{org}/P_{org})_b^{oxic}$

is set to twice the value of the Redfield ratio, and $(C_{org}/P_{org})_b^{anox}$ is an upper limit assumed for organic matter

buried under fully anoxic overlying waters, estimated as 40 times larger than the Redfield value on the basis

of previous studies on various ancient anoxic basin sediments (Ingall et al., 1993; Slomp and Van Cappellen,

2007). In Eq. (49), we also include the dependence of buried $C_{org}/P_{org}$ ratio on $SR$, expressed as $f_\tau$. Modern

observations suggest that $SR$ is a one of the major factors influencing the preservation versus remineralization

of sedimentary organic C and P. Organic C preservation in marine sediments tends to be enhanced at higher

$SR$. In contrast, the $C_{org}/P_{org}$ ratio of sedimentary organic matter shows a non-linear relationship with respect

to $SR$ (Ingall and Cappellen, 1990) (Sect. 3.2.3), suggesting more complex behavior of benthic P cycling.

Specifically, in the pelagic deep ocean, preferential P regeneration is not observed, likely due to the long

timescale of diagenesis prior to burial (Ingall and Cappellen, 1990). In the CANOPS model, $f_\tau$ is formulated

as follows:





$$f_\tau = 0.5 + 0.5\exp\left(-\frac{0.001\,\mathrm{cm}}{SR}\right).\tag{50}$$

Specifically, the $C_{org}/P_{org}$ ratio approaches the Redfield value for oxygenated pelagic sediments.

The burial flux density of $P_{org}$ can be calculated as the $C_{org}$ burial flux density divided by $(C_{org}/P_{org})_b$:

$$j_{Porg}^{b} = \frac{j_{org}^{b,ocn}}{\left(C_{org}/P_{org}\right)_b}.\tag{51}$$

The burial efficiency of $P_{org}$ can be written as follows:

$$BE_{Porg} \equiv \frac{j_{Porg}^{b}}{j_{P}^{dep}} = \frac{j_{Porg}^{b}}{j_{org}^{dep}/\alpha} = \frac{\alpha BE_{org}}{j_{org}^{b}/j_{Porg}^{b}} = \frac{\alpha BE_{org}}{\left(C_{org}/P_{org}\right)_b},\tag{52}$$

where $j_P^{dep}$ denotes the POP settling flux density to sediments, which is coupled to the C/P stoichiometry of
POM $(= j_{org}^{dep}/\alpha)$.

Under oxic bottom water conditions, remineralized organic P can be trapped efficiently at the sediment-water
interface by ferric iron phases. In contrast, under anoxic bottom water conditions, a lack of ferric iron phases
allows most mineralized P to diffuse out of the sediment. This redox-dependent P burial is assumed to be
linearly proportional to the $[O_2]_{bw}$ (Slomp and Van Cappellen, 2007):

$$j_{P\text{-}Fe}^{b} = \begin{cases} BE_{Porg}^{oxic}\left(\dfrac{[O_2]_{bw}}{oxic}\right)j_{Porg}^{dep} & \text{for } [O_2]_{bw} < oxic, \\ BE_{Porg}^{oxic}\, j_{Porg}^{dep} & \text{for } [O_2]_{bw} \geq oxic \end{cases}\tag{53}$$

where $BE_{Porg}^{oxic}$ denotes the burial efficiency of $P_{org}$ under well-oxygenated bottom water conditions ($[O_2]_{bw}$
$> oxic$). We assume that the retention potential of $P_{Fe}$ in oxic sediments is comparable to that of $P_{org}$
(Ruttenberg, 1993b).

Some authors have also proposed that authigenic P burial, the dominant process for P burial today (Ruttenberg,
1993b), depends on the redox conditions of the bottom water (Slomp et al., 2002; Slomp and Van Cappellen,
2007). In the CANOPS-GRB model, we adopt the following redox dependence used by Slomp and Van
Cappellen (2007):





$$j_{P\text{-}Ca}^b = \begin{cases} 2BE_{Porg}^{oxic}\left(a_{auth} + (1-a_{auth})\dfrac{[O_2]_{bw}}{oxic}\right)j_{Porg}^{dep} & \text{for } [O_2]_{bw} < oxic \\ 2BE_{Porg}^{oxic}\, j_{Porg}^{dep} & \text{for } [O_2]_{bw} \geq oxic \end{cases} \tag{54}$$

We assume that $P_{org}$, $P_{Fe}$ and $P_{Ca}$ account roughly for 25%, 25%, and 50%, respectively, of the total reactive
P buried in oxygenated sediments (Ruttenberg, 1993b). Therefore, the burial efficiency of authigenic P phases
is larger than that of $P_{org}$ and $P_{Fe}$ by a factor of 2. The redox-dependency of authigenic P burial is controlled
by a parameter, $a_{auth}$. There is still great uncertainty as to the sensitivity of P retention efficiency of authigenic
P phases to changing redox conditions. For instance, recent modeling study suggests that the burial of
authigenic P is influenced not only by the redox state of bottom water, but also by the seawater chemistry
(especially $Ca^{2+}$) (Zhao et al., 2020). In our reference run, we set $a_{auth}$ at 1, no redox-dependency for authigenic
P burial.

**Nitrogen cycling**


The benthic denitrification rate is estimated with a semi-empirical relationship (Middelburg et al., 1996).
Middelburg and colleagues performed a series of experiments ($n = 2,000$) with a 1-D early diagenetic model
of C-N-$O_2$ to parameterize benthic denitrification $j_{deni}^{sed}$ ($\mu$mol C cm$^{-2}$ d$^{-1}$) as a polynomial function using
$j_{org}^{dep}$ ($\mu$mol C cm$^{-2}$ d$^{-1}$), $z$ (m), and bottom water concentrations of dissolved $O_2$ and $NO_3^-$ ($\mu$M):

$$\begin{aligned} \log j_{deni}^{sed} &= c_0 + c_1 \log j_{org}^{dep} + c_2\left(\log j_{org}^{dep}\right)^2 + c_3 \log[NO_3^-]_{bw}\log[O_2]_{bw} \\ &\quad + c_4 \log[NO_3^-]_{bw} + c_5 \log[O_2]_{bw} + c_6 \log z + c_7 \log j_{org}^{dep}\log[O_2]_{bw} \end{aligned} \tag{55}$$

where $c_0 = -2.2567$, $c_1 = -0.1850$, $c_2 = -0.2210$, $c_3 = -0.3995$, $c_4 = 1.2500$, $c_5 = 0.4721$, $c_6 = -0.0996$, $c_7 = 0.4256$.
This polynomial function was obtained by examining a parameter space spanning 50 m < $z$ < 6,000 m, 10 $\mu$M
< $[O_2]_{bw}$ < 350 $\mu$M, and 1 $\mu$M < $[NO_3^-]_{bw}$ < 60 $\mu$M. $j_{org}^{dep}$ was allowed to vary within 2 orders of magnitude
at each water depth (Middelburg et al., 1996). As pointed out by Romaniello and Derry (2010), the predicted
contribution of denitrification to total decomposition $f_{deni}$ (= $j_{denitr}^{sed}/j_{recy}^{sed}$) can sometimes exceed 100% for
$[O_2]_{bw}$ <10 $\mu$M. When the fraction of benthic denitrification to total decomposition exceeds 90%, benthic
denitrification is limited in order to avoid unphysical values (Ozaki and Tajika, 2013; Romaniello and Derry,

1038  2010).






The burial flux density of $N_{org}$ is calculated by molar ratio of C to N of buried sediments, $(C_{org}/N_{org})_b$, and the
burial flux of $C_{org}$:

$$j_{Norg}^{b} = \frac{j_{org}^{b,ocn}}{\left(C_{org}/N_{org}\right)_b} \tag{56}$$

We assumed an average ratio of 10, which is observed in the Washington and Mexico margin (Hartnett and
Devol, 2003; Hedges et al., 1999).

**Sulfur cycling**


The fractions of MSR and methanogenesis to total decomposition of organic matter in marine sediment are
given by

$$f_{MSR} = \left(1 - f_{aerobic} - f_{denitr}\right)\frac{[SO_4^{2-}]_{bw}}{[SO_4^{2-}]_{bw} + K_{MSR}}, \tag{57}$$

$$f_{meth} = 1 - f_{aero} - f_{deni} - f_{MSR}. \tag{58}$$

The production rate of hydrogen sulfide in sediment, $j_{H2S}^{sed}$ (mol S m$^{-2}$ yr$^{-1}$), is given by

$$j_{H_2S}^{sed} = \frac{1}{2} f_{MSR} j_{recy}^{sed} + j_{AOM}, \tag{59}$$

where $j_{AOM}$ denotes the production rate of sulfide via AOM:

$$j_{AOM} = \frac{1}{2} f_{meth} \frac{[SO_4^{2-}]_{bw}}{[SO_4^{2-}]_{bw} + K_{MSR}} j_{recy}^{sed}. \tag{60}$$

Here we assume that AOM is proportional to the CH$_4$ production rate with a sulfate-dependent term.

The rate of pyrite precipitation in sediments would be proportional to the sulfide production rate at the
sediment-water interface:

$$j_{pyr}^{b,sed} = e_{pyr} j_{H_2S}^{sed}, \tag{61}$$

where the proportional coefficient, $e_{pyr}$, is the pyrite burial efficiency. The rate of MSR is a function of the
marine redox state, [SO$_4^{2-}$], and the availability of degradable organic matter. In the well-oxygenated modern
oceans most sulfide produced in sediments is reoxidized and only a few per cent of total sulfide is buried as





pyrite (Bowles et al., 2014; Canfield, 1991; Jørgensen, 1982; Lin and Morse, 1991; Turchyn and Schrag, 2004).
It has been pointed out that efficient oxidation of sulfide is promoted by animal bioturbation (Berner and
Westrich, 1985; Canfield and Farquhar, 2009). In contrast, the value of $e_{pyr}$ for anoxic sediments is much
greater due to the absence of bioturbation and enhanced sulfide production. We assume that $e_{pyr}$ asymptotes
toward unity with decreasing the bottom water [O$_2$] (Tarhan et al., 2015):

$$e_{pyr} = e_{pyr}^{max} - \left( e_{pyr}^{max} - e_{pyr}^{*} \right) \tanh[O_2]_{bw} , \qquad (62)$$

where $e_{pyr}^{max}$ (= 1 in our reference run) denotes the maximum pyrite precipitation efficiency in anoxic
sediments. The reference value, $e_{pyr}^{*}$, was calibrated using a present-day control simulation such that the
present-day seawater [SO$_4^{2-}$] is ~29 mM. The obtained value of 0.117 is generally consistent with modern
observations (Bottrell and Newton, 2006b; Tarhan et al., 2015; Turchyn and Schrag, 2006a) (see Sect. 3).
Although our approach does not provide a mechanistic description of the complex process of pyrite
precipitation, it is suitable for many purposes.
**Early diagenetic modeling for quantifying the OPD**
A simple 1-D early diagenetic model of C and O$_2$ is employed to obtain the parameterization of OPD (Eq.
(48)). The 100 cm thick sediment is vertically divided into 50 layers with an uneven grid. The grid size
increases from the sediment-water interface ($\Delta z$ = 0.25 mm) to the maximum simulated sediment depth ($\Delta z$ =
1.6 cm). The diagenetic model calculates transport and biogeochemical transformation processes at each grid
point within these sediment columns as well as the sedimentary burial and recycling fluxes at the model
boundaries. The one-dimensional mass conservation equation for POC (wt. %) and dissolved O$_2$ is given by

$$\frac{\partial POC}{\partial t} = D_{bio} \frac{\partial^2 POC}{\partial z^2} - SR \frac{\partial POC}{\partial z} - kPOC , \qquad (63)$$

$$\frac{\partial [O_2]}{\partial t} = D_{O_2} \frac{\partial^2 [O_2]}{\partial z^2} - r_{O_2} kPOC \left( \frac{\rho(1-\phi)}{1.2\phi} \right) \frac{[O_2]}{[O_2] + K_{O_2}} , \qquad (64)$$

where $D_{O2}$ is the diffusion coefficient of O$_2$, $SR$ is the sedimentation rate, and $\phi$ is porosity, which is assumed
to be constant over the entire sediment column for simplicity. Bioturbation is formulated as a diffusive process
with a coefficient $D_{bio}$. The effective diffusion coefficient of O$_2$ is then given by





$$D_{O_2} = \frac{D_{O_2}^{T=0} \times \left(1 + \nu_{O_2} T_{\text{bw}}\right)}{\theta^2} + D_{\text{bio}},$$  (65)

where $D_{O2}^{T=0}$ denotes a tracer diffusion coefficient in seawater of 0°C, $\nu_{O2}$ is a coefficient for temperature
dependence of molecular diffusion coefficient. The in situ diffusion coefficient is further corrected for
tortuosity $\theta$, which is related to pore water resistivity and porosity via the following expressions (Berner, 1980;
Colman and Holland, 2000; Tromp et al., 1995):

$$\theta^2 = \phi F$$  (66)

$$F = \phi^{-m}$$  (67)

where $F$ is the formation factor—defined as the ratio of bulk sediment resistivity to interstitial water
resistivity—and $m$ is an empirical constant, varying with sediment type. We assumed the average value for
unconsolidated muds ($m = 2.7$) in this work (Tromp et al., 1995). The particle mixing coefficient for
bioturbation $D_{\text{bio}}$ is formulated as a function of both sediment accumulation rate and bottom water $O_2$
concentration (Tromp et al., 1995; Wallmann, 2003b):

$$D_{\text{bio}} = 10^{1.63 + 0.85 \log SR} \frac{[O_2]_{\text{bw}}}{[O_2]_{\text{bw}} + K_{O_2}}.$$  (68)

At the bottom of the sediment column, a no-flux condition was applied. The parameters used in the 1-D early
diagenetic model are tabulated in Table 6.

Table 6: Parameters used in the 1-D early diagenetic model.

| Parameters | Label | Value | Unit | Ref. |
|---|---|---|---|---|
| Porosity | $\phi$ | 0.8 | – | |
| Dry bulk density | $\rho$ | 2.6 | g cm$^{-3}$ | |
| $O_2$:C ratio for aerobic respiration | $r_{O2}$ | 1.4 | mol mol$^{-1}$ | |
| Half-saturation constant for aerobic respiration | $K_{O2}$ | 1 | $\mu$M | |
| $O_2$ diffusion coefficient at 0°C | $D_{O2}^{T=0}$ | 281 | cm$^2$ yr$^{-1}$ | |
| Coefficient for a temperature dependence of molecular diffusion coefficient | $\nu_{O2}$ | 0.06 | °C$^{-1}$ | |
| Exponent for the formation factor | $m$ | 2.7 | – | |






### 2.4.5 Air-sea exchange


To calculate the gas exchange of $O_2$, $H_2S$, $NH_3$, and $CH_4$ across the air–sea interface, we employed a stagnant
film model (Liss and Slater, 1974). The flux of a gas X across the air–sea interface is controlled by the
difference in partial pressure between the atmosphere and surface waters, which can be described by the
following formula:

$$J_X^{\text{air-sea}} = v_X^{\text{pis}} \left( [X]_{\text{aq}} - [X]_{\text{sat}} \right), \tag{69}$$

where $v_X^{\text{pis}}$, $[X]_{\text{aq}}$, and $[X]_{\text{sat}}$ denote piston velocity, the dissolved concentration of species $X$, and the saturation
concentration of species $X$, respectively. For $O_2$, the saturation concentration is calculated based on solubility
(Garcia and Gordon, 1992; Sarmiento and Gruber, 2006) and partial pressure:

$$[O_2]_{\text{sat}} = \left( \frac{1000}{22.3916} e^l \right) \left( \frac{pO_2}{pO_2^*} \right), \tag{70}$$

where

$$l = A_0 + A_1 T_{\text{s}} + A_2 T_{\text{s}}^2 + A_3 T_{\text{s}}^3 + A_4 T_{\text{s}}^4 + A_5 T_{\text{s}}^5 + S \times (B_0 + B_1 T_{\text{s}} + B_2 T_{\text{s}}^2 + B_3 T_{\text{s}}^3) + C_0 S^2, \tag{71}$$

$$T_{\text{s}} = \ln\left( \frac{298.15 - T}{273.15 + T} \right), \tag{72}$$

with $T$ in °C. The constants are $A_0 = 2.00907$, $A_1 = 3.22014$, $A_2 = 4.0501$, $A_3 = 4.94457$, $A_4 = -0.256847$, $A_5 =$
$3.88767$, $B_0 = -6.24523 \times 10^{-3}$, $B_1 = -7.3761 \times 10^{-3}$, $B_2 = -1.0341 \times 10^{-2}$, $B_3 = -8.17083 \times 10^{-3}$, and $C_0 = -$
$4.88682 \times 10^{-7}$. The erroneous $A_3 \times T_{\text{s}}^2$ term in the original equation (Garcia and Gordon, 1992) was left out
(Sarmiento and Gruber, 2006).

For $CH_4$, $H_2S$ and $NH_3$, $[X]_{\text{sat}}$ is given by (Kharecha et al., 2005)

$$[X]_{\text{sat}} = K_X^{\text{Henry}} pX, \tag{73}$$

where $K_X^{\text{Henry}}$, and $pX$ denote Henry's law coefficient and the partial pressure of species X, respectively. The
temperature dependence of X's solubility is expressed as:





$$K_X^{\mathrm{Henry}} = K_X^{\mathrm{Henry\circ}} \exp\left[ K_X^T \left( \frac{1}{T} - \frac{1}{298.15} \right) \right], \tag{74}$$

where $K_X^{\mathrm{Henry\circ}}$ denotes the Henry's law coefficient of species X at 25°C, and $K_X^T$ is the temperature
dependence constant.

$[X]_{\mathrm{aq}}$ is the dissolved concentration of X. $[H_2S]_{\mathrm{aq}}$ and $[NH_3]_{\mathrm{aq}}$ can be written as follows:

$$[\mathrm{H_2S}]_{\mathrm{aq}} = \frac{[\Sigma\mathrm{H_2S}]}{1 + K_{\mathrm{H_2S}}^{\mathrm{dis}}/[\mathrm{H^+}]} \tag{75}$$

$$[\mathrm{NH_3}]_{\mathrm{aq}} = \frac{[\Sigma\mathrm{NH_3}]}{1 + [\mathrm{H^+}]/K_{\mathrm{NH_3}}^{\mathrm{dis}}} \tag{76}$$

where $[\Sigma\mathrm{H_2S}] = [\mathrm{H_2S}] + [\mathrm{HS^-}]$ and $[\Sigma\mathrm{NH_3}] = [\mathrm{NH_4^+}] + [\mathrm{NH_3}]$. $K_{\mathrm{H2S}}^{\mathrm{dis}}$ and $K_{\mathrm{NH3}}^{\mathrm{dis}}$ are the dissociation constant,
defined as follows:

$$K_{\mathrm{H_2S}}^{\mathrm{dis}} = \frac{[\mathrm{HS^-}][\mathrm{H^+}]}{[\mathrm{H_2S}]_{\mathrm{aq}}}, \tag{77}$$

$$K_{\mathrm{H_2S}}^{\mathrm{dis}} = \frac{[\mathrm{NH_3}]_{\mathrm{aq}}[\mathrm{H^+}]}{[\mathrm{NH_4^+}]}. \tag{78}$$

Given values of $K_{\mathrm{H2S}}^{\mathrm{dis}}$, $K_{\mathrm{NH3}}^{\mathrm{dis}}$ and $p\mathrm{H}$ (Millero et al., 1988; Yao and Millero, 1995), $[H_2S]_{\mathrm{aq}}$ and $[NH_3]_{\mathrm{aq}}$ can
be calculated.

In the CANOPS-GRB model, atmospheric concentrations of $H_2S$ and $NH_3$ are set at 0. $H_2S$ and $NH_3$ flow past
the surface layer of the ocean to the atmosphere are converted to an equal influx of $SO_4^{2-}$ and $NO_3^-$ to the
surface ocean. The parameters used in the stagnant film model are tabulated in Table 7.

If atmospheric $O_2$ levels are lower than ~1% PAL spatial heterogeneity of the gas exchange flux is expected
(Olson et al., 2016); for example primary productivity (and $O_2$ generation) would be more active in coastal
regions than open-ocean gyres. Because our ocean model resolves only two regions for the surface oceans
(low-mid latitude region L and high latitude region H), it tends to overestimate the oxidation of reductants in





surface mixing layers. To mitigate this model limitation for the $CH_4$ degassing flux, the aerobic oxidation rate
of $CH_4$ is decreased to $1 \times 10^{-7}$ of the standard value in surface layers (Ozaki et al., 2019a).

Table 7: Parameters used in the air-sea exchange module of CANOPS-GRB.

| Parameters | Label | Value | Unit | Ref. |
|---|---|---|---|---|
| Piston velocity of $O_2$ | $v_{O2}^{pis}$ | 1,000 | m yr$^{-1}$ | This study |
| Piston velocity of $NH_3$ | $v_{NH3}^{pis}$ | 300 | m yr$^{-1}$ | (WebBook, 2022) |
| Piston velocity of $H_2S$ | $v_{H2S}^{pis}$ | 1072 | m yr$^{-1}$ | (WebBook, 2022) |
| Piston velocity of $CH_4$ | $v_{CH4}^{pis}$ | 1419 | m yr$^{-1}$ | (WebBook, 2022) |
| Solubility of $NH_3$ at 25°C | $K_{NH3}^{Henry°}$ | $5.6 \times 10^4$ | mol m$^{-3}$ bar$^{-1}$ | |
| Solubility of $H_2S$ at 25°C | $K_{H2S}^{Henry°}$ | 100 | mol m$^{-3}$ bar$^{-1}$ | |
| Solubility of $CH_4$ at 25°C | $K_{CH4}^{Henry°}$ | 1.4 | mol m$^{-3}$ bar$^{-1}$ | |
| Temperature dependence of solubility of $NH_3$ | $K_{NH3}^{T}$ | 4,100 | K | |
| Temperature dependence of solubility of $H_2S$ | $K_{H2S}^{T}$ | 2,100 | K | |
| Temperature dependence of solubility of $CH_4$ | $K_{CH4}^{T}$ | 1,600 | K | |
| Sea surface $p$H at low-mid latitude region | $pH^l$ | 8.17 | – | This study |
| Sea surface $p$H at high latitude region | $pH^h$ | 8.16 | – | This study |
| Partial pressure of atmospheric $NH_3$ | $pNH_3$ | 0 | atm | This study |
| Partial pressure of atmospheric $H_2S$ | $pH_2S$ | 0 | atm | This study |


## 2.5 Land model


## 2.5.1 Net primary productivity


Terrestrial NPP is scaled by global land biomass $V$ normalized to the modern value:

$$J_{NPP}^{lnd} = V \times J_{NPP}^{lnd,*}, \tag{79}$$



where the present value of terrestrial NPP is set at 60 Gt C yr$^{-1}$ (Prentice et al., 2001). The global land biomass
is a function of atmospheric $O_2$ levels:

$$V = f_{UV} f_{fire} f_{O_2},$$  (80)

where $f_{O2}$ represents the direct effect of atmospheric $O_2$ concentration on the $C_3$ plant growth, and $f_{fire}$ denotes
the effect of fires on land biota (Bergman et al., 2004; Lenton and Watson, 2000b):

$$f_{O_2} = \max\left\{1.5 - 0.5\, pO_2^{PAL}, 0\right\},$$  (81)

$$f_{fire} = \frac{k_{fire}}{k_{fire} - 1 + ignit}.$$  (82)

Here $k_{fire}$ (= 3; Lenton, 2013) is the fire frequency constant, and *ignit* is an ignition factor representing the fire
frequency as a function of oxygen (Lenton, 2013; Lenton et al., 2018; Lenton and Watson, 2000b):

$$ignit = \min\left\{\max\left\{c_1\, pO_2 - c_2, 0\right\}, c_3\right\},$$  (83)

with $c_1 = 48$, $c_2 = 9.08$ and $c_3 = 5$ (Lenton, 2013). CANOPS-GRB also includes an additional factor $f_{UV}$
representing the effect of UV on the terrestrial biosphere as a function of atmospheric $O_2$ levels (Ozaki and
Reinhard, 2021):

$$f_{UV} = \tanh\left(\frac{pO_2^{PAL}}{c_{UV}}\right),$$  (84)

where $c_{UV}$ is a model parameter, which, in our standard model is set at 1% PAL, meaning that terrestrial plant
activity is suppressed when atmospheric $O_2$ is lower than a few % PAL.

### 2.5.2 Terrestrial biogeochemical cycles

Phosphorus weathering flux, $J_P^w$ (Eq. (2)), is treated as a boundary condition. A fraction of weathered P is
ultimately buried as terrigenous organic matter (Eq. (3)), whereas the remaining fraction is delivered to the
ocean via rivers (Eq. (4)). In the CANOPS-GRB model, the reference value of $J_P^r$ (= 0.155 Tmol P yr$^{-1}$) is
tuned so that the oceanic P level of the reference state is consistent with modern observations. The burial rate
of terrigenous organic matter (in terms of C) can be written as follows:





$$J_{\text{org}}^{\text{b,lnd}} = \left( C_{\text{org}} / P_{\text{org}} \right)^{\text{lnd}} J_{\text{P}}^{\text{b,lnd}}, \tag{85}$$

where $(C_{\text{org}}/P_{\text{org}})^{\text{lnd}}$ (= 1000) is the average C/P burial ratio of terrigenous organic matter (Bergman et al.,
2004). In this study, the reference value of $J_{\text{org}}^{\text{b,lnd}}$ was set at 3 Tmol C yr$^{-1}$. By combining Eqs. (3), (4), and
(85) for the reference state, the proportional coefficient $k_{11}$ of Eq. (3) is determined by the reference state, as
follows:

$$k_{11} = \frac{J_{\text{org}}^{\text{b,lnd,*}}}{J_{\text{org}}^{\text{b,lnd,*}} + \left( C_{\text{org}} / P_{\text{org}} \right)^{\text{lnd}} J_{\text{P}}^{\text{r,*}}} = 0.0189 \, . \tag{86}$$

The value of $k_{11}$ is treated as a constant in this study.

Almost all organic matter produced by terrestrial NPP is decomposed before burial. The total decomposition
rate is given by:

$$J_{\text{org}}^{\text{r,lnd}} = J_{\text{NPP}}^{\text{lnd}} - J_{\text{org}}^{\text{b,lnd}} \, . \tag{87}$$

CANOPS-GRB includes aerobic respiration and methanogensis as respiration pathways for terrigenous matter,
and the CH$_4$ flux from the terrestrial ecosystem to the atmosphere is evaluated with the assumption that it is
proportional to the burial rate of terrigenous organic matter:

$$J_{\text{CH}_4}^{\text{lnd}} = \left( \frac{J_{\text{org}}^{\text{b,lnd}}}{J_{\text{org}}^{\text{b,lnd,*}}} \right) J_{\text{CH}_4}^{\text{lnd,*}}, \tag{88}$$

where the reference value was set at 1 Tmol CH$_4$ yr$^{-1}$. The net flux of CO$_2$, O$_2$ and CH$_4$ from the terrestrial
ecosystem to the atmosphere can be written, as follows:

$$J_{\text{CO}_2}^{\text{lnd}} = \left( g_{\text{O}_2} + \frac{1}{2}\left(1+\delta\right) g_{\text{CH}_4} \right) J_{\text{org}}^{\text{r,lnd}}, \tag{89}$$

$$J_{\text{O}_2}^{\text{lnd}} = J_{\text{NPP}}^{\text{lnd}} - \left( g_{\text{O}_2} + \delta g_{\text{CH}_4} \right) J_{\text{org}}^{\text{r,lnd}}, \tag{90}$$

$$J_{\text{CH}_4}^{\text{lnd}} = \frac{1}{2}\left(1-\delta\right) g_{\text{CH}_4} J_{\text{org}}^{\text{r,lnd}}, \tag{91}$$





where $g_{O2}$ and $g_{CH4}$ denote the fraction of organic matter decomposed by aerobic respiration and methanogenesis, respectively. $\delta$ represents the fraction of methane that is consumed by aerobic methanotrophy that is a function of $O_2$:

$$\delta = \frac{M_{O_2}^{atm}}{M_{O_2}^{atm} + K'_{O_2}},$$

(92)

with $K'_{O2}$ = $0.273 \times 10^{18}$ mol (Goldblatt et al., 2006). A fraction of organic matter decomposed by methanogenesis, $g_{CH4}$, can be calculated based on Eqs. (89) and (91). Then, $g_{O2}$ is determined from $1 - g_{CH4}$.

### 2.5.3 Weathering

The oxidative weathering of continental crust is a major oxygen sink on geologic timescales, providing a fundamental control on atmospheric $O_2$ levels. The weathering rate in the model is assumed to be proportional to sedimentary reservoir size and a global erosion factor, $f_R$, expressing the effect of continental denudation/erosion on terrestrial weathering:

$$J_{org}^{w} = f_R f_{orgw}^{O_2} \left( \frac{ORG}{ORG^*} \right) J_{org}^{w,*},$$

(93)

$$J_{pyr}^{w} = f_R f_{pyrw}^{O_2} \left( \frac{PYR}{PYR^*} \right) J_{pyr}^{w*},$$

(94)

where $J_{org}^{w}$ and $J_{pyr}^{w}$ denotes the oxidative weathering of organic carbon and pyrite, respectively, and $f_{orgw}^{O2}$ and $f_{pyrw}^{O2}$ represent the $O_2$ dependency. For the oxidative weathering of organic matter, previous biogeochemical models have adapted a $(pO_2^{PAL})^{0.5}$ relationship (Bergman et al., 2004; Lasaga and Ohmoto, 2002). In this study, we employ alternative empirical relationships based on results obtained from a 1-D weathering model (Bolton et al., 2006; Daines et al., 2017):

$$f_{orgw}^{O_2} = c_{orgw} \frac{pO_2^{PAL}}{pO_2^{PAL} + K_{orgw}},$$

(95)

$$f_{pyrw}^{O_2} = c_{pyrw} \frac{pO_2^{PAL}}{pO_2^{PAL} + K_{pyrw}},$$

(96)

where $K_{orgw}$ and $K_{pyrw}$ denote half-saturation constants ($K_{orgw}$ = 0.334 and $K_{pyrw}$ = 0.017) and $c_{orgw}$ and $c_{pyrw}$ are normalized constants ($c_{orgw}$ = 1.334 and $c_{pyrw}$ = 1.017), respectively. The Monod-type relationship captures



the fact that the rate of oxidative weathering reaches its maximum as determined by the erosion rate under
highly oxygenated conditions (i.e., transport-limited regime). For example, due to the fast dissolution kinetics
of pyrite, oxidative weathering can be regarded as transport-limited under modern conditions (Bolton et al.,
2006) (Fig. 10). In the CANOPS-GRB model, $J_{org}^{w*}$ is calibrated based on the global redox budget of the
reference run (see Sect. 2.2.5).

It is important to note that above equations ignore the possible importance of microbial activity and
temperature on the rate of oxidative weathering (Petsch et al., 2001; Soulet et al., 2021). Both represent
important topics for future research.

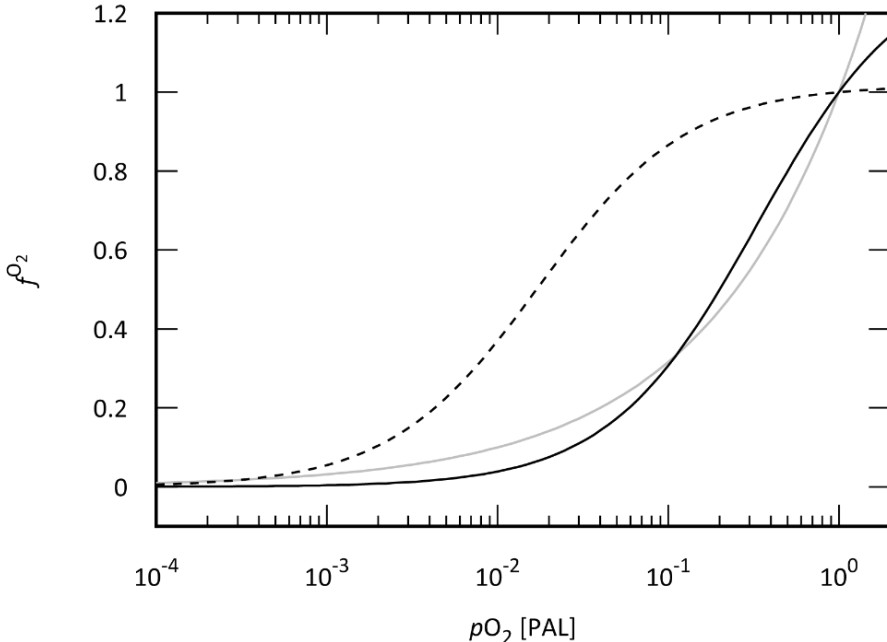


**Figure 10** O$_2$ dependency of the oxidative weathering rate of organic matter and pyrite sulfur. Gray line
denotes the $(pO_2^{PAL})^{0.5}$ relationship assumed in previous biogeochemical models (Daines et al., 2017; Lasaga
and Ohmoto, 2002). Solid and dashed black lines represent the empirical Monod-type relationships for
oxidative weathering of organic matter and pyrite sulfur based on the results obtained from a 1-D weathering
model (Bolton et al., 2006; Daines et al., 2017), which are adopted in the standard model of the CANOPS-
GRB model. PAL = present atmospheric level.





The present riverine flux of sulfur, $J_S^{r*}$, is estimated at 2.6 Tmol S yr$^{-1}$ (Raiswell and Canfield, 2012),
representing the dominant source to the oceans. Riverine flux is written as the sum of the gypsum weathering
flux $J_{gyp}^w$ and the oxidative weathering of pyrite $J_{pyr}^w$ and depends directly or indirectly on the oxidation state
of the atmosphere:

$$J_S^r = J_{gyp}^w + J_{pyr}^w. \tag{97}$$

Based on previous studies (Bergman et al., 2004; Berner, 2009; Markovic et al., 2015; Wortmann and Paytan,
2012), a 3:1 ratio in modern rivers of $SO_4^{2-}$ from gypsum versus pyrite weathering is assumed. Gypsum
weathering flux is assumed to be proportional to its sedimentary reservoir size, $GYP$, and $f_R$:

$$J_{gyp}^w = f_R \left( \frac{GYP}{GYP^*} \right) J_{gyp}^{w*}, \tag{98}$$

where * represents the present value.

In the previous version of the CANOPS (Ozaki et al., 2019a), oxidative weathering of pyrite was divided to
biogenic and abiotic weathering fluxes. In this study, we simplify this (Eq. (94)). Also, oxidative weathering
of Fe(II)-bearing minerals is ignored in this study, which simplifies the framework of the global $O_2$ budget
(Sect. 2.2.5).
**2.5.4 Volcanic degassing**
Volcanic outgassing fluxes of carbon and sulfur are assumed to be proportional to their respective crustal
reservoir sizes:

$$J_{org}^m = \left( \frac{ORG}{ORG^*} \right) J_{org}^{m,*}, \tag{99}$$

$$J_{pyr}^m = \left( \frac{PYR}{PYR^*} \right) J_{pyr}^{m,*}, \tag{100}$$

$$J_{gyp}^m = \left( \frac{GYP}{GYP^*} \right) J_{gyp}^{m,*}. \tag{101}$$

We set the reference value of the volcanic outgassing flux of organic carbon, $J_{org}^{m,*}$, at 1.25 Tmol C yr$^{-1}$
(Bergman et al., 2004). The estimates of modern volcanic fluxes of sulfur fall within the range of ~0.3−3 Tmol





S yr$^{-1}$ (Catling and Kasting, 2017; Kagoshima et al., 2015; Raiswell and Canfield, 2012; Walker and
Brimblecombe, 1985). We adopted a recent estimate of 0.8 Tmol S yr$^{-1}$ (Kagoshima et al., 2015).

### 2.5.5 Sedimentary reservoirs

We extend the original model framework to the explicit calculation of the secular evolution of the sedimentary
reservoirs, linking the biogeochemical cycles in the ocean-atmosphere system to the rock cycle. The mass
balance equation for sedimentary organic carbon (*ORG*) can be written as follows:

$$\frac{dORG}{dt} = J_{\text{org}}^{\text{b}} - J_{\text{org}}^{\text{w}} - J_{\text{org}}^{\text{m}} , \tag{102}$$

where $J_{\text{org}}^{\text{b}}$ denotes the sum of the burial rate of marine and terrigenous organic matter ($J_{\text{org}}^{\text{b,ocn}} + J_{\text{org}}^{\text{b,lnd}}$), the
primary source of sedimentary organic carbon. Primary outputs are oxidative weathering, volcanic outgassing
and metamorphism. Previous estimates of the present reservoir size of *ORG* fall in the range of 1000−1300
Emol (1 E =$10^{18}$) (Berner, 1989; Garrels and Perry, 1974; Mackenzie et al., 1993). We assumed 1250 Emol
for the reference value of *ORG*.

The sedimentary reservoir sizes of pyrite sulfur (*PYR*) and gypsum sulfur (*GYP*) are also written as the balance
between the input (burial) and outputs (weathering and outgassing):

$$\frac{dPYR}{dt} = J_{\text{pyr}}^{\text{b}} - J_{\text{pyr}}^{\text{w}} - J_{\text{pyr}}^{\text{m}} , \tag{103}$$

$$\frac{dGYP}{dt} = J_{\text{gyp}}^{\text{b}} - J_{\text{gyp}}^{\text{w}} - J_{\text{gyp}}^{\text{m}} , \tag{104}$$

where $J_{\text{pyr}}^{\text{b}}$ represents the sum of pyrite precipitation rates in the water column and sediments, $J_{\text{pyr}}^{\text{b,wc}} + J_{\text{pyr}}^{\text{b,sed}}$.
Previous estimates of present reservoir sizes of *GYP* and *PYR* fall in the range of 77−300 Emol and 155−300
Emol (Berner, 2006; Bottrell and Newton, 2006a; Holser et al., 1989; Kump, 1989; Lasaga, 1989; Schlesinger
and Bernhardt, 2013; Sleep, 2005; Yaroshevsky, 2006), respectively. We adopted 200 Emol and 200 Emol for
*GYP*$^{*}$ and *PYR*$^{*}$.





## 2.6 Atmosphere model

### 2.6.1 Hydrogen escape

The rate of hydrogen escape is assumed to be diffusion-limited as it is today. Thus, the total concentration of all H-bearing compounds in the lower stratosphere determines the rate of hydrogen escape (Walker, 1977). For Proterozoic-Phanerozoic atmospheres, $CH_4$ appears to have been the dominant hydrogen-bearing species in the stratosphere, and the flux, $J_{Hesc}$ (mol yr$^{-1}$), is calculated as

$$J_{Hesc} = sM_{CH_4}^{atm},$$ (105)

where $M_{CH4}^{atm}$ denotes the abundance of $CH_4$ in the atmosphere (mol) and $s$ ($= 3.7 \times 10^{-5}$ yr$^{-1}$) is a proportional coefficient (Goldblatt et al., 2006).

### 2.6.2 Photochemistry

CANOPS-GRB includes parameterized $O_2$-$O_3$-$CH_4$ photochemistry that allows quantification of the abundances of atmospheric $O_2$ and $CH_4$. The rate of oxidation of $CH_4$ is calculated by the following empirical parameterization that was obtained from a 1-D photochemistry model (Claire et al., 2006):

$$J_{CH_4ox} = k_{CH_4ox} M_{O_2}^{atm} M_{CH_4}^{atm},$$ (106)

where $M_{O2}^{atm}$ and $M_{CH4}^{atm}$ denote the abundance of $O_2$ and $CH_4$ in the atmosphere (mol). The reaction rate $k_{CH4ox}$ (mol$^{-1}$ yr$^{-1}$) is expressed as a polynomial function of the reservoir sizes of $O_2$ and $CH_4$ (Ozaki and Reinhard, 2021):

$$\log k_{CH_4ox} = \alpha_0^j + \alpha_1^j \cdot \varphi_{O_2} + \alpha_2^j \cdot \varphi_{O_2}^2 + \alpha_3^j \cdot \varphi_{O_2}^3 + \alpha_4^j \cdot \varphi_{O_2}^4 + \alpha_5^j \cdot \varphi_{O_2}^5 + \alpha_6^j \cdot \varphi_{O_2}^6,$$ (107)

where $a_i^j$ are fitting coefficients for given atmospheric $CH_4$ levels and $\varphi_{O2}$ is $\log pO_2$ (in bar) (Supplementary Table 4 of Ozaki and Reinhard, 2021). The oxidation rate was evaluated using Fig. 3 of Claire et al. (2006), showing the oxidation rate as a function of $pO_2$ and $pCH_4$. We took the relationship between $k_{CH4ox}$ and $pO_2$ for $pCH_4$ of $10^{-6}$, $10^{-5}$, $10^{-4}$, $10^{-3}$, $2 \times 10^{-3}$ bar, and $k_{CH4ox}$ is calculated as a function of $pO_2$ and $pCH_4$ with a log-linear interpolation method.





### 2.6.3 Mass balance

CANOPS-GRB accounts for the atmospheric concentrations of $O_2$ and $CH_4$. The atmospheric concentration of $O_2$ is determined by the biogenic source (from the ocean and terrestrial ecosystems) and the consumption through the series of oxidation reaction (the continental weathering of kerogen and pyrite, volcanic outgassing, and photochemical oxidation of methane):

$$\frac{dM_{O_2}^{atm}}{dt} = J_{O_2\uparrow}^{air\text{-}sea} + J_{O_2\uparrow}^{air\text{-}lnd} - \left( J_{Hesc} + 2J_{CH_4ox} \right) - \left( J_{org}^{w} + J_{org}^{m} \right) - 2\left( J_{pyr}^{w} + J_{pyr}^{m} \right), \tag{108}$$

where $M_{O2}^{atm}$ denotes the mass of $O_2$ in the atmosphere (moles), and the first and second term on the right hand side represents the biogenic flux of $O_2$ from marine and terrestrial ecosystems. The third term denotes $O_2$ consumption via photochemistry, and the fourth and fifth terms are the $O_2$ consumption via organic C and pyrite S sub-cycles.

The abundance of $CH_4$ in the atmosphere, $M_{CH4}^{atm}$, is determined by input from the ecosystems and the consumption of $CH_4$ via photolysis, as well as by the hydrogen escape:

$$\frac{dM_{CH_4}^{atm}}{dt} = J_{CH_4\uparrow}^{air\text{-}sea} + J_{CH_4\uparrow}^{air\text{-}lnd} - \left( J_{Hesc} + J_{CH_4ox} \right). \tag{109}$$

No abiotic $CH_4$ input via hydrothermal systems is included.





## 3 Validation against the modern global ocean

Here, a steady-state simulation mimicking the present-day condition was run to evaluate the overall performance of CANOPS-GRB. To do this, the ocean model was run until reaching the steady state, assuming the present atmospheric $O_2$ level and reference values of boundary fluxes (weathering and volcanic fluxes). The simulated circulation and biogeochemistry for the modern global ocean was compared with modern oceanographic observations from the Global Ocean Data Analysis Project (Key et al., 2015; Olsen et al., 2016).

### 3.1 Distribution of circulation tracers

Comparisons of model output with circulation tracers, such as potential temperature ($\theta$) and radiocarbon ($\Delta^{14}C$), permit a test of the physical exchange scheme. Figure 11 depicts the simulated patterns of physical tracers with observational data. The physical circulation in the model generally agrees well with oceanic observations, although we note that model temperatures for low-mid latitudes above 1,000 m water depth tend to be higher than observed because temperature distribution in the real ocean is strongly controlled by vertical structure and advective processes that are not captured in our simple circulation scheme. Despite this model limitation, the modelled temperature distribution generally reproduces the observed distribution. The $\Delta^{14}C$ minimum in the model for the low-mid latitude region corresponds well with observations. The modelled background radiocarbon for young deep waters (about -150±25‰) is closer to the value for the Southern Ocean (approximately -150‰) than for North Atlantic deep waters (approximately -80‰), and old deep waters (-200±15‰) correspond to the South Pacific. We conclude that the simulated circulation tracers generally match well with ocean data.



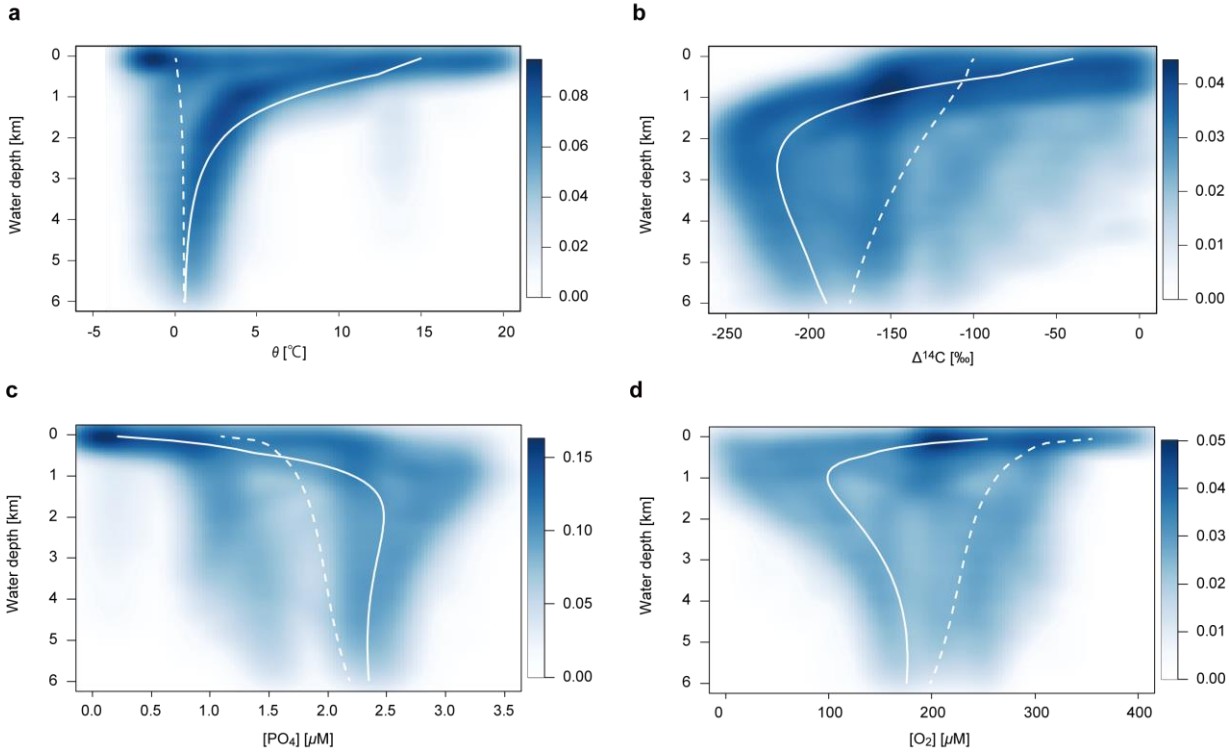

**Figure 11** Simulated steady state depth-profiles of (**a**) potential temperature, $\theta$, (**b**) radio carbon, $\Delta^{14}C$, (**c**) DIP (dissolved inorganic phosphorus, $PO_4^{3-}$), and (**d**) dissolved oxygen, $O_2$. Solid and dashed white lines denote the simulated profiles for LD and HD region, respectively. Simulation results are compared with the dataset from the Global Ocean Data Analysis Project (GLODAP) data base (GLODAPv2_2019; Key et al., 2015; Olsen et al., 2016). The color represents the density of observational points.

### 3.2 Ocean biogeochemistry

Having demonstrated that CANOPS-GRB's ocean circulation model does a reasonable job of representing water mass exchange, we next assess the performance of the oceanic biogeochemistry model by comparing its output to ocean biogeochemical data. Model-generated global fluxes and inventories of C, P, N and S cycles are summarized in Fig. 12. Those compare well with independent observational estimates. Below, we provide a brief discussion of globally-integrated biogeochemical flux estimates.



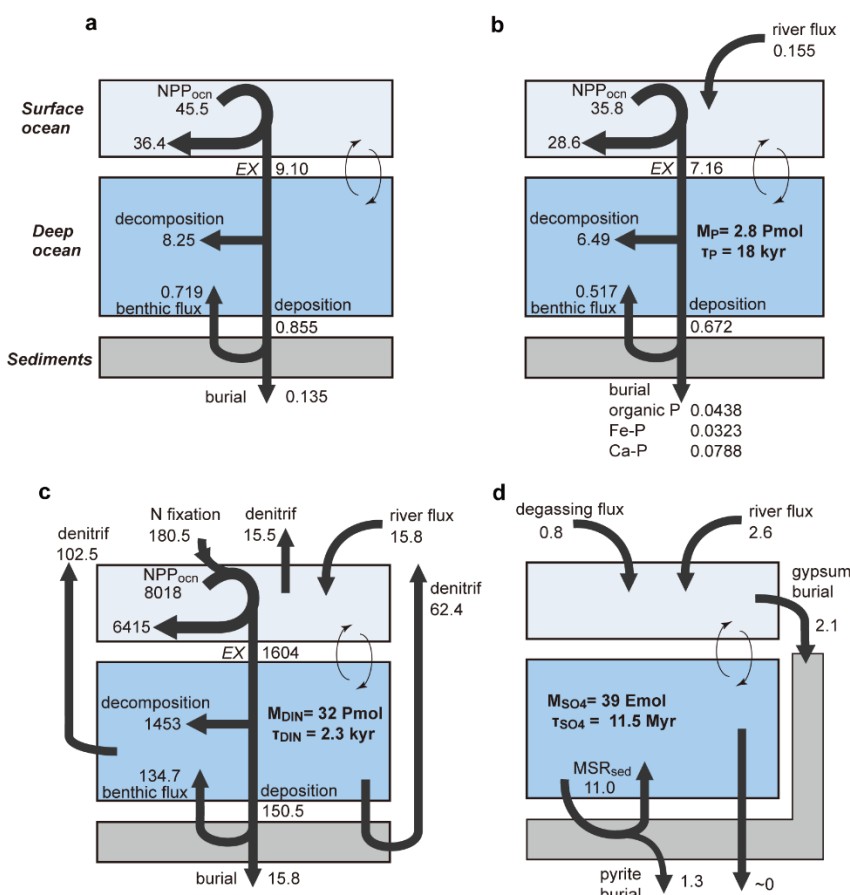

**Figure 12** Schematics of the simulated material flow in the ocean for the reference run. (**a**) Organic carbon (in Gt C yr$^{-1}$), (**b**) phosphorus (in Tmol P yr$^{-1}$), (**c**) nitrogen (in Tg N yr$^{-1}$), and (**d**) sulfur (in Tmol S yr$^{-1}$). NPP$_{ocn}$ = oceanic net primary production. $EX$ = export production. $M_X$ = mass of X in the ocean. $\tau_X$ = residence time of X in the ocean. Pmol = $10^{15}$ mol.

### 3.2.1 Distribution of nutrients and oxygen

The simulated vertical profile of phosphate captures the characteristic features and values of observational data (Fig. 11c). More specifically, the distribution in the low-mid latitude region is more similar to that in the Pacific and Indian Ocean, and distribution of high-mid latitude region is similar to that in the Southern Ocean. This is a consequence of limiting high-latitude productivity (preformed DIP is 1.1 $\mu$M) which results in higher concentrations in the ocean interior. The model dissolved O$_2$ profile for low-mid latitude shows a minimum of approximately 100 $\mu$M at water depth of 1,000 m, corresponding to the oxygen minimum zone (Fig. 11d).



In contrast, dissolved $O_2$ for high-mid latitude sector (HD) shows a monotonically decreasing trend. This is because of oxygen consumption via POM decomposition during downwelling.

### 3.2.2 Carbon cycling

The marine export/new production in our model is 9.1 Gt C yr$^{-1}$ (8.36 Gt C yr$^{-1}$ at L and 0.74 Gt C yr$^{-1}$ at H). This is consistent with previously estimated global values of 8.5–12 Gt C yr$^{-1}$ (Dunne et al., 2007a; Heinze et al., 2009; Laws et al., 2000; Sarmiento and Gruber, 2006). In particular, our estimate is close to the mid-point of the previously estimated range of 9.6±3.6 Gt C yr$^{-1}$ (Dunne et al., 2007b). This is a marked improvement from earlier studies with box models which have underestimated marine new production by a factor of 2 or more (Archer et al., 2000; Shaffer et al., 2008). Simulated global oceanic NPP is 45.5 Gt C yr$^{-1}$. This is also consistent with the previous estimated range of 44–65 Gt C yr$^{-1}$ (Berelson et al., 2007; Carr et al., 2006; Prentice et al., 2001; Woodward, 2007).

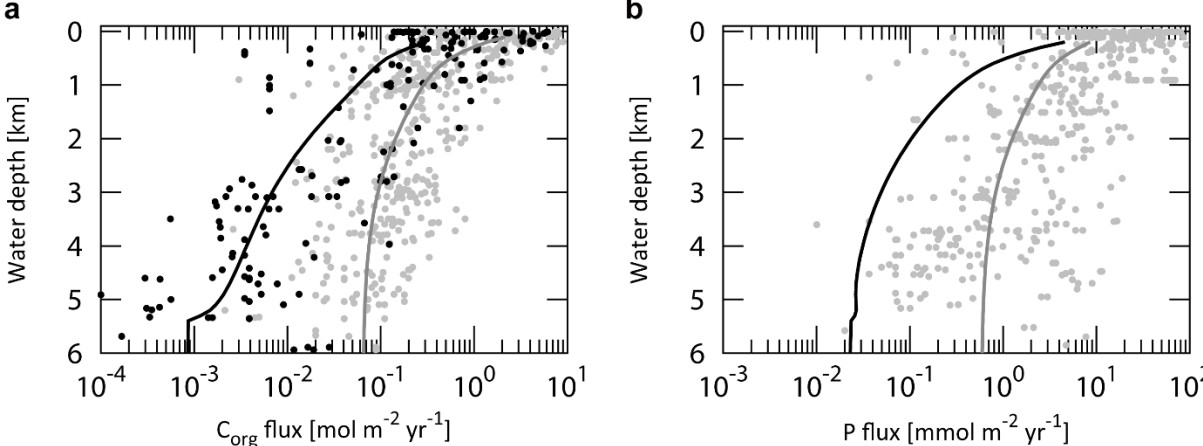

**Figure 13** Simulated steady-state depth-profiles of organic C and reactive P flux density for the LD region. In (**a**), gray dots denote observations of depositional/settling flux density, whereas black dots represent observations of burial flux density compiled from literature survey (Baturin, 2007; Betts and Holland, 1991; Colman et al., 2000; Lutz et al., 2002). Gray and black solid lines denote the simulated POC depositional and burial flux densities obtained from the reference run. (**b**) Gray dots denote the benthic P efflux density obtained from literature survey (Colman et al., 2000; Hartnett and Devol, 2003; Hensen et al., 1998; Ingall and Jahnke, 1994, 1997; McManus et al., 1997; Schenau and De Lange, 2001; Zabel et al., 1998), whereas gray and black solid lines represent the simulated benthic P efflux density and burial flux density of reactive P obtained from the reference run.





The global marine POC flux depends largely on water depth. Model-generated fluxes compare well with independent estimates of deposition, burial, and regeneration. The gray line in Fig. 13a shows the simulated sinking flux density of POC in the water column for LD region, compared with observations (Archer et al., 2002; Baturin, 2007; Betts and Holland, 1991; Lutz et al., 2002). The preferential consumption of labile compounds ($G_1$ and $G_2$) during the settling process leads to a continuous decrease in reactivity and therefore, remineralization rates from the surface ocean down to the deep. Our estimate lies well within the range of observations. The model tends to give lower fluxes than observed above 2,000 m water depth, and higher below 5,000 m water depth. This is probably because of the assumption of homogeneous productivity in the surface ocean. In the real ocean, oceanic productivity is generally greater at the continental margins than in the pelagic gyre regions (Lutz et al., 2002). This is a model limitation, but the simulated biological pump is sufficient to describe the general characteristics of global ocean biogeochemistry.

Of total exported POC, 91% (8.25 Gt C yr$^{-1}$) is decomposed in the water column and the rest (0.85 Gt C yr$^{-1}$) sinks to the sediment surface (Fig. 12a). The simulated global POC depositional flux is comparable not only with observational estimates of 0.93 Gt C yr$^{-1}$ (Muller-Karger et al., 2005) and 0.67±0.48 Gt C yr$^{-1}$ for off-shore regions (Dunne et al., 2007b), but also with an estimate using EMIC (0.87 Gt C yr$^{-1}$) (Ridgwell and Hargreaves, 2007). The depositional fluxes of $C_{org}$ in marginal (<2,000 m) and deep-sea sediments (>2,000 m) are estimated at 0.58 Gt C yr$^{-1}$ and 0.27 Gt C yr$^{-1}$, respectively. These estimates are slightly lower than previous estimates of 0.62–1.98 Gt C yr$^{-1}$ and 0.31–0.62 Gt C yr$^{-1}$ (Bohlen et al., 2012; Burdige, 2007; Dunne et al., 2007a; Muller-Karger et al., 2005).

In our standard run, benthic remineralization removes 7.9% of the exported POC (0.72 Gt C yr$^{-1}$), equivalent of 84% of the global POC sedimentation rate. As a result, only 1.5% (0.135 Gt C yr$^{-1}$ or 11.3 Tmol C yr$^{-1}$) of the global POC export production is ultimately buried in marine sediments. Our model demonstrates that much (91%) of the total burial occurs on the continental margins (<2,000 m water depth), where the settling flux and burial efficiency are relatively high. Previous studies (Dunne et al., 2007b; Muller-Karger et al., 2005) estimated a $C_{org}$ burial rate of 0.29±0.15 Gt C yr$^{-1}$ and >0.06±0.06 Gt C yr$^{-1}$ at the margin. Our estimate of



0.123 Gt C yr$^{-1}$ lies between these values, whereas our estimate for the deep sea, 0.012 Gt C yr$^{-1}$, is on the
lower end of previous estimates of 0.012±0.02 Gt C yr$^{-1}$ (Dunne et al., 2007b) and 0.017±0.005 Gt C yr$^{-1}$
(Hayes et al., 2021). Also, Hayes et al. (2021) and Sarmiento and Gruber (2006) estimated the burial rate
below 1,000 m as 0.02±0.006 Gt C yr$^{-1}$; our estimate of 0.019 Gt C yr$^{-1}$ is consistent with this. Combined with
the prescribed burial rate of terrigenous $C_{org}$ 0.036 Gt C yr$^{-1}$ (3 Tmol C yr$^{-1}$), the total burial rate is calculated
to be 0.17 Gt C yr$^{-1}$ (14.3 Tmol C yr$^{-1}$). This is somewhat higher than previous estimates (Berner, 1982;
Burdige, 2005; Muller-Karger et al., 2005), but given the rather large uncertainty we consider it defensible.

Figure 14 shows OPD as a function of water depth. Although the benthic data could be biased towards highly
specific environments, such as sediments underlying upwelling areas and continental margins, our estimates
capture the general features of modern observations.

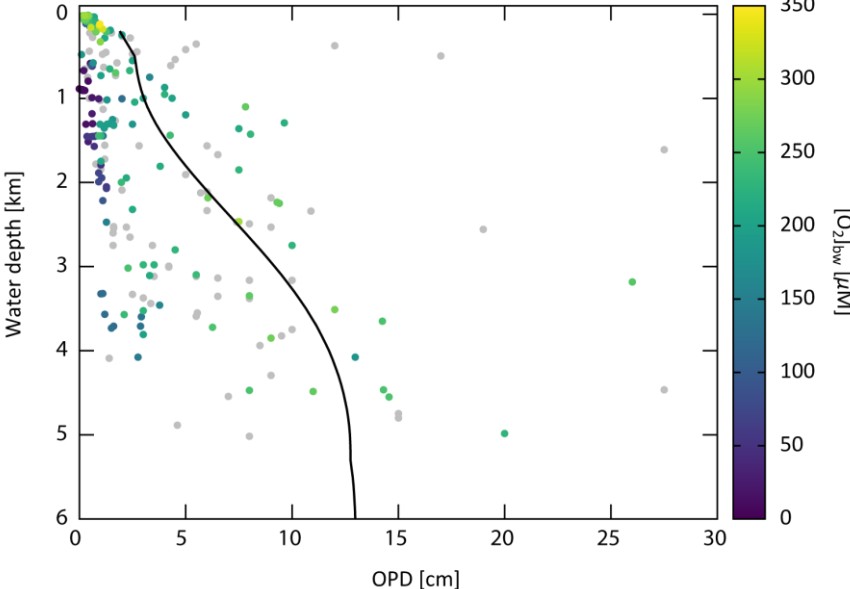


**Figure 14** Oxygen penetration depth (OPD) as a function of water depth. Color dots denote the observational
data obtained from literature survey (Bradley et al., 2020; Devol and Christensen, 1993; Donis et al., 2016;
Gundersen and Jorgensen, 1990; Hartnett et al., 1998; Hedges et al., 1999; Hyacinthe et al., 2001; Martin and
Sayles, 2014; McManus et al., 2005; Morford and Emerson, 1999; Nierop et al., 2017; Pfeifer et al., 2002;
Rowe et al., 2008; Sachs et al., 2009). Colors represent the $O_2$ concentration of bottom water (Gray dots
represents the unknown dissolved $O_2$ concentration). The simulated OPD obtained from the reference run is
shown as a black line.






### 3.2.3 Phosphorus cycling


The removal of phosphate from surface waters occurs through photosynthetic fixation by primary producers
and subsequent export in the form of POP into deeper waters, where it is largely remineralized back into DIP.
Through this process there is a vertical partitioning of DIP within the ocean with reduced surface
concentrations. Phosphorus export production is 7.16 Tmol P yr$^{-1}$, which is coupled with carbon according to
the POM compositional ratio (C:P = 106:1 for our standard model). The remineralization in the water column
(6.49 Tmol P yr$^{-1}$) and total sedimentation rate (0.672 Tmol P yr$^{-1}$) are also proportional to those of POC. In
contrast, the benthic DIP flux is decoupled from the carbon flux. Figure 13b shows modelled DIP benthic
efflux and burial flux together with observed fluxes. Some observational data showing a relatively large
abyssal (4–6 km) benthic flux are from upwelling regions in the South Atlantic (Hensen et al., 1998). The
deviation is therefore not critical for our globally averaged model. Our model gives the total benthic efflux of
DIP as 0.517 Tmol P yr$^{-1}$, which is roughly three times the riverine reactive P input rate. This is within the
range of previous estimates of 0.05–1.25 Tmol P yr$^{-1}$ (Colman and Holland, 2000; Compton et al., 2000;
Wallmann, 2003b; Wallmann, 2010b).

The preservation efficiency (here defined as burial flux divided by the export flux) of P is 2.1%. This is higher
than that of organic carbon (1.5%), indicating that more P is trapped in marine sediments than might be
expected from Redfield stoichiometry. In oxic marine sediments, a fraction of the DIP released to pore waters
from POM decomposition can be absorbed by iron-oxyhydroxide or precipitated as authigenic fluorapatite
(Fig. 9). Therefore, the global averaged $C_{org}/P_{reac}$ ratio of buried sediments is generally less than the Redfield
of 106 (approximately 65±25 based on observations; Algeo and Ingall, 2007). The modelled global average
$C_{org}/P_{reac}$ ratio of buried sediment is 73, consistent with this. The P burial fluxes of organic P, Fe-bound P and
authigenic P are estimated at 0.044 Tmol P yr$^{-1}$, 0.032 Tmol P yr$^{-1}$, and 0.079 Tmol P yr$^{-1}$, respectively.

The $C_{org}/P_{org}$ ratio of burying organic matter shows a non-linear relationship with respect to sedimentation
rate. The observed $C_{org}/P_{org}$ ratios are generally greater than the Redfield value of 106, especially for sediments





in oxygen minimum zones (OMZs), which are characterized by a high depositional flux of organic matter
($C_{org}/P_{org}$ ratios up to 600 for the present open ocean) (Ingall and Cappellen, 1990). For example, the averaged
$C_{org}/P_{org}$ molar ratio at the Peru-Chile OMZ and Black Sea are 600, and the estimated burial ratio of sapropel
S1 of Mediterranean Sea is in the range of 400−800 (Slomp et al., 2002). This reflects the preferential
regeneration of P relative to C during microbial remineralization of marine organic matter and reflect the more
labile nature of P-biochemicals relative to most non-phosphorus containing organic carbon compounds.
Additional rationale for this observation is that P is preferentially targeted for remineralization to support
subsequent biological productivity as an essential and potentially limiting nutrient. Our model demonstrates
that we can reproduce the first-order relationship between $C_{org}/P_{org}$ and sediment accumulation rate (Fig. 15).

The modelled marine DIP inventory is $2.75 \times 10^{15}$ mol, consistent with the observational estimate of around
$3 \times 10^{15}$ mol (e.g., Delaney, 1998; Guidry et al., 2000). Given the riverine reactive P input flux of 0.155 Tmol
P yr$^{-1}$, the phosphorus residence time is estimated at 18 kyr, which is also consistent with previous estimates
of 20 kyr or shorter (Benitez-Nelson, 2000; Ruttenberg, 2003).

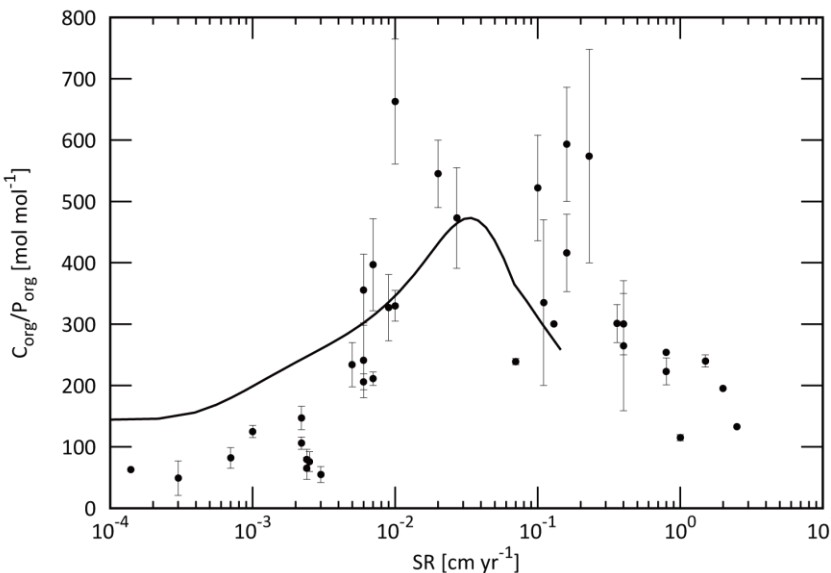


**Figure 15** $C_{org}/P_{org}$ ratios of buried sediments as a function of sedimentation rate. Black dots represent the
observational data (Ingall and Cappellen, 1990). The simulated $C_{org}/P_{org}$ ratios for the LD region obtained
from our reference run is shown as a black line.






### 3.2.4 Nitrogen cycling

Nitrogen export production is 1603 Tg N yr$^{-1}$, which is coupled with carbon according to the C:N
stoichiometry of organic matter. Simulated N fixation required for the N balance in the ocean is 180 Tg N yr$^{-}$
$^{1}$, which is higher than the range of previous estimates of 110–150 Tg N yr$^{-1}$ (Deutsch et al., 2007; Duce et al.,
2008; Fowler et al., 2013; Galloway et al., 2004; Gruber and Sarmiento, 1997; Karl et al., 2002; Luo et al.,
2012), while a recent study (Großkopf et al., 2012) suggests a higher value of ~180 Tg N yr$^{-1}$. This discrepancy
is partly because atmospheric deposition is ignored in the CANOPS-GRB model, which contributes 25.8 Tg
N yr$^{-1}$ (Wang et al., 2019). Gruber and Sarmiento (2002) estimated the pre-industrial value of the total source
of N as 188±44 Tg N yr$^{-1}$. Our estimate of 196 Tg N yr$^{-1}$ is within this range.

Nitrogen fluxes in an oxic water column are tightly coupled with the C$_{org}$ fluxes, whereas decoupling appears
in suboxic environments. Simulated denitrification in the water column is 102 Tg N yr$^{-1}$, within the range of
the observational estimates (50–150 Tg N yr$^{-1}$) (Brandes and Devol, 2002; DeVries et al., 2012; DeVries et
al., 2013; Gruber, 2008; Gruber and Sarmiento, 2002; Oschlies et al., 2008; Wang et al., 2019). Modelled
benthic denitrification is 62 Tg N yr$^{-1}$, which is lower than the estimated range of 90–300 Tg N yr$^{-1}$ (Brandes
and Devol, 2002; Devol, 2015; DeVries et al., 2012; DeVries et al., 2013; Eugster and Gruber, 2012; Wang
et al., 2019) by a factor of 1.5−5, suggesting that further efforts are required to improve representation of this
process. One possible explanation for this discrepancy is that our model is not sufficient to express benthic N
cycling because we ignore localized upwelling regions (such as the eastern Tropical Pacific and the Arabian
Sea) and coastal regions where benthic denitrification is significant POM decomposition pathway in favor of
globally averaged parameterizations. The separate treatment of continental shelves and margin sediments from
the pelagic ocean could improve this issue. We also ignore another denitrification mechanism: anaerobic
ammonium oxidation (anammox), which will often play an important role in the loss of fixed nitrogen in
marine sediments and pelagic anoxic zones (Karthäuser et al., 2021; Kuypers et al., 2005).





The modelled DIN inventory is $4.5 \times 10^5$ Tg N. Given the total source flux of 196 Tg N yr$^{-1}$, the residence time
of DIN is estimated at 2.3 kyr.

### 3.2.5 Sulfur cycling

MSR is a major early diagenetic pathway of carbon oxidation in organic-rich sediments deposited below
oxygenated waters. For the standard run, aerobic oxidation is a dominant process in the water column, but
MSR contributes 37% of benthic degradation. CANOPS-GRB estimates a global rate of benthic sulfate
reduction at 11 Tmol S yr$^{-1}$. This is lower than the previously reported value of gross MSR (40–75 Tmol S yr$^{-1}$
; Canfield and Farquhar, 2009; Jørgensen and Kasten, 2006) but agrees better with net MSR (Bowles et al.,
2014). Bowles et al. (2014) have estimated global net MSR at 6.2 Tmol S yr$^{-1}$ and 11.3 Tmol S yr$^{-1}$ for $z >$
200 m depth and $z > 0$ m depth, respectively. Our estimate is within this range. MSR is most pronounced on
the shelf where high fluxes of organic matter to the seafloor lead to shallow OPD, high sulfide production,
and consequently high pyrite precipitation (Fig. 16).

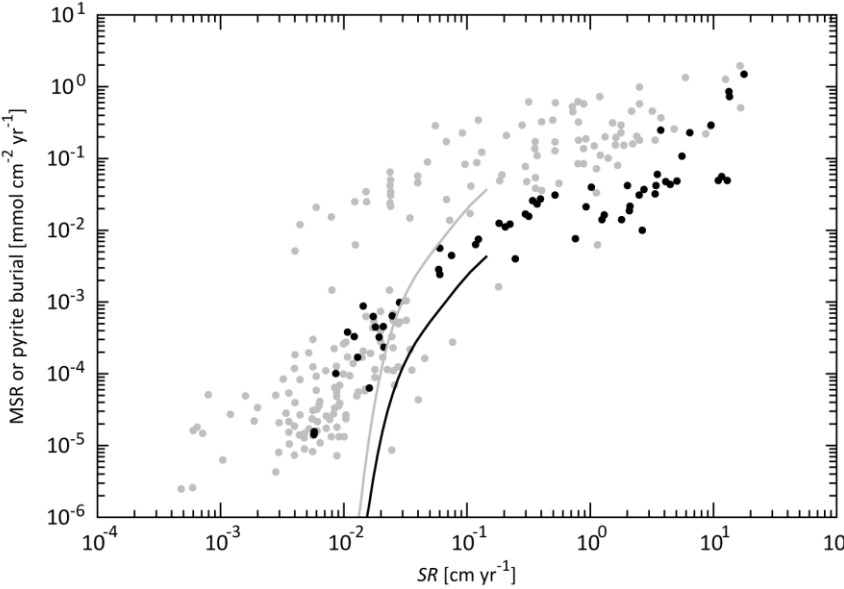


**Figure 16** MSR and pyrite burial flux density as a function of sedimentation rate. Gray and black dots depict
observational data compilation of depth-integrated MSR flux density and pyrite burial flux density for normal
(oxic) marine sediments (Berner and Canfield, 1989; Canfield, 1989; Raiswell and Canfield, 2012). The unit
of sedimentation rate was converted from g cm$^{-2}$ yr$^{-1}$ to cm yr$^{-1}$ with assuming the dry bulk density of 2.5 g
cm$^{-3}$ and porosity of 0.9. Solid lines are the results obtained from the reference run of the CANOPS-GRB
model.





In the CANOPS-GRB model, pyrite burial efficiency $e_{pyr}$ (Sect. 2.3.4) for sediments underlying oxic bottom
waters is set such that simulated seawater [$SO_4^{2-}$] of the reference run is consistent with the modern value of
28.9 mM. The tuned value of 11.7% agrees well with observations suggesting that pyrite precipitation rate is
about 10−20% of the rate of MSR (Fig. 16). Our reference value is also consistent with other estimates of
11−20% (Bottrell and Newton, 2006a; Tarhan et al., 2015; Turchyn and Schrag, 2006b).

The sulfate inventory of our reference state is $39\times10^{18}$ mol. Given the total source flux of 3.4 Tmol S yr$^{-1}$, the
residence time of sulfate is 11.5 Myr.

## 3.3 Global oxygen cycling

The global $O_2$ budget for our reference state is shown in Fig. 17. The simulated $O_2$ inventory in the ocean-
atmosphere system is $38\times10^{18}$ mol (atmosphere $= 38\times10^{18}$ mol, ocean $= 0.23\times10^{18}$ mol). Organic carbon burial
represents a major $O_2$ source flux (marine = 11.3 Tmol $O_2$ equiv. yr$^{-1}$ and terrigenous = 3 Tmol $O_2$ equiv. yr$^{-1}$
$^{1}$). Pyrite burial and hydrogen escape to space contribute 2.6 Tmol $O_2$ equiv. yr$^{-1}$ and 0.001 Tmol $O_2$ equiv.
yr$^{-1}$, respectively. Given the total source/sink flux of 16.9 Tmol $O_2$ yr$^{-1}$, the residence time of O= in the ocean-
atmosphere system of our reference state is estimated as 2.26 Myr, which is consistent with previous estimates
of 2−4 Myr (Berner, 1989; Berner, 2004b; Garrels and Perry, 1974).

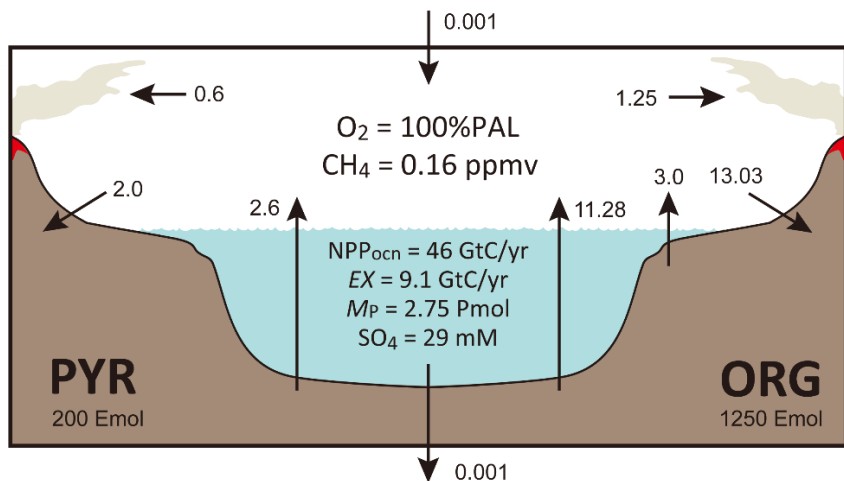


**Figure 17** Schematics of global redox ($O_2$) budget for the reference run. Arrows represent the $O_2$ flux in terms
of $10^{12}$ mol $O_2$ equiv. yr$^{-1}$. PAL = present atmospheric level. Pmol = $10^{15}$ mol. Emol = $10^{18}$ mol. ORG =
sedimentary organic carbon. PYR = sedimentary pyrite sulfur.





## 4 Sensitivity experiment

Based on the results obtained above, we conclude that the CANOPS-GRB model is sufficient to describe basic biogeochemical characteristics in the modern ocean-atmosphere system. As a next step, we assess the dynamic response of the full model by performing sensitivity experiments with respect to P availability in surface environments.

### 4.1 Dynamic response to changes in P weathering

Here, we conduct a sensitivity experiment with respect to the P weathering rate in order to see how the atmospheric and oceanic $O_2$ levels respond to changes in P availability in the exogenic system over a wide range of timescales. Specifically, we performed four simulations, varying the values of $f_P$ in Eq. (2) over two orders of magnitude. The reference state presented in the previous section is assumed for the initial condition, and the full model is allowed to evolve freely for three billion model years. These experiments demonstrate how P availability in surface environments affects global biogeochemical cycles and redox states of the atmosphere and oceans over a diverse range of timescales.

The simulated transient response is shown in Fig. 18. As expected, lower P availability leads to lower oceanic P inventory (Fig. 18a), resulting in suppressed biological productivity in the ocean (Fig. 18b). Given the residence time of P in the ocean (20 kyr, see Sect. 3.2.4), these responses occur within $10^5$ yr. The suppressed biological productivity leads to a decline of burial rate of organic matter in sediments (Fig. 18c). Specifically, 10% and 1% of $f_P$ give rise to the burial rate of marine $C_{org}$ of 1 Tmol C yr$^{-1}$ and 0.13 Tmol C yr$^{-1}$ at $10^5$ yr, respectively (cf. the reference value of 11.3 Tmol C yr$^{-1}$).

On the timescales of $10^5$–$10^6$ yr, the system reaches a quasi-steady state, but there is still a large redox imbalance due mainly to the suppression of $C_{org}$ burial (Fig. 18i). This gives rise to deoxygenation of the atmosphere on a time scale of millions of years (Fig. 18d). Note that once the ocean interior becomes anoxic, the enhanced P recycling and preservation of organic matter in anoxic marine sediments tend to buffer the atmospheric deoxygenation (Figs. 18a and c). However, these passive responses do not alter the fundamental behavior: lower P availability results in lower atmospheric $O_2$ levels. After atmospheric deoxygenation (>~4



1521 Myr), the system again reaches its quasi-steady state. Specifically, $f_P$ values of 10% and 1% result in

atmospheric $O_2$ levels of 9% PAL and 0.6% PAL, respectively.

**Figure 18** Biogeochemical responses obtained from the CANOPS-GRB model with different values of P availability, $f_P$. (**a**) Oceanic phosphate inventory, $M_P$. (**b**) Oceanic net primary production ($NPP_{ocn}$). (**c**) Burial rate of organic carbon ($C_{org}$) in marine sediments. (**d**) Atmospheric partial pressure of $O_2$. PAL = present atmospheric level. (**e**) Atmospheric $CH_4$ mixing ratio. (**f**) Sulfate concentration in the surface ocean layer. (**g**) Sedimentary reservoir size of pyrite sulfur, $PYR$. (**h**) Sedimentary reservoir size of organic carbon, $ORG$. (**i**) Global redox budget, $GRB$. For the $f_P$ = 1% run (black line), the calculation stopped when the atmospheric $O_2$ level decreased to ~$10^{-5}$ PAL due to the numerical instability.




The following change is driven by the response of oceanic S cycle, which is characterized by the long residence
time of 11.5 Myr (see Sect. 3.2.6). Ocean anoxia promotes the MSR and subsequent precipitation of pyrite in
the ocean interior. However, our model demonstrates that the decline of seawater $SO_4^{2-}$ on a timescale of tens
of millions of years is small (Fig. 18f), because the rate of MSR depends not only on the oceanic redox state
but on the availability of organic matter for the MSR. The significant reduction of seawater $SO_4^{2-}$ occurs on
the longer timescales (>100 Myr) for extremely low $f_P$ scenarios (0.016 and 0.01), in which atmospheric $O_2$
levels decrease to <1% PAL. These scenarios also accompany with a growth of sedimentary S from gypsum
to pyrite (Fig. 18g).

On longer timescales, sedimentary reservoirs affect the redox state of the atmosphere and oceans. The present
result demonstrates that $f_P$ of 1% finally leads to the catastrophic decrease in atmospheric $O_2$ level at around
0.9 billion years (Fig. 18d). The simulation was stopped at this point due to the numerical instability. For other
scenarios, the system reaches a new steady state after roughly three billion model years.

Biogenic $CH_4$ production tends to be enhanced in the anoxic oceans. However, the present result demonstrates
that $CH_4$ degassing to the atmosphere is inhibited by both limited availability of organic matter for
methanogenesis and the anaerobic oxidation of $CH_4$ by $SO_4^{2-}$. Once the seawater $[SO_4^{2-}]$ decreases below 1
mM, $CH_4$ can escape from oceans to the atmosphere, promoting the buildup of $CH_4$ in the atmosphere.
Nevertheless, because of the limited biological activity, atmospheric $CH_4$ levels are comparable to the modern
value (~1 ppmv) (Fig. 18e).
**4.2 $O_2$ budget for the less oxygenated scenario**
Figure 19 shows the $O_2$ budget of the less oxygenated state ($f_P = 1.6\%$ scenario). Because P availability exerts
a primary control on biospheric $O_2$ production, the strongly suppressed P delivery to the ocean leads to low
oceanic P levels and commensurately low biological productivity (0.08 Pmol and 1.3 Gt C $yr^{-1}$, respectively).
As a consequence, the atmospheric $O_2$ level is low (0.75% PAL). In this scenario the ocean interior is globally
anoxic, and the preservation of organic C in marine sediments is enhanced. However, the suppressed biological





productivity results in a low overall burial rate of organic C (0.9 Tmol $O_2$ equiv. $yr^{-1}$; ~9% of the reference
value). When combined with the burial rate of terrigenous organic C, total $O_2$ production by the organic C
sub-cycle is 0.97 Tmol $O_2$ equiv. $yr^{-1}$. This $O_2$ source is balanced by the sum of oxidative weathering and
metamorphism. The role of the pyrite S sub-cycle in the global redox budget is also shown in Fig. 19. Most
of the $SO_4^{2-}$ entering the anoxic ocean is buried as pyrite, representing a major $O_2$ source (2.64 Tmol $O_2$ equiv.
$yr^{-1}$). This $O_2$ source is balanced by oxidation of sedimentary pyrite S through weathering (1.33 Tmol $O_2$ equiv.
$yr^{-1}$) and metamorphism (1.31 Tmol $O_2$ equiv. $yr^{-1}$). In other words, the $O_2$ budget for the weakly-oxygenated
Earth system is largely controlled by the crustal S sub-cycle. This is in marked contrast to the well-oxygenated
Earth system, on which the $O_2$ budget is mainly controlled by organic C sub-cycle.

The present result demonstrates that low atmospheric $O_2$ states (~1% PAL) can be achieved in scenarios where
the availability of P is strongly limited. However, a slight decrease of $f_P$ to 1% leads to the destabilization of
global $O_2$ budget, providing implications for the stability and evolution of atmospheric $O_2$ levels during the
Proterozoic. This point will be further systematically examined in future work.

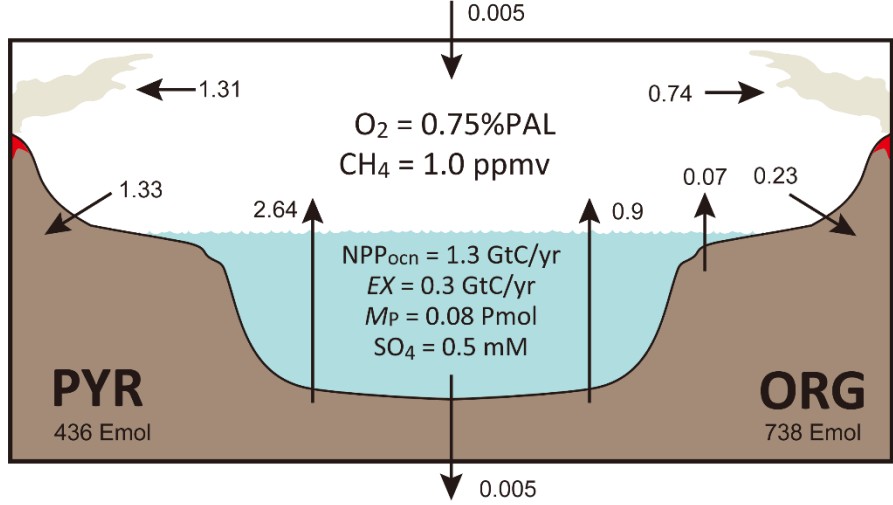
**Figure 19** Schematics of global redox ($O_2$) budget for the $f_P$ = 1.6% (=$10^{-1.8}$) run. Arrows represent the $O_2$
flux in terms of $10^{12}$ mol $O_2$ equiv. $yr^{-1}$. PAL = present atmospheric level. Pmol = $10^{15}$ mol. Emol = $10^{18}$ mol.
Gt C = $10^{15}$ g C. ORG = sedimentary organic C. PYR = sedimentary pyrite S.



## 5 Discussion

The reference run under the present condition demonstrates generally good agreement with modern observations (Sect. 3). The water circulation scheme provides an adequate representation of general ocean circulation, resulting in robust and reliable tracer distributions that are comparable to the modern observations. This provides a mechanistic foundation for simulating generalized ocean biogeochemical cycles. The ocean biogeochemistry module includes a series of biogeochemical processes in oxic-anoxic-sulfidic environments. The reference run gives rise to the distributions of nutrients and dissolved $O_2$ that capture fundamental properties observed in the modern ocean. Integrated biogeochemical fluxes of the global ocean, such as biological productivity, material flow in the water column, and burial into sediments are also consistent with observational data. Some future developments to the N cycle may be needed, especially with regard to denitrification (e.g., anammox, coastal benthic denitrification). Nevertheless, our ocean biogeochemical model is adequate for representing the general property of the coupled C-N-P-$O_2$-S cycles.

A new scheme for oxidative weathering of organic matter and pyrite sulfur, mass balance calculation of $O_2$ in the atmosphere, and time evolution of sedimentary reservoirs are explicitly included in the CANOPS-GRB model. These are a significant improvement from the previous versions of CANOPS (Lenton, 2020). The simplified framework for the global $O_2$ budget is also useful to understand the response of complex biogeochemical systems. The computational efficiency of our CANOPS-GRB model allows us to conduct simulations over billions of model years with reasonable wall times (on the order of weeks), providing a useful tool for exploring the wide range of topics about the oxygenation history of Earth's atmosphere.

Sensitivity experiments with respect to the terrestrial weathering rate of P were conducted in order to see how the redox state of the ocean-atmosphere system responds to varying P availability in the surface system (Sect. 4). The CANOPS-GRB model appears to adequately simulate the biogeochemical dynamics over a wide range of timescales and is applicable for quantitative assessment of the evolution and stability of Earth's $O_2$ cycling. Perhaps even more importantly, our results encourage us to perform further systematic examinations with Earth system models which have different complexities. Such an 'Earth system model intercomparison' would





be a critical step towards better mechanistic understanding of the stability and dynamics of atmospheric $O_2$
levels over Earth's history.

Due to a lack of explicit Fe cycling and anaerobic metabolisms (such as anoxygenic photosynthesis), the
current version of the model cannot be applicable for the simulation under the Archean-like weakly-
oxygenated ($pO_2 < 10^{-5}$ PAL) conditions. These topics are left to future studies, but it would be an achievable
goal (Ozaki et al., 2018; Ozaki et al., 2019b; van de Velde et al., 2021). The model design presented here also
ignores the interaction between the surface system and the mantle (e.g., subduction) except for the degassing
of reducing gases from the mantle. We note, however, that the surface-mantle interaction would have exerted
a primary control on the long-term redox budget of Earth's surface system through the Earth's history
(Canfield, 2004; Eguchi et al., 2020; Hayes and Waldbauer, 2006) and may be important for the discussion
about the distant future (Ozaki and Reinhard, 2021). The importance of mantle and solid Earth controls on
surficial environments is a crucially important topic for future research.



## 6 Conclusions

A new Earth system box model was developed—CANOPS-GRB. The new code release provides an improved description of the coupled C-N-P-$O_2$-S biogeochemical cycles in the ocean-atmosphere-crust system, which can be utilized to examine the dynamics and stability of Earth's $O_2$ cycle over a wide range of timescales. The computational efficiency and simple model design of CANOPS-GRB make it relatively easy to modify existing processes or add entirely new processes and components. CANOPS-GRB is thus a new and uniquely flexible tool capable of providing a coherent mechanistic framework for quantifying the biogeochemical cycles regulating Earth's $O_2$ cycle. CANOPS-GRB is also a useful tool for the development of more comprehensive, low- to intermediate-complexity Earth system box models of biogeochemistry.

CANOPS-GRB will provide an important step forward when coupled to new and existing geochemical proxy data. The accumulating geological/geochemical records have led to new hypothesis for the evolution of atmospheric $O_2$ levels on Earth. CANOPS-GRB was designed to facilitate simulation of a wide range of past conditions so as to permit more explicit testing of hypothesis about the function of biogeochemical cycles and its effect on the redox budget through Earth history. Through the model-data synergy, CANOPS-GRB has a great potential to provide an integrated, quantitative, and statistically informative picture of biogeochemical states, opening new perspectives on a wide range of scientific questions in research seeking to understand the Earth's chemical evolution, and in particular the cause-and-effect relationships with evolving biosphere.

CANOPS-GRB also provides significant steps forward in our predictive understanding of the links between geology, biogeochemistry, and the evolution of Earth's biosphere. It will allow for a fundamentally new and more precise quantitative understanding of evolving atmospheric biosignatures ($O_2$, $O_3$, $CH_4$) on Earth, and will broaden the interpretive power of Earth system evolution in the search for life beyond our planet. CANOPS-GRB could be a central piece for developing a more robust tool for diagnosing atmospheric biosignatures for future analysis of extrasolar worlds. In sum, it is anticipated that CANOPS-GRB will have many applications for problems linking the coupled evolution of life and the atmosphere on Earth and habitable rocky exoplanets.



**Code availability.** The bulk of CANOPS-GRB is written in Fortran as a stand-alone model. The model code can be found at GitHub (doi:10.5281/zenodo.5893804). This model is still undergoing regular development and it is recommended that potential users contact the corresponding author (Kazumi Ozaki; kazumi.ozaki@sci.toho-u.ac.jp) to obtain the latest version. The CANOPS-GRB code is provided freely but with the requirement that prospective users contact the corresponding author with their research plans to avoid parallel projects emerging.

**Author contributions.** KO designed the study, wrote the code, and ran model simulations. DBC and CTR contributed to code debugging. KO wrote the manuscript, with inputs from DBC, CTR, and ET.

**Competing interests.** The authors declare that they have no conflict of interest.

**Acknowledgements.** This work was supported by JSPS KAKENHI Grant Number JP16K05618, 19K21055, and JP20K04066. CTR acknowledges the NASA Exobiology Program (Grant Number 80NSSC19K0461) and the NASA Interdisciplinary Consortia for Astrobiology Research (ICAR) (Grant Number 80NSSC21K0594).





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
