# Peer review of "CANOPS-GRB v1.0: a new Earth system model for simulating the evolution of ocean-atmosphere chemistry over geologic timescales"

_Geoscientific Model Development, 2022_

## Author Comment (AC1)

**Response to Reviewer 1**:

Review of "CANOPS-GRB v1.0: a new Earth 1 system model for simulating the evolution of ocean-atmosphere chemistry over geologic timescales"

This paper describes a new version of the CANOPS Earth system model, CANOPS-GRB (Global Redox Budget). The main improvements of this model over previous versions include dynamic atmospheric $O_2$, sedimentary reservoirs that evolve self-consistently with the atmosphere-ocean system, and a simplified global redox budget. Collectively, these improvements make the model suitable for investigating long timescale geochemical evolution, but with the benefits of resolving ocean transport and sediment diagenesis. Model outputs are validated against the observed modern Earth steady state, and example time-evolution calculations are also presented to illustrate feedbacks on multiple timescales.

I will preface this review by saying evaluating the validity of model assumptions in the abstract is challenging; the appropriateness of the chosen functions, parameterizations, and simplifications are strongly context dependent. However, I have no doubt that this manuscript and open-source model it describes will be incredibly valuable to the biogeochemistry community. The authors ought to be commended for such a thorough description of model features – this will be invaluable for trying to understand future applications of the model. I don't have any major criticisms – the primary limitations of the model are largely already identified and discussed in the manuscript. However, I do have several minor suggestions for improvement:

We are grateful to Reviewer 1 for the useful suggestions and critical comments, which have helped us to significantly improve our paper. As indicated in the responses that follow, we have taken all these comments and suggestions into account in the process of revision. We are pleased to say that the comments and suggestions have both greatly improved the quality of the manuscript.

The Introduction motivates the CANOPS-GRB model through a discussion of terrestrial exoplanets and exoplanet biosignatures such as oxygen and methane. Moreover, the abstract mentions exoplanet biogeochemistry as one potential application of CANOPS-GRB. It would be helpful if the discussion also had a paragraph or two on both the potential and limitations of the current model for terrestrial exoplanets. For example, by adopting the Claire et al. photochemical parameterization and by omitting a climate model, the ability to investigate O2-CH4 biosignatures is limited to Earth-twins around a sun-like stars (see also line 351). Obviously, these parameterizations could be modified to accommodate different stellar types etc., but I think it is helpful to make the current limitations of the model clear to the exoplanet community.

This is an important point. We have added brief discussion to the revised Discussion section to clarify this more explicitly (Line 1681-1686): "The CANOPS-GRB model has the basic capability

to simulate the time evolution of the abundance of atmospheric biosignature gases ($O_2$ and $CH_4$) on a wide range of timescales. While the biogeochemical model is based on process studies to the extent possible, many processes are derived from empirical calibrations to Earth-like planets around sun-like stars. Clearly, some these parameterizations, such as the photochemical parameterization among $O_2$-$O_3$-$CH_4$ (Eq. 107), must be modified when applying the model to a range of habitable Earth-like exoplanets."

Line 229-230: I appreciate that the omission of $Fe^{2+}$ is discussed later in the paper, but it might be worth mentioning this here (e.g. see discussion of iron species below) since upon first reading I immediately wondered why the iron budget was being ignored.

This is also an important point. We have added a brief sentence to make this point clear (Line 235-236) and appreciate this opportunity to clarify this point.

Line 291-292: Neglecting the inorganic carbon cycle is an interesting choice that merits more discussion. To be clear, I understand the reasoning behind this simplification – carbon cycle feedbacks are uncertain and introduce more unconstrained complexity to an already complex model. However, changes in inorganic carbon cycling have been proposed as drivers of atmospheric oxygenation (e.g. Williams et al. 2019; Nature Communications), and the 13C record provides a useful sanity check on any proposed oxygenation story. Neglecting the inorganic carbon cycle also means climate feedbacks are absent, which prevents the model from exploring the coevolution of life and the environment in many deep time contexts. More discussion of these limitations would be helpful in preventing misapplications or misunderstandings of the model.

We agree with the reviewer. We have added explanations behind this simplification in the revised manuscript (Line 301-303): "Although we ignore the inorganic carbon cycle here for clarity and in order to simplify interpretation of model results, the full coupling of the inorganic carbon cycle within CANOPS-GRB is an important topic for future work."

Line 329: 14.3 Tmol/yr total organic carbon burial feels a bit on the high side. For example, see the compilation of estimates in Table 1 of Kipp et al. (2020; Global Biogeochemical Cycles). Similarly, organic weathering in CANOPS-GRB is about double most literature estimates. The discrepancy is noted on line 1368, but I am left wondering why such a high organic carbon flux is required. Given isotope mass balance, a high organic burial flux would also imply quite a high total carbon burial flux, which potentially conflicts with other literature estimates.

The reviewer is correct to point out that our estimate of organic C burial is relatively high compared to previous estimates (though some key estimates that are similar to ours are left out of the Kipp et al. summary (e.g., Wallmann and Aloisi, 2012; Wallmann et al., 2012)). To clarify this point, we have added a brief discussion to the revised Discussion section (Line 1622-1624): "In our

reference run the estimated organic carbon burial and oxidative weathering fluxes are relatively high compared to some previous estimates, though there remains significant uncertainty in globally integrated organic carbon weathering and burial fluxes."

Wallmann et al. (2012). The global inventory of methane hydrate in marine sediments: A theoretical approach. *Energies*, 5, 2449.

Wallmann, K. and Aloisi, G. (2012). The Global Carbon Cycle: Geological Processes. In *Fundamentals of Geobiology* (eds A.H. Knoll, D.E. Canfield and K.O. Konhauser).

Equation 2: This is another place where it might be worth mentioning that climate feedbacks on silicate weathering are an important omission that could have consequences for global redox.

We have added a brief sentence in Line 396-399: "Note that our representation of P weathering ignores the effect of climate (Eq. 2). In the current version of the model the rate of P weathering is treated as one of the model forcings. Although ignoring the climate feedback on P mobility makes interpretation of the model results more straightforward, the incorporation of a climate-sensitive crustal P cycle is an important avenue for future work."

Line 458: Although this is stated later in the manuscript, this discussion of sulfur oxidation might be a good place to mention that the model in its current form is not suitable for modeling the Archean atmosphere-ocean system.

We added a brief sentence in the revised manuscript (Line 236-239): "All $H_2S$ and $NH_3$ degassing from the ocean to the atmosphere is assumed to be completely oxidized by $O_2$ to $SO_4^{2-}$ and $NO_3^-$ and returns to the ocean surface. These simplifications limit application of the model to very poorly oxygenated Earth system states ($pO_2 < 10^{-5}$ PAL)."

Section 2.4.2: I appreciate that the vertical and horizontal transport parameterizations developed here are carefully validated later in the paper. However, one unanswered question I had is how well do the authors expect these parameterizations to work when they are applied to radically different surface climates in deep time (e.g. hothouse Earth, snowball Earth)? Some discussion of this might be helpful for readers considering applying the model to different times in Earth's history (or indeed different exoplanet climates, continental configurations etc.)

Our water transport scheme is flexible enough to represent the general water circulation, such as thermohaline circulation and high-latitude convection, on any rotating planet, with a limited set of free parameters. However, the reviewer is correct to point out that the users need to be careful when applying the model to the ancient oceans or oceans on exoplanets. We added brief explanations of this point in the revised manuscript (Line 640-646): "Thermohaline circulation

and high-latitude convection are considered to be general physical modes on any rotating planet, and our simplified water transport scheme allows us to represent them with a limit number of free parameters. However, we emphasize that the water transport scheme explored here is designed to represent the modern ocean circulation on Earth. As a result, some of these parameterizations may need to be modified when applying to ancient oceans or oceans on exoplanets. Nevertheless, given our simple design, our water transport scheme is relatively flexible to modify the water circulations that are markedly different from the modern ocean."

Equation 108 and 109: It's possible I'm misunderstanding a sign convention here, but it's not clear to me why H escape represents a loss of oxygen mass from the atmosphere system. The signs in equations (13) and (14) seem more intuitive. A word of explanation would be helpful.

The photochemical oxidation of $CH_4$ by $O_2$ in the atmosphere is represented as

$$CH_4 + 2O_2 + hv \rightarrow 2H_2O + CO_2. \text{ (E1)}$$

The sink of $CH_4$ and $O_2$ has a stoichiometric ratio of 1:2. However, when we consider hydrogen escape to space and following oxidation of C to $CO_2$, the net reaction can be represented as (e.g., Goldblatt et al., 2006 *Nature*):

$$CH_4 + O_2 + hv \rightarrow 4H \uparrow + CO_2. \text{ (E2)}$$

The sink of $CH_4$ and $O_2$ has a stoichiometric ratio of 1:1. Thus, the hydrogen escape to space halves the amount of $O_2$ consumed in the oxidation of $CH_4$, representing a net $O_2$ production. In terms of the redox budget, the loss of hydrogen in equation (E2) represents the gain of the oxidizing power equivalent to one molecule of $O_2$: $2H_2 + O_2 = 2H_2O$.

In equation (13), the net impact on the $O_2$ budget is represented, whereas internal source/sink fluxes of $O_2$ in the atmosphere are explicitly represented in equation (109). Both expressions are mutually consistent.

To clarify this point, we added a brief explanation of this to the revised manuscript (Line 1296-1299): "Note that the hydrogen escape to space is represented as the $O_2$ sink, because the hydrogen escape via $CH_4$ followed by the oxidation of carbon to $CO_2$ is represented as:

$CH_4 + O_2 + hv \rightarrow 4H \uparrow + CO_2$.

On the other hand, the photochemical oxidation of $CH_4$ can be written, as follows:

$CH_4 + 2O_2 + hv \rightarrow 2H_2O + CO_2$.

Thus, the hydrogen escape to space represents the net gain of oxidizing power to the system (see Eq. (13))."

Finally, somewhere in the paper (perhaps in the introduction) it would be helpful to provide a brief history of the CANOPS model and its previous applications. This would help give the reader a better appreciation of the recent GRB improvements, as well as how the improved model might be applied.

We have added a brief explanation of this to the revised Introduction section (Line 110-114), and appreciate the opportunity to further clarify this point: "The core of this model is an ocean biogeochemical model, CANOPS (Ozaki et al., 2011; Ozaki and Tajika, 2013; Ozaki et al., 2019a). This model has been used to examine conditions for the development of widespread oceanic anoxia/euxinia during the Phanerozoic (Ozaki et al., 2011; Ozaki and Tajika, 2013; Kashiyama et al., 2011) and to quantitatively constrain biogeochemical cycles during the Precambrian (Cole et al., 2022; Ozaki et al., 2019a; Ozaki et al., 2019b; Reinhard et al., 2017b)."

**Additional changes:**

   – Various minor changes in wording and corrections throughout the manuscript.

We hope that we have adequately addressed all the suggestions raised by Reviewer 1, and appreciate their constructive feedback.

---

## Author Comment (AC2)

**Response to Reviewer 2**:

This carefully prepared paper describes an Earth System model of intermediate complexity that captures Earth's C, N, P, O, and S cycles. The model is well-described and two relevant applications are presented to demonstrate the capabilities of the model. The model is particularly useful for simulations on long (geological) time scales. The model code is made available allowing the work to be reproduced (but see additional comment below).

I read this paper with pleasure and have relatively few comments. Most of the detailed comments below are suggestions to rephrase the text to improve the clarity. The manuscript is quite long but because it is well-written, I don't think that is a problem.

We are grateful to Reviewer 2 for the useful suggestions and comments, which have helped us to significantly improve our paper. As indicated in the responses that follow, we have taken all these comments and suggestions into account in the process of revision. We are pleased to say that the comments and suggestions have both greatly improved the quality of the manuscript.

Detailed comments

1. Lines 25 and 1650. The authors write that "The model source code is available on GitHub and represents a unique community tool" and later "The CANOPS-GRB code is provided freely but with the requirement that prospective users contact the corresponding author with their research plans….". I don't think the latter requirement is appropriate. Other scientists should be free to use the model without the author's consent or knowledge. That's open science.

   We agree. The sentence has been removed from the revised manuscript for the sake of open science.

2. Line 151. "preferred" instead of "required" since this will depend on the application.

   Corrected.

3. Line 173. Suggested change: "of the Earth system"

   Corrected.

4. Line 258: "to describe the overall design of the biogeochemical cycles"

   Corrected.

5. Figure 2. caption needs editing. Line 267: "anaerobic" instead of "anoxygenic"

   Corrected.

6. Line 278 and 283: "are transformed each other" needs rephrasing.

We have modified the corresponding sentences: "The abundance of inorganic nitrogen species (ammonium and nitrate), which are lumped into DIN (dissolved inorganic nitrogen), is affected by denitrification and nitrification.", "The reservoir sizes of sedimentary sulfur (pyrite sulfur, *PYR*, and gypsum sulfur, *GYP*) and two sulfur species ($SO_4^{2-}$ and $\Sigma H_2S$) in the ocean are controlled by volcanic outgassing, weathering, burial, MSR, AOM, and sulfide oxidation reactions."

7. Line 312: change to "the water column"

Corrected.

8. Line 318. "2x60 sediment segments." Can you explain why this is two times 60?

The ocean interior below the photic zone is divided into two sectors, HD and LD. The each sector is vertically subdivided into 60 layers. We have added a brief explanation about this (Line 329), and are grateful for the opportunity to clarify this point: "HD and LD have 60 layers, respectively"

9. Table 2: Better to use the term "Fe bound P" since a lot of the P is not sorbed and instead is more strongly bound.

Corrected.

10. Table 2: typo in "simulated" and "denitrification"

Corrected.

11. Line 344. AOM is more commonly referred to as "anaerobic oxidation of methane"

We agree and have thoroughly re-examined the use of the terms throughout the revised manuscript.

12. Line 345. "as a CH4 degassing flux"

Corrected.

13. Line 399-401. There is more work that is relevant in this context. You could cite Algeo and Ingall (2007) since they clearly showed the role of hydrogen sulfide in P retention and/or Papadomanolaki et al. (2022; Science Advances) who show a potential role for acidification and warming.

We thank the reviewer for bringing these relevant papers to our attention. We referred these papers in the revised manuscript to better clarify this point.

14. Line 544: change to "papers"

Corrected.

15. Line 797. Here, methanogenesis is termed "organic matter production from carbon dioxide" whereas this is a degradation process. Needs to be reformulated.

Corrected.

16. Line 802: remove "most"

Corrected.

17. Line 803: You write: "oxidants for organic matter decomposition change gradually, depending on the amount of each oxidant". Gradually with time? Or depth? Any change will also depend on the supply of reductants.

The reviewer is correct to point out that our original wording may be unclear as used in the original manuscript. We modified the sentence and are grateful for the opportunity provided by the reviewer to clarify this point: "The oxidants for organic matter decomposition change with the availability of each oxidant, which vary with time and water depth."

18. Line 804: "on previous studies of early…"

Corrected.

19. Line 817: change to "in the geological past"

Corrected.

20. Line 821: "oxidation to" not "oxidation by"

Corrected.

21. Equation (32) remove the dot.

We keep the period to express the end of the sentence.

22. Line 831: change to "using a bimolecular.."

Corrected.

23. Line 834: by definition, the suboxic layer does not contain hydrogen sulfide so it would be better to reformulate.

The reviewer is correct to point out that our use of term of "suboxic layer" may be unclear. We corrected it to "chemocline".

24. Line 844 Sentence needs reformulation for clarity

We deleted the sentence.

25. Line 858. Change to "are listed in"

Corrected.

26. Figure 6: the line corresponding to "This study (oxic)" is not visible in the figure.

Corrected.

27. Line 925. Here the authors write "such as" but it appears that these are the only terminal electron acceptors considered.

We deleted the sentence to avoid the confusion.

28. Line 960 See earlier comment on "Fe-sorbed P"

We have thoroughly re-examined the use of the terms "Fe-sorbed P" throughout the revised manuscript.

29. Line 1011-. Better to write "sediments overlain by oxic bottom waters". This will only hold for sediments with a relatively low input of organic matter. Otherwise the Fe-bound P will be lost, as observed in many modern coastal sediments.

Corrected.

30. Line 1021-1025. See earlier comment.

We referred Papadomanolaki et al. paper in the revised manuscript.

31. Line 1270 change to "until reaching steady state"

Corrected.

32. Line 1291 change to "LD and HD regions"

Corrected.

33. Line 1380-1381. It's not clear what is meant by "(Gray dots represent the unknown dissolved O2 concentrations)".

We modified the corresponding sentence, as follows: "The color represents the $O_2$ concentration of bottom water, $[O_2]_{bw}$, with grey dots for the unknown $[O_2]_{bw}$ value."

34. Figure 13b and lines 1404 and further. The authors focus on the benthic flux of P here but this gives a very large spread – as is commonly the case with benthic fluxes. No attempt was made to compile P burial fluxes from the literature. This needs some justification in the text, i.e. were there insufficient data to compare to or was there another reason?

This is a good point. The primary reason was a fact that the observation of the burial flux of reactive P is rare. We can find total P burial flux in the literature, but it is expected to contain detritus P, not bioavailable P. The sequestration methods, SEDEX, can distinguish reactive P from total P, but such observation is rare even in the modern ocean (as far as we know). To clarify this point, we added a sentence in the revised manuscript (Line 1369-1370): "The burial flux density of reactive P is not shown due to the sparseness of such observations."

35. Lines 1450. Here you mention the role of the coastal ocean for N cycling. This is also relevant for the P cycle, since most P burial takes place in the coastal ocean. This could be mentioned somewhere in the text.

This is also a good point. We have recognized that the separate treatment of coastal sea from the open ocean affects the strength of the redox-dependent P feedback. We added a brief sentence for this (Line 1624-1625): "Further work will also be needed to better quantify the biogeochemical cycling in the continental shelf, which is a major locus of organic matter burial." We also added a brief explanation in Line (1055-1057): "Given that the continental

shelves are a main locus of reactive P burial, the separate treatment of continental shelves and margin sediments from the pelagic ocean could affect the non-linearity of the redox-dependent P cycle. However, this was left as one of the subjects of future work."

36. Line 1508. "….leads to a lower"

   Corrected.

37. Line 1511. "leads"

   Corrected.

**Additional changes:**

– Various minor changes in wording and corrections throughout the manuscript.

We hope that we have adequately addressed all the suggestions raised by Reviewer 2, and appreciate their constructive feedback.

---

## Author Response (AR2)

**Response to Topical Editor**:

Dear Kazumi Ozaki and coauthors,

Thank you for your extensive revisions and response. All but one response fully addresses what the reviewers have asked for, so after a minor revision to the response to this one point I would be pleased to accept this manuscript.

The comment I would like you to look at again is:

Line 291-292: Neglecting the inorganic carbon cycle is an interesting choice that merits more discussion. To be clear, I understand the reasoning behind this simplification – carbon cycle feedbacks are uncertain and introduce more unconstrained complexity to an already complex model. However, changes in inorganic carbon cycling have been proposed as drivers of atmospheric oxygenation (e.g. Williams et al. 2019; Nature Communications), and the 13C record provides a useful sanity check on any proposed oxygenation story. Neglecting the inorganic carbon cycle also means climate feedbacks are absent, which prevents the model from exploring the coevolution of life and the environment in many deep time contexts. More discussion of these limitations would be helpful in preventing misapplications or misunderstandings of the model.

Your text change ("Although we ignore the inorganic carbon cycle here for clarity and in order to simplify interpretation of model results, the full coupling of the inorganic carbon cycle within CANOPS-GRB is an important topic for future work.") does not constitute a discussion of the limitations as the reviewer has suggested. I agree with the reviewer that this discussion is necessary.

I look forward to seeing this minor amendment to the manuscript.

We are most grateful to the Topical Editor for their handling of our manuscript and for the helpful comment on the revised manuscript. Based on the comment above, we have revised the text to more clearly explain the limitations of the model (Line 302-309): "Atmospheric $CO_2$, dissolved inorganic carbon (DIC), and dissolved organic carbon (DOC) are not explicitly modelled in the current version of the model, and the full coupling of the inorganic carbon cycle within CANOPS-GRB is left as an important topic for future work. Neglecting the inorganic carbon cycle means that there are no climatic feedbacks in the system, and because of this simplification, the CANOPS-GRB model cannot be applied to problems such as those in which the Earth's climate and redox states of the ocean-atmosphere system are closely related each other or to validate model predictions based on geologic records (such as $\delta^{13}C$), but this allows us to avoid introducing the additional complexities and uncertainties in the model."

We hope that our explanations and revisions are satisfactory and that the revised version of our paper is now suitable for publication in the *Geoscientific Model Development*.

---

## Author Response (AR3)

**Response to Topical Editor**:

Thank you for your edits. I'm very pleased to accept the manuscript for publication.

Please make one final technical change to address the issue below:

The title page of *pdf. manuscript file must include the full institutional addresses of all authors. However, country and city names are missing from the affiliation #2.

We have included the full institutional addresses of all authors. We hope that the revised version of our paper is now suitable for publication in the *Geoscientific Model Development*.

Yours sincerely (on behalf of all authors),

Kazumi Ozaki